# BCE vs. CE in Deep Feature Learning

Qiufu Li [1 2 3]   Huibin Xiao [1 2 3]   Linlin Shen [1 2 3]

## Abstract

When training classification models, it expects that the learned features are compact within classes, and can well separate different classes. As the dominant loss function for training classification models, minimizing cross-entropy (CE) loss maximizes the compactness and distinctiveness, i.e., reaching neural collapse (NC). The recent works show that binary CE (BCE) performs also well in multi-class tasks. In this paper, we compare BCE and CE in deep feature learning. For the first time, we prove that BCE can also maximize the intra-class compactness and inter-class distinctiveness when reaching its minimum, i.e., leading to NC. We point out that CE measures the relative values of decision scores in the model training, implicitly enhancing the feature properties by classifying samples one-by-one. In contrast, BCE measures the absolute values of decision scores and adjust the positive/negative decision scores across all samples to uniformly high/low levels. Meanwhile, the classifier biases in BCE present a substantial constraint on the decision scores to explicitly enhance the feature properties in the training. The experimental results are aligned with above analysis, and show that BCE could improve the classification and leads to better compactness and distinctiveness among sample features. The codes have be released.

## 1. Introduction

Cross-entropy (CE) loss is the most commonly used loss for classifications and feature learning. In a classification with $K$ categories, for any sample $\boldsymbol{X}^{(k)}$ from category $k$, a model

[1]School of Artificial Intelligence, Shenzhen University, Shenzhen, 518060, China [2]Department of Computer Science, Wenzhou-Kean University, Wenzhou, 325060, China [3]Guangdong Provincial Key Laboratory of Intelligent Information Processing, Shenzhen University, Shenzhen, 518060, China. Correspondence to: Linlin Shen <llshen@szu.edu.cn>.

*Proceedings of the 42$^{nd}$ International Conference on Machine Learning*, Vancouver, Canada. PMLR 267, 2025. Copyright 2025 by the author(s).

$\mathcal{M}$ extracts its feature $\boldsymbol{h}^{(k)} = \mathcal{M}(\boldsymbol{X}^{(k)}) \in \mathbb{R}^d$, which is output from the penultimate layer in deep model. Then a linear classifier with weight $\boldsymbol{W} = [\boldsymbol{w}_1, \cdots, \boldsymbol{w}_K]^T \in \mathbb{R}^{K \times d}$ and bias vector $\boldsymbol{b} = [b_1, \cdots, b_K]^T \in \mathbb{R}^K$ transforms the feature into $K$ logits/decision scores, $\{\boldsymbol{w}_j^T \boldsymbol{h}^{(k)} - b_j\}_{j=1}^K$, which are finally converted into predicted probabilities by Softmax, and computed the loss using cross-entropy,

$$\mathcal{L}_{\text{ce}}\big(\boldsymbol{z}^{(k)}\big) = \log\left(1 + \sum_{\substack{\ell=1 \\ \ell \neq k}}^K \frac{\mathrm{e}^{\boldsymbol{w}_\ell^T \boldsymbol{h}^{(k)} - b_\ell}}{\mathrm{e}^{\boldsymbol{w}_k^T \boldsymbol{h}^{(k)} - b_k}}\right), \qquad (1)$$

where $\boldsymbol{z}^{(k)} = \boldsymbol{W} \boldsymbol{h}^{(k)} - \boldsymbol{b} \in \mathbb{R}^K$.

For the multi-class classification, binary CE (BCE) loss is deduced by decomposing the task into $K$ binary tasks and predicting whether the sample $\boldsymbol{X}^{(k)}$ belongs to the $j$th category, for $\forall j \in [K] = \{1, 2, \cdots, K\}$,

$$\mathcal{L}_{\text{bce}}\big(\boldsymbol{z}^{(k)}\big) = \log\left(1 + \mathrm{e}^{-\boldsymbol{w}_k^T \boldsymbol{h}^{(k)} + b_k}\right) + \sum_{\substack{j=1 \\ j \neq k}}^K \log\left(1 + \mathrm{e}^{\boldsymbol{w}_j^T \boldsymbol{h}^{(k)} - b_j}\right), \qquad (2)$$

which has been widely used in the multi-label classification (Kobayashi, 2023) and attracted increasing attentions in the multi-class classification (Beyer et al., 2020; Fang et al., 2023; Touvron et al., 2022; Wen et al., 2022; Wightman et al., 2021; Zhou et al., 2023).

The pre-trained classification models can be used as feature extractors for downstream tasks that request well intra-class compactness and inter-class distinctiveness across the sample features, such as person re-identification (He et al., 2021), object tracking (Cai et al., 2023), image segmentation (Guo et al., 2022), and facial recognition (Wen et al., 2022), etc. For CE, a remarkable theoretical result is that when it reaches its minimum, both the compactness and distinctiveness on the training samples will be maximized, which refers to neural collapse (NC) found by Papyan et al. (2020). NC gives peace of mind in training classification models by using CE, and it was extended to the losses satisfying contrastive property by Zhu et al. (2021) and Zhou et al. (2022b), including focal loss, and label smoothing loss. However, while BCE in Eq. (2) is a linear combination of CEs, it is not sufficient to guarantee that BCE satisfies the

contrastive property, as this property is not a linear property, and it remains unclear whether BCE can lead to NC.

Besides that, in the practical training of classification models, the classifiers $\{w_k\}_{k=1}^K$ play the role of proxy for each category (Wen et al., 2022). Intuitively, when the distances between the sample features and their class proxy are closer, or the *positive* decision scores between them are larger, it usually leads to better intra-class compactness. Similarly, when the distances between sample features and the proxy of different classes are farther, or the *negative* decision scores between them are smaller, it could results in better inter-class distinctiveness. However, according to Eq. (1), CE measures the *relative* value between the exponential positive and negative decision scores using Softmax and logarithmic functions, to pursue that the positive decision score is greater than all its negative ones for each sample, making it unable to explicitly and directly enhance the feature properties across samples. In contrast, BCE in Eq. (2) respectively measures the *absolute* values of the exponential positive decision score and the exponential negative ones using Sigmoid and logarithmic functions, which makes it is possible to explicitly and directly enhance the compactness and distinctiveness of features in the training.

In this paper, we compare BCE and CE in deep feature learning. We primarily address two questions: **Q1**. Can BCE result in NC, i.e., maximizing the compactness and distinctiveness in theoretical? **Q2**. In practical training of classification models, does BCE perform better than CE in terms of the feature compactness and distinctiveness? Our contributions are summarized as follows.

(1) We provide the first theoretical proof that BCE can also lead to the NC, i.e., maximizing the intra-class compactness and inter-class distinctiveness.

(2) We find that BCE performs better than CE in enhancing the compactness and distinctiveness across sample features, and, BCE can explicitly enhance the feature properties, while CE only implicitly enhance them.

(3) We reveal that in training with BCE, the classifier biases play a substantial role in enhancing the feature properties, while they almost do not work in that with CE.

(4) We conduct extensive experiments, and find that, compared to CE, BCE can more quickly lead to NC on the training dataset and achieves better feature compactness and distinctiveness, resulting in higher classification performance on the test dataset.

## 2. Related works

### 2.1. CE vs. BCE

The CE loss is the most popular loss used in the multi-class classification and feature learning, which has been evolved into many variants in different scenarios, such as focal loss (Lin et al., 2017), label smoothing loss (Szegedy et al., 2016), normalized Softmax loss (Wang et al., 2017), and marginal Softmax loss (Liu et al., 2016), etc. The classification models are often applied to the downstream tasks, such as image segmentation (Guo et al., 2022), person re-identification (He et al., 2021), object tracking (Cai et al., 2023), etc., which request well intra-class compactness and inter-class distinctiveness among the sample features. For the multi-class task, the BCE loss can be deduced by decomposing the task into $K$ binary tasks and adding the $K$ naive binary CE losses, which has been widely applied in the multi-label classification (Kobayashi, 2023).

BCE and CE are expected to train the models to fit the sample distribution. When Wightman et al. (2021) applied BCE to the training of ResNets for a multi-class task, they considered that the loss is consistent with Mixup (Zhang et al., 2018) and CutMix (Yun et al., 2019) augmentations, which mix multiple objects from different samples into one sample. DeiT III (Touvron et al., 2022) adopted this approach and achieved a improvement in the multi-class task on ImageNet-1K by using BCE loss. Currently, though CE loss dominates the training of multi-class and feature learning models, BCE loss is also gaining more attention and is increasingly being applied in the fields (Chun, 2024; Fang et al., 2023; Hao et al., 2024; Mehta & Rastegari, 2023; Wang et al., 2023; Xu et al., 2023). However, none of these works reveals the essential advantages of BCE over CE.

### 2.2. Neural collapse

Neural collapse (NC) was first found by Papyan et al. (2020), referring to admirable properties about the sample features $\{h_i^{(k)}\}$ and classifiers $\{w_k\}$ at the terminal phase of training.

- **NC1**, within-class variability collapse. Each feature $h_i^{(k)}$ collapse to its class center $\bar{h}^{(k)} = \frac{1}{n_k} \sum_{i'=1}^{n_k} h_{i'}^{(k)}$, indicating the *maximal intra-class compactness*

- **NC2**, convergence to simplex equiangular tight frame. The set of class centers $\{\bar{h}^{(k)}\}_{k=1}^K$ form a simplex equiangular tight frame (ETF), with equal and maximized cosine distance between every pair of feature means, i.e., the *maximal inter-class distinctiveness*.

- **NC3**, convergence to self-duality. The class center $\bar{h}^{(k)}$ is ideally aligned with the classifier vector $w_k$.

The current works about NC (Kothapalli, 2023) are focused on the CE loss (Graf et al., 2021; Lu & Steinerberger, 2022; Zhu et al., 2021) and mean squared error (MSE) loss (Han et al., 2022; Tirer & Bruna, 2022; Zhou et al., 2022a). It has been proved that the models will fall to NC when the loss reaches its minimum. A comprehensive analysis (Zhou et al., 2022b) for various losses, including CE, focal loss (Lin et al., 2017), and label smoothing loss (Szegedy et al.,

2016), shows that they perform equally as any global minimum point of them satisfies NC. The NC has also been investigated in the imbalanced datasets (Fang et al., 2021; Wang et al., 2024; Yang et al., 2022), out-of-distribution data (Ammar et al., 2024), contrastive learning (Xue et al., 2023), and models with fixed classifiers (Kim & Kim, 2024; Yang et al., 2022). All these studies are conducted on CE or MSE; and whether BCE can lead to NC remains unexplored.

## 3. Main results

In this section, we first theoretically prove that BCE can maximize the compactness and distinctiveness when reaching its minimums (**Q1**). Then, through in-depth analyzing the decision scores in the training, we explain that BCE can better enhance the compactness and distinctiveness of sample features in practical training (**Q2**).

### 3.1. Preliminary

Let $\mathcal{D} = \bigcup_{k=1}^{K} \bigcup_{i=1}^{n_k} \{\boldsymbol{X}_i^{(k)}\}$ be a sample set, where $\boldsymbol{X}_i^{(k)}$ is the $i$th sample of category $k$, $n_k$ denotes the sample number of this category, and $\boldsymbol{h}_i^{(k)} = \mathcal{M}(\boldsymbol{X}_i^{(k)})$. In classification tasks, a linear classifier with vectors $\{\boldsymbol{w}_k\}_{k=1}^{K} \subset \mathbb{R}^d$ and biases $\{b_k\}_{k=1}^{K} \subset \mathbb{R}$ predicts the category for each sample according to its feature. For the well predication results, the CE or BCE loss is applied to tune the parameters of the model $\mathcal{M}$ and classifier parameters.

Following the previous works (Fang et al., 2021; Graf et al., 2021; Han et al., 2022; Lu & Steinerberger, 2022; Tirer & Bruna, 2022; Zhu et al., 2021) for neural collapse (NC), we compare CE and BCE in training of unconstrained model or layer-peeled model in this paper, i.e, treating the features $\bigcup_{k=1}^{K} \{\boldsymbol{h}_i^{(k)}\}_{i=1}^{n_k}$, classifier vectors $\{\boldsymbol{w}_k\}_{k=1}^{K}$, and classifier biases $\{b_k\}_{k=1}^{K}$ as free variables, without considering the sophisticated structure or the parameters of the model $\mathcal{M}$. Then, taking the regularization terms on the variables, the CE or BCE loss is

$$f_\mu(\boldsymbol{W}, \boldsymbol{H}, \boldsymbol{b}) = \frac{1}{\sum_{k=1}^{K} n_k} \sum_{k=1}^{K} \sum_{i=1}^{n_k} \mathcal{L}_\mu(\boldsymbol{W}\boldsymbol{h}_i^{(k)} - \boldsymbol{b})$$
$$+ \frac{\lambda_{\boldsymbol{W}}}{2} \|\boldsymbol{W}\|_F^2 + \frac{\lambda_{\boldsymbol{H}}}{2} \|\boldsymbol{H}\|_F^2 + \frac{\lambda_{\boldsymbol{b}}}{2} \|\boldsymbol{b}\|_2^2, \quad (3)$$

where $\mathcal{L}_\mu$ is presented in Eqs. (1-2), $\mu \in \{\text{ce}, \text{bce}\}$,

$$\boldsymbol{W} = \begin{bmatrix} \boldsymbol{w}_1, \boldsymbol{w}_2, \cdots, \boldsymbol{w}_K \end{bmatrix}^T \in \mathbb{R}^{K \times d}, \quad (4)$$

$$\boldsymbol{b} = [b_1, b_2, \cdots, b_K]^T \in \mathbb{R}^K, \quad (5)$$

$$\boldsymbol{H} = \begin{bmatrix} \boldsymbol{H}_1, \boldsymbol{H}_2, \cdots, \boldsymbol{H}_K \end{bmatrix} \in \mathbb{R}^{d \times (\sum_{k=1}^{K} n_k)}, \quad \text{with}$$

$$\boldsymbol{H}_k = \begin{bmatrix} \boldsymbol{h}_1^{(k)}, \boldsymbol{h}_2^{(k)}, \cdots, \boldsymbol{h}_{n_k}^{(k)} \end{bmatrix} \in \mathbb{R}^{d \times n_k}, \forall k \in [K], \quad (6)$$

and $\lambda_{\boldsymbol{W}}, \lambda_{\boldsymbol{H}} > 0, \lambda_{\boldsymbol{b}} \geq 0$ are weight decay parameters for the regularization terms.

### 3.2. Neural collapse with CE and BCE losses

On the balanced dataset, i.e., $n = n_k, \forall k \in [K]$, Zhu et al. (2021) proved that the CE loss can result in neural collapse (NC), and in Theorem 3.1, Zhou et al. (2022b) extended the proof to the losses satisfying the contrastive property (see Definition C.1 in supplementary), such as focal loss and label smoothing loss. Though BCE loss is a combination of CE, there is no evidence to suggest it satisfies the contrastive property, as this property is not a simple linear one. Despite that, we prove that BCE can result in NC, i.e., Theorem 3.2. The primary difference between BCE and CE losses lies in the bias parameter $\boldsymbol{b}$ of their classifiers.

**Theorem 3.1.** *(Zhou et al., 2022b) Assume that the feature dimension $d$ is larger than the category number $K$, i.e., $d \geq K - 1$, and $\mathcal{L}_\mu$ is satisfying the contrastive property. Then any global minimizer $(\boldsymbol{W}^\star, \boldsymbol{H}^\star, \boldsymbol{b}^\star)$ of $f_\mu(\boldsymbol{W}, \boldsymbol{H}, \boldsymbol{b})$ defined using $\mathcal{L}_\mu$ with Eq. (3) obeys the following properties,*

$$\|\boldsymbol{w}^\star\| = \|\boldsymbol{w}_1^\star\| = \|\boldsymbol{w}_2^\star\| = \cdots = \|\boldsymbol{w}_K^\star\|, \quad (7)$$

$$\boldsymbol{h}_i^{(k)\star} = \sqrt{\frac{\lambda_{\boldsymbol{W}}}{n\lambda_{\boldsymbol{H}}}} \boldsymbol{w}_k^\star, \ \forall\, k \in [K], \ i \in [n], \quad (8)$$

$$\tilde{\boldsymbol{h}}_i^\star := \frac{1}{K} \sum_{k=1}^{K} \boldsymbol{h}_i^{(k)\star} = \boldsymbol{0}, \ \forall\, i \in [n], \quad (9)$$

$$\boldsymbol{b}^\star = b^\star \boldsymbol{1}_K, \quad (10)$$

*where either $b^\star = 0$ or $\lambda_{\boldsymbol{b}} = 0$. The matrix $\boldsymbol{W}^\star$ forms a $K$-simplex ETF in the sense that*

$$\frac{1}{\|\boldsymbol{w}^\star\|_2^2} (\boldsymbol{W}^\star)^T \boldsymbol{W}^\star = \frac{K}{K-1} \left( \boldsymbol{I}_K - \frac{1}{K} \boldsymbol{1}_K \boldsymbol{1}_K^T \right), \quad (11)$$

*where $\boldsymbol{I}_K \in \mathbb{R}^{K \times K}$ denotes the identity matrix, and $\boldsymbol{1}_K \in \mathbb{R}^K$ denotes the all ones vector.* ∎

**Theorem 3.2.** *Assume that the feature dimension $d$ is larger than the category number $K$, i.e., $d \geq K - 1$. Then any global minimizer $(\boldsymbol{W}^\star, \boldsymbol{H}^\star, \boldsymbol{b}^\star)$ of $f_{\text{bce}}(\boldsymbol{W}, \boldsymbol{H}, \boldsymbol{b})$ defined using $\mathcal{L}_{\text{bce}}$ with Eq. (3) obeys the properties (7) - (11), where $b^\star$ is the solution of equation*

$$0 = -\frac{K-1}{K\left(1 + \exp\left(b + \sqrt{\frac{\lambda_{\boldsymbol{W}}}{n\lambda_{\boldsymbol{H}}}} \frac{\rho}{K(K-1)}\right)\right)}$$
$$+ \frac{1}{K\left(1 + \exp\left(\sqrt{\frac{\lambda_{\boldsymbol{W}}}{n\lambda_{\boldsymbol{H}}}} \frac{\rho}{K} - b\right)\right)} + \lambda_{\boldsymbol{b}} b, \quad (12)$$

*and $\rho = \|\boldsymbol{W}^\star\|_F^2$ is the squared Frobenius norm of $\boldsymbol{W}^\star$.*

**Proof** *The detailed proof is presented in the supplementary, i.e., Theorem C.3, which similar to that of Lu & Steinerberger (2022); Zhou et al. (2022b); Zhu et al. (2021), studies lower bounds for the BCE loss in Eq. (3) and finds the conditions for achieving the lower bounds.* ∎

Theorem 3.2 broadens the range of losses that can lead to NC, i.e., the contrastive property (Zhou et al., 2022b) is not necessarily satisfied.

**The decision scores.** According to Theorems 3.1 and 3.2, when CE or BCE reaches its minimum and results in NC, the sample feature $\boldsymbol{h}_i^{(k)}$ will converge to its class center $\bar{\boldsymbol{h}}^{(k)} = \frac{1}{n}\sum_{i=1}^n \boldsymbol{h}_i^{(k)}$, indicating the maximum intra-class compactness. Furthermore, the class center $\bar{\boldsymbol{h}}^{(k)}$ becomes a multiple of the corresponding classifier $\boldsymbol{w}_k$, and the $K$ classifiers $\{\boldsymbol{w}_k\}_{k=1}^K$ form an ETF, indicating the maximum inter-class distinctiveness. Furthermore, the positive and negative decision scores without the biases of all samples will converge to fixed values, i.e., for $\forall j \neq k \in [K]$,

$$s_{\text{pos}}^{(kk,i)} = \boldsymbol{w}_k^T \boldsymbol{h}_i^{(k)} \to \sqrt{\frac{\lambda_{\boldsymbol{W}}}{n\lambda_{\boldsymbol{H}}}}\frac{\rho}{K} \quad \text{and} \quad (13)$$

$$s_{\text{neg}}^{(jk,i)} = \boldsymbol{w}_j^T \boldsymbol{h}_i^{(k)} \to -\sqrt{\frac{\lambda_{\boldsymbol{W}}}{n\lambda_{\boldsymbol{H}}}}\frac{\rho}{K(K-1)}. \quad (14)$$

**The classifier bias.** Comparing Theorems 3.1 and 3.2, one can find that the primary difference between CE and BCE losses lies in their classifier biases. According to Theorem 3.1, when $\lambda_{\boldsymbol{b}} > 0$, the minimum point of CE loss satisfies $\boldsymbol{b} = \boldsymbol{0}$; when $\lambda_{\boldsymbol{b}} = 0$, any point that satisfies properties (7) - (11) and $\boldsymbol{b} = b^\star \boldsymbol{1}$ is a minimum point of CE loss, which implies that the minimum points of CE loss form a ridge line in term of $\boldsymbol{b}$. In contrast, the classifier bias $\boldsymbol{b}$ of the minimum points of BCE loss satisfy Eq. (12) whenever $\lambda_{\boldsymbol{b}} = 0$ or not. According to Lemma C.8 in the supplementary, Eq. (12) has only one solution, indicating that the BCE loss has only one minimum point in term of $\boldsymbol{b}$. This optimal classifier bias $\boldsymbol{b} = b^\star \boldsymbol{1}$ will separate the positive and negative decision scores if it satisfies the Eq. (168) (see Lemma C.9 in supplementary for details).

### 3.3. The decision scores in training with BCE and CE

Both CE and BCE can theoretically optimize the feature properties, while they perform different in practice.

**A geometric comparison for CE and BCE.** In practical training with CE or BCE, to minimize the loss, it is desirable for their exponential function variables to be as small as possible, and less than zero at least. For CE in Eq. (1), it is desirable that, for $\forall j \neq k \in [K]$,

$$\underbrace{\boldsymbol{w}_k^T \boldsymbol{h}^{(k)} - b_k}_{\text{positive decision score}} > \underbrace{\boldsymbol{w}_j^T \boldsymbol{h}^{(k)} - b_j}_{\text{negative decision score}}, \quad (15)$$

while, for BCE in Eq. (2), it is desirable that, for $\forall j \neq k$,

$$\boldsymbol{w}_k^T \boldsymbol{h}^{(k)} - b_k > 0 \quad \text{and} \quad \boldsymbol{w}_j^T \boldsymbol{h}^{(k)} - b_j < 0. \quad (16)$$

In Fig. 1, we apply the distance of vectors to reflect their inner product or similarity in the distance space. Without

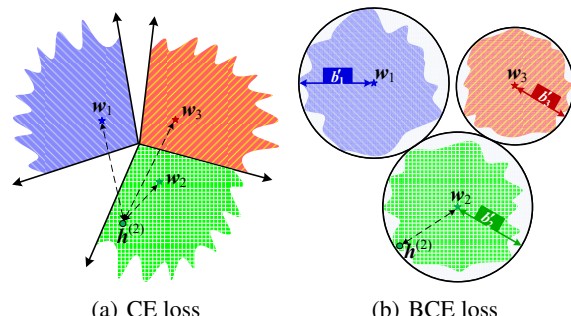

(a) CE loss         (b) BCE loss

*Figure 1.* The feature distributions of CE and BCE losses in the distance space. We respectively apply the blue, red, and green shading to indicate the feature regions of three categories. The pentagrams represent their classifiers, and the solid dot represents a general feature $\boldsymbol{h}^{(2)}$ in the second category. Since the distance between two vectors is inversely proportional to their similarity/inner product, CE loss requires the distance from the feature to its classifier vector to be less than the distance to other classifier vectors, while BCE loss requires the distance to be less than its corresponding bias. Small $b_k'$ implies large $b_k$ in Eq. (16).

considering the bias $\boldsymbol{b}$, the CE push feature $\boldsymbol{h}^{(k)}$ closer to its classifier $\boldsymbol{w}_k$ compared to others $\{\boldsymbol{w}_j\}_{j\neq k}$, implying a *unbounded* feature region for each category and unsatisfactory intra-class compactness. In addition, any two unbounded feature regions could share the same decision boundary, indicating unsatisfactory inter-class distinctiveness. In the training with CE, the bias $b_k$ acts as a compensation to adjust the distance/decision score between the features and the classifiers, introducing indirect constraint across sample features by Eq. (15). This constraint will vanish if $b_k = b_j$ for $\forall k \neq j$, which could be reached at the minima of CE. Overall, the CE only requires the positive decision score to be relatively greater than the negative ones for each sample, to implicitly enhance the features' properties by correctly classifying samples one-by-one.

In contrast, for BCE, Eq. (16) requires the feature $\boldsymbol{h}^{(k)}$ to fall within a *closed* hypersphere centered at its classifier $\boldsymbol{w}_k$ with a "radius" of $b_k$, meanwhile it requires that any two hypersphere do not intersect, indicating well intra-class compactness and inter-class distinctiveness. In other words, BCE presents explicitly constraint across-samples in the training. While Eq. (16) requires the positive decision scores of all samples are uniformly larger than threshold $t = 0$, and the all negative ones are uniformly smaller than the unified threshold, i.e.,

$$\min \bigcup_{k=1}^K \bigcup_{i=1}^n \{\boldsymbol{w}_k^T \boldsymbol{h}_i^{(k)} - b_k\} > t$$

$$\geq \max \bigcup_{k=1}^K \bigcup_{i=1}^n \{\boldsymbol{w}_j^T \boldsymbol{h}_i^{(k)} - b_j\}_{\substack{j=1 \\ j\neq k}}^K, \quad (17)$$

while the unified threshold $t$ might be not exactly zero in

practice. In the training, the classifier biases $\{b_k\}$ would be absorbed into the threshold. Then, in contrary, the final biases could reflect the intra-class compactness and the inter-class distinctiveness. Therefore, BCE can explicitly enhance the compactness and distinctiveness across sample features by learning well classifier biases.

**The decision scores in practical training.** In deep learning, gradient descent and back propagation are the most commonly used techniques for the model training. We here analyze the gradients in terms of the positive decision score $(\boldsymbol{w}_k^T \boldsymbol{h}_i^{(k)} - b_k)$ from category $k$,

$$\frac{\partial f_{\text{ce}}(\boldsymbol{W}, \boldsymbol{H}, \boldsymbol{b})}{\partial (\boldsymbol{w}_k^T \boldsymbol{h}_i^{(k)} - b_k)} = \frac{\mathrm{e}^{\boldsymbol{w}_k^T \boldsymbol{h}_i^{(k)} - b_k}}{\sum_\ell \mathrm{e}^{\boldsymbol{w}_\ell^T \boldsymbol{h}_i^{(k)} - b_\ell}} - 1, \qquad (18)$$

$$\frac{\partial f_{\text{bce}}(\boldsymbol{W}, \boldsymbol{H}, \boldsymbol{b})}{\partial (\boldsymbol{w}_k^T \boldsymbol{h}_i^{(k)} - b_k)} = \frac{1}{1 + \mathrm{e}^{-\boldsymbol{w}_k^T \boldsymbol{h}_i^{(k)} + b_k}} - 1. \qquad (19)$$

According to Eq. (18), in the training with CE, for any two samples $\boldsymbol{X}_i^{(k)}, \boldsymbol{X}_{i'}^{(k)}$ with diverse initial positive decision scores, if their predicted probabilities are equal, i.e., $\frac{\mathrm{e}^{\boldsymbol{w}_k^T \boldsymbol{h}_i^{(k)} - b_k}}{\sum_\ell \mathrm{e}^{\boldsymbol{w}_\ell^T \boldsymbol{h}_i^{(k)} - b_\ell}} = \frac{\mathrm{e}^{\boldsymbol{w}_k^T \boldsymbol{h}_{i'}^{(k)} - b_k}}{\sum_\ell \mathrm{e}^{\boldsymbol{w}_\ell^T \boldsymbol{h}_{i'}^{(k)} - b_\ell}}$, which is somewhat likely to occur during the practical training, then their positive scores will experience the same update of amplitude during back propagation. Consequently, it will be difficult to update the positive scores to the uniformly high level, impeding the enhancement of intra-class compactness within the same category.

In contrast, according to Eq. (19), during training with BCE loss, the large positive decision scores $(\boldsymbol{w}_k^T \boldsymbol{h}_i^{(k)} - b_k)$ were updated for the small amplitude $1 - \frac{1}{1 + \exp(-\boldsymbol{w}_k^T \boldsymbol{h}_i^{(k)} + b_k)}$, while the small ones were updated for the large update amplitude, facilitating a more rapid adjustment of positive scores across different samples to a uniformly high level, to enhance the intra-class compactness of sample features.

A similar phenomenon can also occur with the negative decision scores, resulting in unsatisfactory inter-class distinctiveness in the training with CE loss, while BCE could adjust them in a uniform way and push them to a uniformly low level, enhancing the inter-class distinctiveness.

**The classifier bias in practice.** During the model training, the classifier biases are also updated through the gradient descents, and the positive and negative decision scores are constrained by approaching the stable point of the biases. For CE, the gradient of bias $b_k$ is

$$\frac{\partial f_{\text{ce}}}{\partial b_k} = \frac{1}{nK} \left( n - \sum_{j=1}^{K} \sum_{i=1}^{n} \frac{\mathrm{e}^{\boldsymbol{w}_k^T \boldsymbol{h}_i^{(j)} - b_k}}{\sum_\ell \mathrm{e}^{\boldsymbol{w}_\ell^T \boldsymbol{h}_i^{(j)} - b_\ell}} \right) + \lambda_{\boldsymbol{b}} b_k$$
$$\rightarrow \lambda_{\boldsymbol{b}} b. \qquad (20)$$

As approaching the stable point of the bias, i.e., the points

satisfying $\frac{\partial f_{\text{ce}}}{\partial b_k} = 0$, Eq. (20) presents constraint on the relative value of the exponential decision scores, while the constraint will vanish as reaching the minimum of CE, and the bias gradient $\frac{\partial f_{\text{ce}}}{\partial b_k}$ approaches $\lambda_{\boldsymbol{b}} b$, according to Eq. (14). At the minimum points, the update amplitude of bias is $\eta \lambda_{\boldsymbol{b}} b$, where $\eta$ denotes the learning rate. If $\lambda_{\boldsymbol{b}} = 0$, the update is zero, and the final bias can locate at any point on the ridge line $\boldsymbol{b} = b\mathbf{1}$, where $b$ is depended on some other factors, such as the bias initial value, but not the relationship between the bias and the decision scores. If $\lambda_{\boldsymbol{b}} > 0$, one can concluded $b = 0$; however, in practice, this theoretical value might be not reached due to that $\eta \lambda_{\boldsymbol{b}}$ will be very small at the terminal phase of practical training. The above analysis implies that the classifier biases of CE cannot provide consistent and explicit constraints on the decision scores in the practical training, and thus do not substantively affect the final features' properties.

In contrast, for BCE, the gradient of bias $b_k$ is

$$\frac{\partial f_{\text{bce}}}{\partial b_k} = \frac{1}{nK} \left( n - \sum_{j=1}^{K} \sum_{i=1}^{n} \frac{1}{1 + \mathrm{e}^{-\boldsymbol{w}_k^T \boldsymbol{h}_i^{(j)} + b_k}} \right) + \lambda_{\boldsymbol{b}} b_k$$
$$\rightarrow \text{RHS of Eq. (12)}, \qquad (21)$$

which presents clear constraint on the absolute value of the exponential decision scores for the all samples. The constraint evolve into Eq. (12) when BCE reaches its minimum points. Therefore, as approaching the stable point, the classifier bias consistently and explicitly constrain the decision scores, regardless $\lambda_{\boldsymbol{b}} = 0$ or not, and it will separate the final positive and negative decision scores if Eq. (168) holds. In other words, the classifier biases in BCE play a substantial role in enhancing the final features' properties.

## 4. Experiments

### 4.1. Comparison of BCE and CE in NC

To compare CE and BCE in deep feature learning, we train deep classification models, ResNet18, ResNet50 (He et al., 2016), and DenseNet121 (Huang et al., 2017), on three popular datasets, including MNIST (LeCun et al., 1998), CIFAR10, and CIFAR100 (Krizhevsky et al., 2009). We train the models using SGD and AdamW for 100 epochs with batch size of 128. The initial learning rate is set to 0.01 and 0.001 for SGD and AdamW, which is respectively decayed in "step" and "cosine" schedulers.

**NC across models and datasets** Similar to the works of Zhu et al. (2021) and Zhou et al. (2022b), we do not apply any data augmentation in the experiments of NC, and adopt the metrics, $\mathcal{NC}_1, \mathcal{NC}_2$, and $\mathcal{NC}_3$ (see supplementary for their definitions), to measure the properties of NC1, NC2, and NC3 resulted by CE and BCE. The lower metrics reflect the better NC properties.

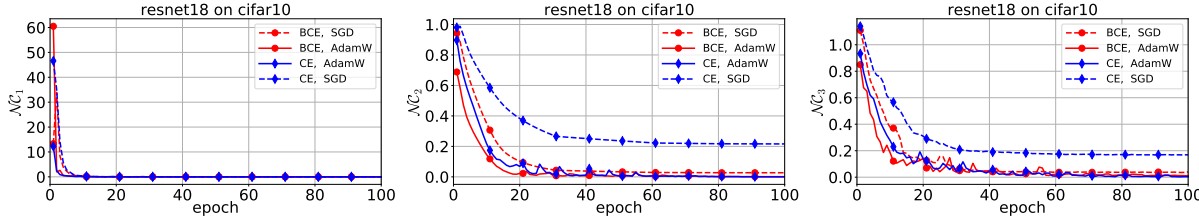

*Figure 2.* NC metrics of ResNet18 trained on CIFAR10 with CE and BCE using SGD and AdamW, respectively. The NC metrics approach zero at the terminal phase of training, while the NC metrics of BCE decrease faster than that of CE in the first 20 epochs.

In the training, we set $\lambda_W = \lambda_H = \lambda_b = 5 \times 10^{-4}$, and no weight decay is applied on the other parameters of model $\mathcal{M}$. Fig. 2 shows the NC results of ResNet18 trained by CE and BCE with two optimizers on CIFAR10, and the other results are presented in the supplementary. From the figure, one can find that all the three NC metrics consistently approach zero in the training with different losses and optimizers, which matches Theorem 3.1 and 3.2. Meanwhile, in the initial training stage of the first 20 epochs, the NC metrics (the red curves with dots) of BCE usually decrease faster than that (the blue curves with diamonds) of CE, implying that BCE is easier to result in NC. More numerical results are presented in supplementary.

**The bias decay factor $\lambda_b$.** To illustrate the different effects of classifier biases of CE and BCE on the decision scores, we conduct two groups of experiments by respectively applying fixed and varying classifier bias decay factor $\lambda_b$ in the training of ResNet18 on MNIST: (1) with fixed $\lambda_b = 0$ and default other hyper-parameters, setting the mean of the initialized classifier biases to $0, 1, 2, 3, 4, 5, 6, 8$, and $10$, respectively; (2) with varying $\lambda_b = 0.5, 0.05, 5 \times 10^{-3}, 5 \times 10^{-4}, 5 \times 10^{-5}$, and $5 \times 10^{-6}$, respectively, setting the mean of initialized classifier biases to $10$.

Fig. 3 shows the distributions of final classifier biases and positive/negative decision scores (without bias) using violin plots for the 60 trained models in these two groups of experiments. One can find from Fig. 3(top), for the CE-trained models with $\lambda_b = 0$, the final classifier bias values are almost entirely determined by their initial values, no matter which optimizer was applied. For the CE-trained models in Fig. 3(bottom), the means of the final classifier biases reach to zero from the initial mean of $10$ with appropriate lager $\lambda_b$ ($\geq 5 \times 10^{-3}$ for SGD and $\geq 5 \times 10^{-4}$ for AdamW), and they do not achieve the theoretical value when $\lambda_b$ is too small. As a comparison, for these CE-trained models, their final positive and negative decision scores respectively converge to around $5.64$ and $-0.63$ (see supplementary for details). In total, in CE-trained models, the classifier biases hardly affect the decision scores, and thus almost does not affect the final feature properties.

In contrast, for the BCE-trained models in Fig. 3, their final positive and negative decision scores are always separated by the final classifier biases, no matter what the initial mean

*Table 1.* The classification on the test set of CIFAR10 and CIFAR100. The accuracy ($\mathcal{A}$) of most BCE-trained models is higher than that of CE-trained ones, while BCE-trained models perform consistently and significantly better than CE-trained models in terms of uniform accuracy ($\mathcal{A}_{\text{Uni}}$).

| $\mathcal{D}$ | $\mathcal{M}$ | Loss | SGD | | | | AdamW | | | |
|---|---|---|---|---|---|---|---|---|---|---|
| | | | DA1 | | DA1+DA2 | | DA1 | | DA1+DA2 | |
| | | | $\mathcal{A}$ | $\mathcal{A}_{\text{Uni}}$ | $\mathcal{A}$ | $\mathcal{A}_{\text{Uni}}$ | $\mathcal{A}$ | $\mathcal{A}_{\text{Uni}}$ | $\mathcal{A}$ | $\mathcal{A}_{\text{Uni}}$ |
| CIFAR10 | R18 | CE | 92.8 | 85.2 | 92.7 | 89.1 | 93.4 | 89.0 | **95.7** | 94.3 |
| | | BCE | **93.2 91.9** | | **93.6 91.9** | | **93.9 93.4** | | 95.6 | **95.2** |
| | R50 | CE | 92.7 | 85.2 | 92.7 | 89.6 | **94.5** | 87.9 | 96.0 | 94.3 |
| | | BCE | **93.4 92.5** | | **93.2 91.5** | | 94.0 | **93.6** | **96.2 95.7** | |
| | D121 | CE | 87.9 | 78.8 | 86.7 | 81.5 | 90.4 | 83.6 | 92.6 | 90.7 |
| | | BCE | **88.7 87.6** | | **87.8 85.0** | | **90.6 89.9** | | 92.6 | **91.8** |
| CIFAR100 | R18 | CE | 71.2 | 43.2 | 71.8 | 56.7 | 71.7 | 49.2 | 76.5 | 64.4 |
| | | BCE | **72.2 63.3** | | **72.3 62.9** | | **73.2 66.3** | | **76.7 70.0** | |
| | R50 | CE | 71.6 | 44.2 | 70.3 | 55.2 | 75.0 | 48.8 | **78.6** | 67.8 |
| | | BCE | **71.8 64.1** | | **71.9 62.8** | | **75.3 68.8** | | 78.5 | **72.7** |
| | D121 | CE | 60.8 | 32.9 | 57.2 | 39.8 | **63.7** | 38.8 | 69.0 | 57.2 |
| | | BCE | **61.1 53.5** | | **58.4 47.7** | | 63.6 | **57.3** | **69.4 63.5** | |

of classifier biases, $\lambda_b$, or optimizer are, and clear correlation exists between the biases and positive/negative decision scores. These results imply that, in the training with BCE, the classifier biases have a substantial impact on the sample feature distribution, thereby enhancing the compactness and distinctiveness across samples.

## 4.2. BCE outperforming CE in practice

To further demonstrate the advantages of BCE over CE, we trained classification models using these two losses on CIFAR10, CIFAR100, and ImageNet.

**Results on CIFAR10 and CIFAR100.** On the CIFARs, we train ResNet18, ResNet50, and DeseNet121 by applying two different data augmentation techniques, (1) DA1: random cropping and horizontal flipping, (2) DA2: Mixup and CutMix, on CIFAR10 and CIFAR100 using SGD and AdamW, respectively. In the experiments, we take a global weight decay factor $\lambda$ for the all parameters in the models, including the classifiers and biases, and $\lambda = 5 \times 10^{-4}$ for SGD, $\lambda = 0.05$ for AdamW. The other hyper-parameters are presented in the supplementary. To compare the results, besides the classification accuracy ($\mathcal{A}$), we define and apply three other metrics, uniform accuracy ($\mathcal{A}_{\text{Uni}}$), compactness ($\mathcal{E}_{\text{com}}$), and distinctiveness ($\mathcal{E}_{\text{dis}}$), seeing Eqs. (45,49,50) in supplementary for the definitions. While $\mathcal{A}_{\text{Uni}}$ is evolved

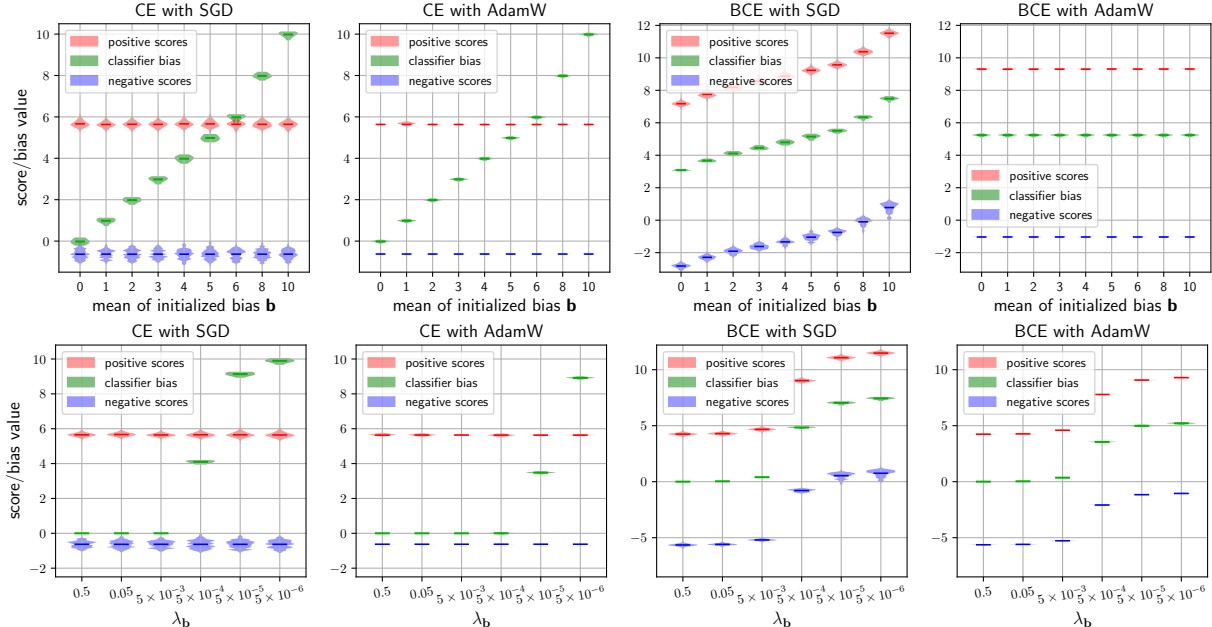

*Figure 3.* The distributions of the final classifier bias and positive/negative decision scores for 60 ResNet18s trained on MNIST with fixed weight decay factor $\lambda_b$ (top) and varying $\lambda_b$ (bottom), while $\lambda_W = \lambda_H = 5 \times 10^{-4}$. The mean $\bar{b} = \frac{1}{K} \sum_{k=1}^{K} b_k$ of initialized biases is respectively set as $0, 1, 2, 3, 4, 5, 6, 8, 10$ in the experiments with fixed $\lambda_b = 0$, and the bias mean is set as 10 in that with varying $\lambda_b$.

*Table 2.* The feature properties on test set of CIFAR10 and CI-FAR100. The compactness ($\mathcal{E}_{\text{com}}$, %) and distinctiveness ($\mathcal{E}_{\text{dis}}$, %) of BCE-trained models are usually better than that of CE-trained models. See supplementary for the definitions of $\mathcal{E}_{\text{com}}$ and $\mathcal{E}_{\text{dis}}$.

| $\mathcal{D}$ | $\mathcal{M}$ | Loss | SGD | | | | AdamW | | | |
| | | | DA1 | | DA1+DA2 | | DA1 | | DA1+DA2 | |
| | | | $\mathcal{E}_{\text{com}}$ | $\mathcal{E}_{\text{dis}}$ | $\mathcal{E}_{\text{com}}$ | $\mathcal{E}_{\text{dis}}$ | $\mathcal{E}_{\text{com}}$ | $\mathcal{E}_{\text{dis}}$ | $\mathcal{E}_{\text{com}}$ | $\mathcal{E}_{\text{dis}}$ |
| CIFAR10 | R18 | CE | 85.4 | 25.5 | 81.5 | 20.9 | 85.5 | 26.9 | 89.3 | 33.1 |
| | | BCE | **90.6** | **30.5** | **84.4** | **23.9** | **91.4** | **32.5** | **91.8** | **36.7** |
| | R50 | CE | 83.5 | 17.8 | 85.6 | 20.3 | 85.5 | 23.3 | **95.3** | **37.8** |
| | | BCE | **89.9** | **23.2** | **86.9** | **21.6** | **89.1** | **27.2** | 91.7 | 35.7 |
| | D121 | CE | 78.7 | 31.2 | 76.7 | 28.1 | 74.6 | 30.7 | 82.0 | **31.9** |
| | | BCE | **84.6** | **33.2** | **80.9** | **29.7** | **83.0** | **33.7** | **83.7** | 31.9 |
| CIFAR100 | R18 | CE | 72.3 | **27.0** | 71.3 | 25.8 | 69.2 | 29.0 | 71.4 | **30.7** |
| | | BCE | **73.3** | 26.2 | **72.9** | **26.9** | **72.7** | **29.3** | **74.2** | 29.1 |
| | R50 | CE | 70.8 | 20.0 | 71.0 | 18.7 | 68.9 | 25.8 | 72.3 | **36.5** |
| | | BCE | **73.3** | **22.0** | **74.0** | **21.8** | **75.2** | **27.8** | **76.3** | 32.5 |
| | D121 | CE | 71.2 | **31.0** | 72.8 | **31.7** | 64.7 | 29.8 | 70.0 | **34.0** |
| | | BCE | **73.2** | 29.5 | **73.6** | 30.5 | **70.9** | **30.1** | **72.6** | 32.6 |

from Eq. (17), it is calculated on the decision scores across samples, simultaneously reflecting the feature compactness and distinctiveness; as their name implies, $\mathcal{E}_{\text{com}}$ and $\mathcal{E}_{\text{dis}}$ respectively measure the intra-class compactness and inter-class distinctiveness among sample features.

Table 1 shows the classification results of the three models ("R18", "R50", and "D121" respectively stand for ResNet18, ResNet50, and DenseNet121) on the test set of CIFAR10 and CIFAR100. From the table, one can find that, BCE is better than CE in term of accuracy ($\mathcal{A}$) in most cases, and in term of uniform accuracy ($\mathcal{A}_{\text{Uni}}$), it performs consistently and significantly superior to CE. Taking CIFAR10 for example, among the twelve pairs of models trained by CE and

BCE, BCE slightly reduced the accuracy of only two pairs of models, while the gain of uniform accuracy introduced by BCE is 0.82% at least for the all models. For CIFAR100, the gain of BCE in uniform accuracy could be more than 20%, and the classification accuracy of BCE is still higher than that of CE in most cases. These results illustrate that BCE can usually achieve better classification results than CE, which is likely resulted from its enhancement in compactness and distinctiveness among sample features.

Furthermore, similar to BCE, the better data augmentation techniques and optimizer can simultaneously improve the classification results of models. For example, Mixup, Cut-Mix, and AdamW increase $\mathcal{A}$ and $\mathcal{A}_{\text{Uni}}$ from 92.82% and 85.20% to 95.72% and 94.34%, respectively, for ResNet18 trained on CIFAR10. In addition, the higher performance of BCE than CE with only DA1 implies that the superiority of BCE is not resulted from the alignment with Mixup and CutMix, which is not consistent with the statements about BCE by Wightman et al. (2021).

As the uniform accuracy simultaneously reflect the intra-class compactness and inter-class distinctiveness, the higher uniform accuracy $\mathcal{A}_{\text{Uni}}$ of BCE-trained models implies their better feature properties. Table 2 presents the compactness ($\mathcal{E}_{\text{com}}$) and distinctiveness ($\mathcal{E}_{\text{dis}}$) of the trained models. One can clearly observe that, in most cases, BCE improves the compactness and distinctiveness of the sample features extracted by the models compared to CE, which is consistent with our expectations and provides a solid and reasonable explanation for the higher performance of BCE in tasks that require feature comparison, such as facial recognition and

Table 3. The results of CE and BCE on ImageNet-1k.

| | ResNet50 | | ResNet101 | | DenseNet161 | |
|---|---|---|---|---|---|---|
| | $\mathcal{A}$ | $\mathcal{A}_{\text{Uni}}$ | $\mathcal{A}$ | $\mathcal{A}_{\text{Uni}}$ | $\mathcal{A}$ | $\mathcal{A}_{\text{Uni}}$ |
| CE | 76.74 | 34.48 | 78.47 | 38.85 | 78.58 | 43.56 |
| BCE | **77.12** | **66.92** | **78.88** | **70.46** | **79.19** | **69.08** |

verification (Wen et al., 2022; Zhou et al., 2023).

**Results on ImageNet**. We trained ResNet50, ResNet101, and DenseNet161 on ImageNet-1k by using CE and BCE, respectively. As the classification on this dataset is a complicated task, we did not train these three models from scratch for saving the training time. Instead, we applied the two losses to fine-tune the models that have been pretrained for 90 epochs, and each fine-tuning runs 30 epochs using AdamW optimizer. From the table, one can find that on the medium-scale ImageNet-1k, BCE still shows a consistent and clear advantage over CE.

**Results on long-tailed datasets**.

Though we theoretically analyzed the advantages of BCE over CE on only balanced dataset, BCE also has potential on the imbalanced datasets. On the

Table 4. The performance of CE and BCE on CIFAR100-LT with three different IF.

| IF | 10 | 50 | 100 |
|---|---|---|---|
| CE | 70.91 | 57.59 | 51.48 |
| BCE | **71.54** | **58.49** | **52.88** |

long-tailed dataset CIFAR100-LT with imbalanced factor (IF) of 10, 50, and 100, we trained ResNet32 using CE and BCE, respectively, following the protocol in (Alshammari et al., 2022). Table 4 shows the recognition results on balanced test set. One can observe that BCE consistently achieves better results on the imbalanced datasets than CE.

## 5. Discussion

$K > d$. When the category number $K$ is **greater** than the feature dimension $d$, analyzing the neural collapse with CE or BCE becomes quite challenging. Lu & Steinerberger (2022) proved that when $K$ approaches infinity and CE reaches its minimum, the features and classifiers weakly converge to a uniform distribution on the hypersphere. Liu et al. (2023) and Jiang et al. (2024) propose generalized neural collapse to analyze CE with $K > d$. We speculate that BCE will exhibit similar properties and results in this context, and its classifier biases will still substantially constrain the positive and negative decision scores during the training, thereby accelerating the convergence. Both Wen et al. (2022) and Zhou et al. (2023) achieved better face recognition results using BCE, where the category number $K$ is much greater than the feature dimension $d$.

**NC with imbalanced data.** On imbalanced datasets, Fang et al. (2021) found that when the imbalance ratio exceeds a threshold, the classifiers for the tail classes trained with CE will collapse to a single vector, while the results in Table 4 lead us believe the BCE can amplify this threshold.

**Transformer and other deep architectures.** Though we validate the conclusions of this paper using the classic convolutional neural networks in the experiments, the advantages of BCE, as indicated by our theoretical analysis, can also be demonstrated in other deep network models such as Transformers. DeiT III (Touvron et al., 2022) has already shown the potential of BCE with Transformer on ImageNet classification, and in addition, LiVT (Xu et al., 2023) is another Transformer model trained with BCE that has achieved excellent results in long-tailed recognition (LTR) tasks.

**The losses for binary classification.** When $K = 2$, we denotes the two classes as 0 and 1. For $\forall \boldsymbol{h}^{(k)}$ with $k = 0$ or 1, the CE in Eq. (1) requires only one decision score $z^{(k)} = (\boldsymbol{w}_0 - \boldsymbol{w}_1)^T \boldsymbol{h}^{(k)} - (b_0 - b_1) = \boldsymbol{w}^T \boldsymbol{h}^{(k)} - b$ to classify the sample, and the CE degenerates the naive BCE,

$$\mathcal{L}_{\text{n-bce}}(z^{(k)}) = \begin{cases} \log\left(1 + \exp(z^{(k)})\right), & k = 0, \\ \log\left(1 + \exp(-z^{(k)})\right), & k = 1, \end{cases} \quad (22)$$

which is also referred to Sigmoid loss in previous literatures. In contrast, when $K = 2$, for $\forall \boldsymbol{h}^{(k)}$, the BCE in Eq. (2) is

$$\mathcal{L}_{\text{bce}}(\boldsymbol{z}^{(k)}) = \log\left(1 + e^{-z_k^{(k)}}\right) + \log\left(1 + e^{z_{1-k}^{(k)}}\right), \quad (23)$$

where $z_j^{(k)} = \boldsymbol{w}_j^T \boldsymbol{h}^{(k)} - b_j, j, k \in \{0, 1\}$. In total, when $K = 2$, the naive BCE uses only one Sigmoid to measure the relative values of the exponential positive and negative decision scores, while the BCE analyzed in this paper uses two Sigmoid to respectively measure the absolute values of the exponential positive and negative scores.

## 6. Conclusions

This paper compares CE and BCE losses in deep feature learning. Both the losses can maximize the intra-class compactness and inter-class distinctiveness among sample features, i.e., leading to neural collapse when reaching their minima. In the training, CE implicitly enhances the feature properties by correctly classifying samples one-by-one. In contrast, BCE can adjust the positive and negative decision scores across samples, and, in this process, the classifier biases play a substantial and consistent role, making it explicitly enhance the intra-class compactness and inter-class distinctiveness of features. Therefore, BCE has potential to achieve better classification performance.

## Acknowledgements

The research was supported by National Natural Science Foundation of China under Grant 8226113862, Guangdong Provincial Key Laboratory under Grant 2023B1212060076, and Scientific Foundation for Youth Scholars of Shenzhen University under Grant 868-000001032180.

## Impact Statement

This paper presents work whose goal is to advance the field of Machine Learning. There are many potential societal consequences of our work, none which we feel must be specifically highlighted here.

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

# BCE vs. CE in Deep Feature Learning

## Supplementary Material

## A. Neural collapse and feature property

### A.1. Neural collapse

The neural collapse was first found by Papyan et al. (2020), which refers to four properties about the sample features $\{\boldsymbol{h}_i^{(k)}\}$ and the classifier vectors $\{\boldsymbol{w}_k\}$ at the terminal phase of training (Han et al., 2022), as list in Sec. 2.2. These four properties can be formulized as follows.

- **NC1**, within-class variability collapse, $\boldsymbol{\Sigma}_B^{\dagger}\boldsymbol{\Sigma}_W \to \boldsymbol{0}$, where

$$\boldsymbol{\Sigma}_B = \frac{1}{K}\sum_{k=1}^{K}\big(\bar{\boldsymbol{h}}^{(k)} - \bar{\boldsymbol{h}}\big)\big(\bar{\boldsymbol{h}}^{(k)} - \bar{\boldsymbol{h}}\big)^T \tag{24}$$

$$\boldsymbol{\Sigma}_W = \frac{1}{\sum_k n_k}\sum_{k=1}^{K}\sum_{i=1}^{n_k}\big(\boldsymbol{h}_i^{(k)} - \bar{\boldsymbol{h}}^{(k)}\big)\big(\boldsymbol{h}_i^{(k)} - \bar{\boldsymbol{h}}^{(k)}\big)^T \tag{25}$$

$$\bar{\boldsymbol{h}}^{(k)} = \frac{1}{n_k}\sum_{i=1}^{n_k}\boldsymbol{h}_i^{(k)}, \tag{26}$$

$$\bar{\boldsymbol{h}} = \frac{1}{\sum_k n_k}\sum_{k=1}^{K}\sum_{i=1}^{n_k}\boldsymbol{h}_i^{(k)} \tag{27}$$

  and $\dagger$ denotes the Mooer-Penrose pseudo-inverse;

- **NC2**, convergence to simplex equiangular tight frame,

$$\big\|\bar{\boldsymbol{h}}^{(k)} - \bar{\boldsymbol{h}}\big\|_2 - \big\|\bar{\boldsymbol{h}}^{(k')} - \bar{\boldsymbol{h}}\big\|_2 \to 0, \tag{28}$$

$$\frac{\big\langle\bar{\boldsymbol{h}}^{(k)} - \bar{\boldsymbol{h}},\, \bar{\boldsymbol{h}}^{(k')} - \bar{\boldsymbol{h}}\big\rangle}{\big\|\bar{\boldsymbol{h}}^{(k)} - \bar{\boldsymbol{h}}\big\|_2\big\|\bar{\boldsymbol{h}}^{(k')} - \bar{\boldsymbol{h}}\big\|_2} \to \begin{cases} 1, & k = k', \\ -\frac{1}{K-1}, & k \neq k'; \end{cases} \tag{29}$$

- **NC3**, convergence to self-duality,

$$\frac{\boldsymbol{w}_k}{\big\|\boldsymbol{w}_k\big\|_2} - \frac{\bar{\boldsymbol{h}}^{(k)} - \bar{\boldsymbol{h}}}{\big\|\bar{\boldsymbol{h}}^{(k)} - \bar{\boldsymbol{h}}\big\|_2} \to 0; \tag{30}$$

- **NC4**, simplification to nearest class center,

$$\arg\max_{j}\big\{\boldsymbol{w}_j\boldsymbol{h} - b_j\big\}_{j=1}^{K} \to \arg\min_{j}\big\{\|\boldsymbol{h} - \bar{\boldsymbol{h}}^{(j)}\|_2\big\}_{j=1}^{K}. \tag{31}$$

In Sec. 4, we applied three metrics, $\mathcal{NC}_1, \mathcal{NC}_2, \mathcal{NC}_3$, to measure the above properties, similar to that defined in (Zhou et al., 2022b; Zhu et al., 2021):

$$\mathcal{NC}_1 := \frac{1}{K}\text{trace}\big(\boldsymbol{\Sigma}_W\boldsymbol{\Sigma}_B^{\dagger}\big), \tag{32}$$

$$\mathcal{NC}_2 := \Big\|\frac{\tilde{\boldsymbol{W}}\tilde{\boldsymbol{W}}^T}{\|\tilde{\boldsymbol{W}}\tilde{\boldsymbol{W}}^T\|_F} - \frac{1}{\sqrt{K-1}}\big(\boldsymbol{I}_K - \frac{1}{K}\boldsymbol{1}_K\boldsymbol{1}_K^T\big)\Big\|_F, \tag{33}$$

$$\mathcal{NC}_3 := \Big\|\frac{\boldsymbol{W}\tilde{\boldsymbol{H}}}{\|\boldsymbol{W}\tilde{\boldsymbol{H}}\|_F} - \frac{1}{\sqrt{K-1}}\big(\boldsymbol{I}_K - \frac{1}{K}\boldsymbol{1}_K\boldsymbol{1}_K^T\big)\Big\|_F, \tag{34}$$

where

$$\tilde{W} = [w_1 - \bar{w}, w_2 - \bar{w}, \cdots, w_K - \bar{w}]^T \in \mathbb{R}^{K \times d}, \tag{35}$$

$$\tilde{H} = [\bar{h}^{(1)} - \bar{h}, \bar{h}^{(2)} - \bar{h}, \cdots, \bar{h}^{(K)} - \bar{h}] \in \mathbb{R}^{d \times K}, \tag{36}$$

$$\bar{w} = \frac{1}{K} \sum_{k=1}^{K} w_k. \tag{37}$$

When defining $\mathcal{NC}_2$, Zhu et al. (2021) and Zhou et al. (2022b) did not subtract the classifier vectors with their mean, i.e., the original $\mathcal{NC}_2$ is defined as $\left\| \frac{WW^T}{\|WW^T\|_F} - \frac{1}{\sqrt{K-1}} \left( I_K - \frac{1}{K} 1_K 1_K^T \right) \right\|_F$, with $W = [w_1, w_2, \cdots, w_K]^T \in \mathbb{R}^{K \times d}$.

As mentioned by Zhu et al. (2021) and Zhou et al. (2022b), due to the "ReLU" activation functions before the FC classifiers in the deep models, the feature mean $\tilde{h}_i = \frac{1}{K} \sum_{k=1}^{K} h_i^{(k)}$ will be non-negative, which conflicts with $\tilde{h}_i = 0$ required by Theorems 3.1 and 3.2. Then, the average features/class centers of $K$ categories do not directly form an ETF, while the globally-centered average features form ETF, i.e., NC2 properties described by Eqs. (28) and (29). As the proof of Theorems 3.1 and 3.2, in the neural collapse, the features of each category will be parallel to its classifier vector, i.e., $h_i^{(k)} = \sqrt{\frac{\lambda_W}{n\lambda_H}} w_k$ in Eqs (129,130). Therefore, the classifier vectors $\{w_k\}$ should also subtract their global mean before form an ETF. In other words, the third NC property should be

$$\textbf{NC3'}: \quad \frac{w_k - \bar{w}}{\|w_k - \bar{w}\|_2} - \frac{\bar{h}^{(k)} - \bar{h}}{\|\bar{h}^{(k)} - \bar{h}\|_2} \to 0. \tag{38}$$

As our analysis, when a model falling to the neural collapse, its classification accuracy $\mathcal{A}$ and uniform accuracy $\mathcal{A}_{\text{Uni}}$ must be 100% on the training dataset.

## A.2. Feature property

In the experiments, we applied four metrics to compare the performance of CE and BCE, i.e., classification accuracy $\mathcal{A}$, uniform accuracy $\mathcal{A}_{\text{Uni}}$, feature compactness $\mathcal{E}_{\text{com}}$, and distinctiveness $\mathcal{E}_{\text{dis}}$. These metrics on the training data will be maximized when the model, classifier, and loss in the neural collapse.

In a classification task, suppose a dataset $\mathcal{D} = \bigcup_{k=1}^{K} \mathcal{D}_k = \bigcup_{k=1}^{K} \bigcup_{i=1}^{n_k} \{X_i^{(k)}\}$ from $K$ categories, where $X_i^{(k)}$ denotes the $i$th sample from the category $k$. For the sample $X_i^{(k)}$ in $\mathcal{D}$, a model $\mathcal{M}$ converts it into its feature $h_i^{(k)} = \mathcal{M}(X_i^{(k)}) \in \mathbb{R}^d$, where $d$ is the length of the feature vector. A linear, full connection (FC) classifier $\mathcal{C} = \{(w_k, b_k)\}_{k=1}^{K}$ transform the feature into $K$ decision scores $\{w_j h_i^{(k)} - b_j\}_{j=1}^{K}$. Then, the sample can be correctly classified if

$$w_k h_i^{(k)} - b_k = \max \{w_j h_i^{(k)} - b_j\}_{j=1}^{K}, \tag{39}$$

which is equivalent to

$$k = \arg\max_{\ell} \{w_\ell^T h^{(k)} - b_\ell\}. \tag{40}$$

The the commonly used **classification accuracy** can be defined as

$$\mathcal{A}(\mathcal{M}, \mathcal{C}) = \frac{|\mathcal{D}(\mathcal{M}, \mathcal{C})|}{|\mathcal{D}|} \times 100\%, \tag{41}$$

where

$$\mathcal{D}(\mathcal{M}, \mathcal{C}) = \bigcup_{k=1}^{K} \left\{ X^{(k)} : k = \arg\max_{\ell} \{w_\ell^T h^{(k)} - b_\ell\}, X^{(k)} \in \mathcal{D}_k, h^{(k)} = \mathcal{M}(X^{(k)}) \right\}, \tag{42}$$

consisting of the all samples correctly classified by $\mathcal{M}$ and $\mathcal{C}$ in $\mathcal{D}$.

Eq. (39) implies a dynamic threshold $t_{\boldsymbol{X}}$ separating the positive and negative decision scores. Inspired by Eq. (17), we define uniform accuracy by using a unified threshold. Firstly, for given dataset $\mathcal{D}$ and model $\mathcal{M}$, classifier $\mathcal{C}$ with a fixed threshold $t$, we denote a subset of $\mathcal{D}$ as

$$\mathcal{D}(\mathcal{M},\mathcal{C};t) = \bigcup_{k=1}^{K} \left\{ \boldsymbol{X}^{(k)} \in \mathcal{D}_k : \boldsymbol{w}_k \boldsymbol{h}^{(k)} - b_k > t \geq \max\left\{ \boldsymbol{w}_j^T \boldsymbol{h}^{(k)} - b_j \right\}_{\substack{j=1 \\ j \neq k}}^{K}, \boldsymbol{h}^{(k)} = \mathcal{M}(\boldsymbol{X}^{(k)}) \right\} \quad (43)$$

which is the biggest subset of $\mathcal{D}$ uniformly classified by $\mathcal{M}$ and $\mathcal{C}$ with $t$. Then the ratio

$$\mathcal{A}_{\mathrm{Uni}}(\mathcal{M},\mathcal{C};t) = \frac{|\mathcal{D}(\mathcal{M},\mathcal{C};t)|}{|\mathcal{D}|} \times 100\%, \quad (44)$$

is the corresponding uniform accuracy, and the maximum ratio with varying thresholds, i.e.,

$$\mathcal{A}_{\mathrm{Uni}}(\mathcal{M},\mathcal{C}) = \max_{t \in \mathbb{R}} \mathcal{A}_{\mathrm{Uni}}(\mathcal{M},\mathcal{C};t), \quad (45)$$

is defined as the final **uniform accuracy**.

In practice, to calculate the uniform accuracy $\mathcal{A}_{\mathrm{Uni}}$, the sets of positive and negative decision scores for the all samples

$$\mathcal{S}_{\mathrm{pos}} = \bigcup_{k=1}^{K} \left\{ \boldsymbol{w}_k \boldsymbol{h}_i^{(k)} - b_k : i = 1, 2, \cdots, n_k \right\}, \quad (46)$$

$$\mathcal{S}_{\mathrm{neg}} = \bigcup_{k=1}^{K} \bigcup_{\substack{j=1 \\ j \neq k}}^{K} \left\{ \boldsymbol{w}_j \boldsymbol{h}_i^{(k)} - b_j : i = 1, 2, \cdots, n_k \right\} \quad (47)$$

are first computed, and denote

$$s_{\mathrm{pos\text{-}min}} = \min(\mathcal{S}_{\mathrm{pos}}) \qquad \text{and} \qquad s_{\mathrm{neg\text{-}max}} = \max(\mathcal{S}_{\mathrm{neg}}). \quad (48)$$

If $s_{\mathrm{pos\text{-}min}} \geq s_{\mathrm{neg\text{-}max}}$, the classification accuracy $\mathcal{A}$ and the uniform one $\mathcal{A}_{\mathrm{Uni}}$ must be $100\%$, otherwise, $N = 200$ thresholds $\{t_i\}_{i=1}^{N}$ are evenly taken from the interval $[s_{\mathrm{pos\text{-}min}}, s_{\mathrm{neg\text{-}max}}]$, and $N = 200$ uniform accuracy $\mathcal{A}_{\mathrm{Uni}}(\mathcal{M},\mathcal{C};t_i)$ are figured out, while the best one $\max\left\{ A_{\mathrm{Uni}}(\mathcal{M},\mathcal{C};t_i) \right\}_{i=1}^{N}$ is chosen as the final uniform accuracy $\mathcal{A}_{\mathrm{Uni}}$. In this calculation, the final results will be slightly different when different numbers ($N$) of thresholds are taken in the score interval.

By Eqs. (17), a model with higher uniform accuracy, it would lead to more samples from category $k, \forall k \in [K]$, whose inner products (positive similarities/decision scores without bias) with the classifier vector $\boldsymbol{w}_k$ are greater than $b_k + t$, implying higher intra-class compactness in each category, and requires more samples whose inner products (negative similarities/decision scores without bias) with the classifier vectors of other categories are less than $b_j + t$, implying higher inter-class distinctiveness among all categories. For the intra-class **compactness** $\mathcal{E}_{\mathrm{com}}$ and inter-class **distinctiveness** $\mathcal{E}_{\mathrm{dis}}$ among sample features, we define them as

$$\mathcal{E}_{\mathrm{com}} = \frac{1}{2}\left[ \frac{1}{K} \sum_{k=1}^{K} \left( \frac{1}{n_k^2} \sum_{i=1}^{n_k} \sum_{i'=1}^{n_k} \frac{\langle \boldsymbol{h}_i^{(k)} - \bar{\boldsymbol{h}}, \boldsymbol{h}_{i'}^{(k)} - \bar{\boldsymbol{h}} \rangle}{\|\boldsymbol{h}_i^{(k)} - \bar{\boldsymbol{h}}\|\|\boldsymbol{h}_{i'}^{(k)} - \bar{\boldsymbol{h}}\|} \right) + 1 \right] \times 100\%, \quad (49)$$

$$\mathcal{E}_{\mathrm{dis}} = \frac{1}{2}\left[ 1 - \frac{1}{K(K-1)} \sum_{k=1}^{K} \sum_{\substack{k'=1 \\ k' \neq k}}^{K} \left( \frac{1}{n_k} \frac{1}{n_{k'}} \sum_{i=1}^{n_k} \sum_{i'=1}^{n_{k'}} \frac{\langle \boldsymbol{h}_i^{(k)}, \boldsymbol{h}_{i'}^{(k')} \rangle}{\|\boldsymbol{h}_i^{(k)}\|\|\boldsymbol{h}_{i'}^{(k')}\|} \right) \right] \times 100\%, \quad (50)$$

where $\bar{\boldsymbol{h}} = \frac{1}{|\mathcal{D}|} \sum_{k=1}^{K} \sum_{i=1}^{n_k} \boldsymbol{h}_i^{(k)}$ is the global feature center.

Due to the neural collapse, the compactness $\mathcal{E}_{\mathrm{com}}$ might be higher than $\frac{1}{2} - \frac{1}{2(K-1)}$, and the distinctiveness $\mathcal{E}_{\mathrm{dis}}$ might be lower than $\frac{1}{2} + \frac{1}{2(K-1)}$, for the model $\mathcal{M}$ and classifier $\mathcal{C}$ which have been well trained on the dataset $\mathcal{D}$.

# B. Experimental settings and results

## B.1. Experimental settings

*Table 5.* Experimental settings in our experiments.

| | | Neural collapse | | Classification | | | |
| --- | --- | --- | --- | --- | --- | --- | --- |
| | | setting-1 | setting-2 | setting-3 | setting-4 | setting-5 | setting-6 |
| Hyper-parameter | epochs | 100 | 100 | 100 | 100 | 100 | 100 |
| | optimizer | SGD | AdamW | SGD | AdamW | SGD | AdamW |
| | batch size | 128 | 128 | 128 | 128 | 128 | 128 |
| | learning rate | 0.01 | 0.001 | 0.01 | 0.001 | 0.01 | 0.001 |
| | learning rate decay | step | cosine | step | cosine | step | cosine |
| | weight decay $\lambda$ | ✗ | ✗ | $5 \times 10^{-4}$ | 0.05 | $5 \times 10^{-4}$ | 0.05 |
| | weight decay $\lambda_W$ | $5 \times 10^{-4}$ | $5 \times 10^{-4}$ | ✗ | ✗ | ✗ | ✗ |
| | weight decay $\lambda_H$ | $5 \times 10^{-4}$ | $5 \times 10^{-4}$ | ✗ | ✗ | ✗ | ✗ |
| | weight decay $\lambda_b$ | $5 \times 10^{-4}$ | $5 \times 10^{-4}$ | ✗ | ✗ | ✗ | ✗ |
| | warmup epochs | 0 | 0 | 0 | 0 | 0 | 0 |
| Data Aug. | random cropping | ✗ | ✗ | ✓ | ✓ | ✓ | ✓ |
| | horizontal flipping | ✗ | ✗ | 0.5 | 0.5 | 0.5 | 0.5 |
| | label smoothing | ✗ | ✗ | ✗ | ✗ | 0.1 | 0.1 |
| | mixup alpha | ✗ | ✗ | ✗ | ✗ | 0.8 | 0.8 |
| | cutmix alpha | ✗ | ✗ | ✗ | ✗ | 1.0 | 1.0 |
| | mixup prob. | ✗ | ✗ | ✗ | ✗ | 0.8 | 0.8 |
| | normalization | mean = $[0.4914, 0.4822, 0.4465]$, std = $[0.2023, 0.1994, 0.2010]$ | | | | | |

In Sec. 4, we train ResNet18, ResNet50, and DenseNet121 on MNIST, CIFAR10, and CIFAR100, respectively. Table 5 shows the experimental settings. In default, we train the models using setting-1 and setting-2 in the experiments of neural collapse, and apply setting-3, setting-4, setting-5, and setting-6 in the experiments of classification.

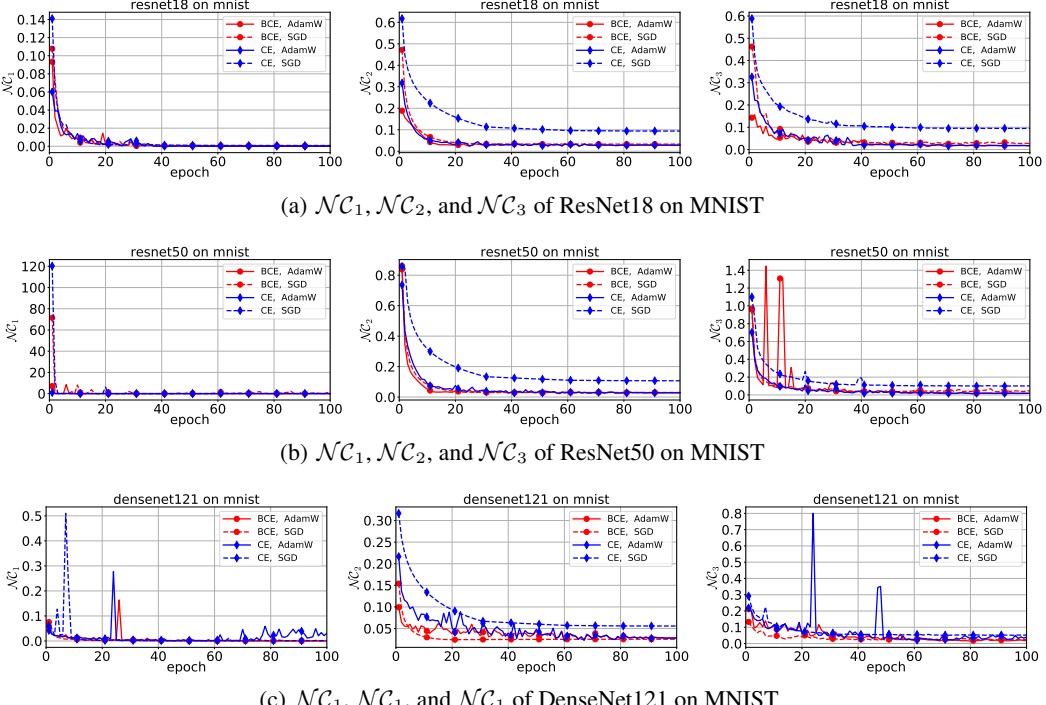

(a) $\mathcal{NC}_1$, $\mathcal{NC}_2$, and $\mathcal{NC}_3$ of ResNet18 on MNIST

(b) $\mathcal{NC}_1$, $\mathcal{NC}_2$, and $\mathcal{NC}_3$ of ResNet50 on MNIST

(c) $\mathcal{NC}_1$, $\mathcal{NC}_1$, and $\mathcal{NC}_1$ of DenseNet121 on MNIST

*Figure 4.* The evolution of the three NC metrics in the training of ResNet18 (top), ResNet50 (middle), DenseNet121 (bottom) on MNIST with CE and BCE using SGD and AdamW, respectively, with $\lambda_W = \lambda_H = \lambda_b = 5 \times 10^{-4}$.

Table 6. The numerical results of the three models trained on MNIST, with $\lambda_W = \lambda_H = \lambda_b = 5 \times 10^{-4}$.

| | | MNIST | | | |
|---|---|---|---|---|---|
| | | SGD | | AdamW | |
| | | CE | BCE | CE | BCE |
| ResNet18 | $\hat{\rho}$ | 219.0960 | 407.1362 | 212.2180 | 357.9696 |
| | $s_{\text{pos}}$ | $5.6439 \pm 0.1437$ | $6.4008 \pm 0.1236$ | $5.6331 \pm 0.0120$ | $7.7460 \pm 0.0113$ |
| | $s_{\text{neg}}$ | $-0.6302 \pm 0.2073$ | $-3.4987 \pm 0.1137$ | $-0.6259 \pm 0.0127$ | $-2.1233 \pm 0.0291$ |
| | $\hat{b}$ | $-0.0074 \pm 0.0852$ | $2.2170 \pm 0.0308$ | $0.0001 \pm 0.0328$ | $3.5134 \pm 0.0337$ |
| | $\alpha(\hat{b})$ | — | $-0.0268$ | — | $-0.0086$ |
| | $\mathcal{A}/\mathcal{A}_{\text{Uni}}$ for training | 100.00/100.00 | 100.00/100.00 | 100.00/100.00 | 100.00/100.00 |
| | $\mathcal{A}/\mathcal{A}_{\text{Uni}}$ for testing | 99.43/99.31 | 99.59/99.52 | 99.62/99.57 | 99.65/99.61 |
| ResNet50 | $\hat{\rho}$ | 217.7276 | 396.7711 | 212.2304 | 357.2365 |
| | $s_{\text{pos}}$ | $5.6383 \pm 0.6400$ | $6.5393 \pm 1.6509$ | $5.6389 \pm 0.0380$ | $7.7706 \pm 0.0573$ |
| | $s_{\text{neg}}$ | $-0.6271 \pm 0.5978$ | $-3.2512 \pm 1.9658$ | $-0.6266 \pm 0.0220$ | $-2.1029 \pm 0.0429$ |
| | $\hat{b}$ | $0.0039 \pm 0.0733$ | $2.4674 \pm 0.0492$ | $0.0001 \pm 0.0328$ | $3.5322 \pm 0.0329$ |
| | $\alpha(\hat{b})$ | — | $-0.0217$ | — | $-0.0084$ |
| | $\mathcal{A}/\mathcal{A}_{\text{Uni}}$ for training | 99.68/99.64 | 99.79/99.76 | 100.00/100.00 | 100.00/100.00 |
| | $\mathcal{A}/\mathcal{A}_{\text{Uni}}$ for testing | 98.98/98.79 | 99.01/98.88 | 99.60/99.57 | 99.53/99.52 |
| DenseNet121 | $\hat{\rho}$ | 224.1426 | 414.7491 | 212.2337 | 355.5479 |
| | $s_{\text{pos}}$ | $5.5774 \pm 0.1217$ | $6.1977 \pm 0.0987$ | $5.6318 \pm 0.1132$ | $7.8030 \pm 0.0377$ |
| | $s_{\text{neg}}$ | $-0.6193 \pm 0.1221$ | $-3.6421 \pm 0.1048$ | $-0.6258 \pm 0.3427$ | $-2.0508 \pm 0.0314$ |
| | $\hat{b}$ | $0.0010 \pm 0.0570$ | $2.0705 \pm 0.0264$ | $0.0002 \pm 0.0324$ | $3.5767 \pm 0.0344$ |
| | $\alpha(\hat{b})$ | — | $-0.0302$ | — | $-0.0081$ |
| | $\mathcal{A}/\mathcal{A}_{\text{Uni}}$ for training | 100.00/99.99 | 100.00/100.00 | 99.63/99.62 | 100.00/100.00 |
| | $\mathcal{A}/\mathcal{A}_{\text{Uni}}$ for testing | 99.45/99.40 | 99.54/99.52 | 99.29/99.22 | 99.64/99.60 |

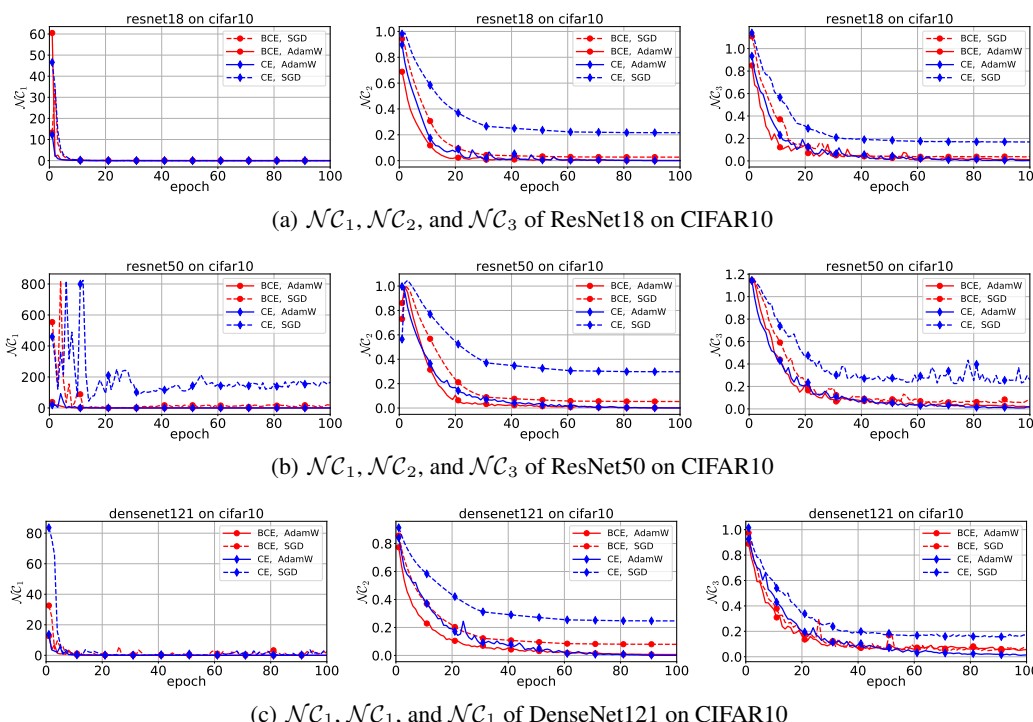

(a) $\mathcal{NC}_1$, $\mathcal{NC}_2$, and $\mathcal{NC}_3$ of ResNet18 on CIFAR10

(b) $\mathcal{NC}_1$, $\mathcal{NC}_2$, and $\mathcal{NC}_3$ of ResNet50 on CIFAR10

(c) $\mathcal{NC}_1$, $\mathcal{NC}_1$, and $\mathcal{NC}_1$ of DenseNet121 on CIFAR10

Figure 5. The evolution of the three NC metrics in the training of ResNet18 (top), ResNet50 (middle), DenseNet121 (bottom) on CIFAR10 with CE and BCE using SGD and AdamW, respectively, with $\lambda_W = \lambda_H = \lambda_b = 5 \times 10^{-4}$.

*Table 7.* The numerical results of the three models trained on CIFAR10, with $\lambda_{\boldsymbol{W}} = \lambda_{\boldsymbol{H}} = \lambda_{\boldsymbol{b}} = 5 \times 10^{-4}$.

| | | CIFAR10 | | | |
| --- | --- | --- | --- | --- | --- |
| | | SGD | | AdamW | |
| | | CE | BCE | CE | BCE |
| ResNet18 | $\hat{\rho}$ | 221.7685 | 395.3918 | 212.4173 | 366.6813 |
| | $s_{\text{pos}}$ | $5.7103 \pm 0.2252$ | $6.5627 \pm 0.2042$ | $5.6393 \pm 0.0568$ | $7.5025 \pm 0.0549$ |
| | $s_{\text{neg}}$ | $-0.6386 \pm 0.3574$ | $-3.4557 \pm 0.1939$ | $-0.6265 \pm 0.0066$ | $-2.3582 \pm 0.0225$ |
| | $\hat{b}$ | $-0.0085 \pm 0.0430$ | $2.2618 \pm 0.0678$ | $-0.0001 \pm 0.0038$ | $3.2905 \pm 0.0080$ |
| | $\alpha(\hat{b})$ | — | $-0.0266$ | — | $-0.0105$ |
| | $\mathcal{A}/\mathcal{A}_{\text{Uni}}$ for training | 99.99/99.98 | 100.00/100.00 | 100.00/100.00 | 100.00/100.00 |
| | $\mathcal{A}/\mathcal{A}_{\text{Uni}}$ for testing | 79.22/75.71 | 81.19/78.78 | 86.66/84.72 | 86.58/85.07 |
| ResNet50 | $\hat{\rho}$ | 220.8594 | 382.4440 | 212.3374 | 369.2447 |
| | $s_{\text{pos}}$ | $5.7365 \pm 8.2056$ | $6.5614 \pm 4.3923$ | $5.6386 \pm 0.1062$ | $7.4351 \pm 0.2787$ |
| | $s_{\text{neg}}$ | $-0.6439 \pm 14.1340$ | $-3.5695 \pm 7.0134$ | $-0.6266 \pm 0.0150$ | $-2.4493 \pm 0.2165$ |
| | $\hat{b}$ | $0.0045 \pm 0.1430$ | $2.4002 \pm 0.1496$ | $-0.0000 \pm 0.0053$ | $3.2051 \pm 0.0309$ |
| | $\alpha(\hat{b})$ | — | $-0.0242$ | — | $-0.0114$ |
| | $\mathcal{A}/\mathcal{A}_{\text{Uni}}$ for training | 99.61/99.52 | 99.65/99.32 | 99.99/99.99 | 100.00/100.00 |
| | $\mathcal{A}/\mathcal{A}_{\text{Uni}}$ for testing | 76.28/73.08 | 78.41/76.35 | 85.73/84.33 | 85.76/84.98 |
| DenseNet121 | $\hat{\rho}$ | 225.0609 | 392.8198 | 212.7966 | 360.5613 |
| | $s_{\text{pos}}$ | $5.7225 \pm 1.7228$ | $6.2376 \pm 0.8437$ | $5.6150 \pm 0.2851$ | $7.6743 \pm 0.1239$ |
| | $s_{\text{neg}}$ | $-0.6348 \pm 0.8664$ | $-3.6171 \pm 1.6284$ | $-0.6240 \pm 0.0330$ | $-2.1715 \pm 0.0604$ |
| | $\hat{b}$ | $0.0012 \pm 0.0364$ | $2.0875 \pm 0.1229$ | $0.0003 \pm 0.0061$ | $3.4612 \pm 0.0203$ |
| | $\alpha(\hat{b})$ | — | $-0.0318$ | — | $-0.0090$ |
| | $\mathcal{A}/\mathcal{A}_{\text{Uni}}$ for training | 99.40/99.03 | 99.72/99.62 | 99.87/99.86 | 100.00/100.00 |
| | $\mathcal{A}/\mathcal{A}_{\text{Uni}}$ for testing | 77.30/74.41 | 79.16/77.95 | 81.54/80.15 | 82.34/81.70 |

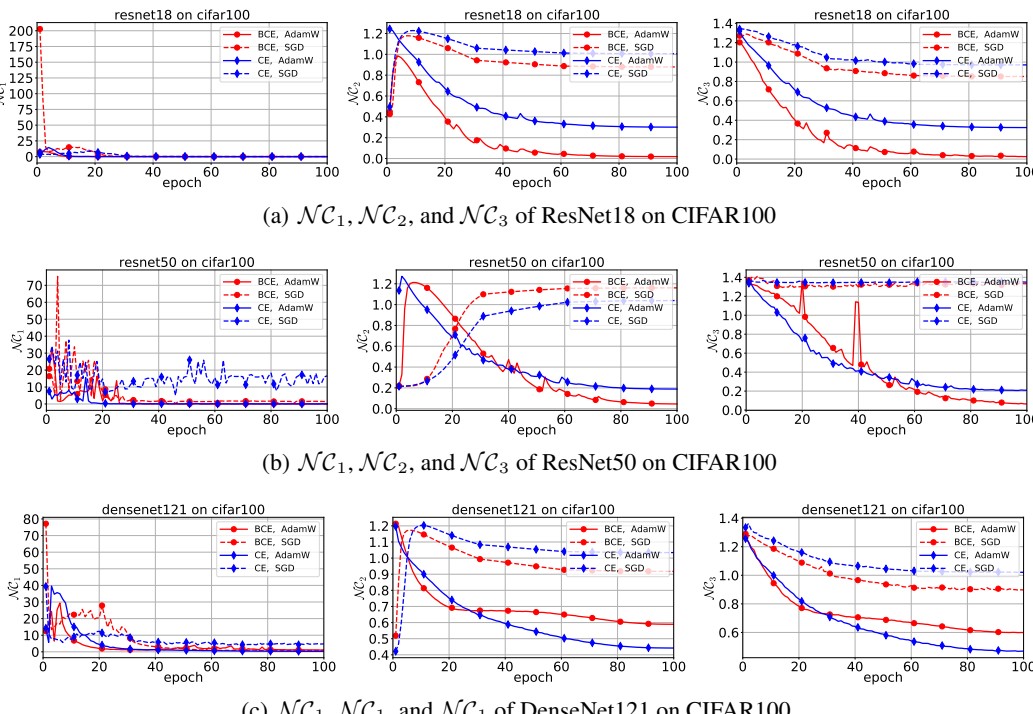

(a) $\mathcal{NC}_1, \mathcal{NC}_2$, and $\mathcal{NC}_3$ of ResNet18 on CIFAR100

(b) $\mathcal{NC}_1, \mathcal{NC}_2$, and $\mathcal{NC}_3$ of ResNet50 on CIFAR100

(c) $\mathcal{NC}_1, \mathcal{NC}_1$, and $\mathcal{NC}_1$ of DenseNet121 on CIFAR100

*Figure 6.* The evolution of the three NC metrics in the training of ResNet18 (top), ResNet50 (middle), DenseNet121 (bottom) on CIFAR100 with CE and BCE using SGD and AdamW, respectively, with $\lambda_{\boldsymbol{W}} = \lambda_{\boldsymbol{H}} = \lambda_{\boldsymbol{b}} = 5 \times 10^{-4}$.

Table 8. The numerical results of the three models trained on CIFAR100, with $\lambda_{\boldsymbol{W}} = \lambda_{\boldsymbol{H}} = \lambda_{\boldsymbol{b}} = 5 \times 10^{-4}$.

| | | CIFAR100 | | | |
| --- | --- | --- | --- | --- | --- |
| | | SGD | | AdamW | |
| | | CE | BCE | CE | BCE |
| ResNet18 | $\hat{\rho}$ | 954.3918 | 1732.6035 | 846.4734 | 1708.9231 |
| | $s_{\text{pos}}$ | $8.3613 \pm 0.4316$ | $3.5152 \pm 0.2392$ | $7.5183 \pm 0.0997$ | $4.0202 \pm 0.0696$ |
| | $s_{\text{neg}}$ | $-0.0848 \pm 1.3897$ | $-6.5934 \pm 1.2718$ | $-0.0754 \pm 0.2580$ | $-5.6834 \pm 0.0438$ |
| | $\hat{b}$ | $0.0004 \pm 0.2356$ | $0.8407 \pm 0.0678$ | $0.0005 \pm 0.0097$ | $1.1317 \pm 0.0007$ |
| | $\alpha(\hat{b})$ | — | $-0.2672$ | — | $-0.2147$ |
| | $\mathcal{A}/\mathcal{A}_{\text{Uni}}$ for training | 99.95/99.81 | 99.98/99.97 | 99.98/99.96 | 99.98/99.97 |
| | $\mathcal{A}/\mathcal{A}_{\text{Uni}}$ for testing | 34.61/17.99 | 42.06/30.61 | 56.58/47.29 | 60.48/43.04 |
| ResNet50 | $\hat{\rho}$ | 36.2794 | 289.5987 | 838.0098 | 1710.3754 |
| | $s_{\text{pos}}$ | $0.5404 \pm 9.8551$ | $-4.6656 \pm 16.0695$ | $7.3906 \pm 0.3560$ | $3.9356 \pm 1.5798$ |
| | $s_{\text{neg}}$ | $0.6182 \pm 11.4828$ | $-6.2663 \pm 29.8421$ | $-0.0745 \pm 0.1935$ | $-5.7441 \pm 1.1971$ |
| | $\hat{b}$ | $0.0006 \pm 0.0592$ | $0.3210 \pm 0.0241$ | $0.0005 \pm 0.0073$ | $1.1239 \pm 0.0044$ |
| | $\alpha(\hat{b})$ | — | $-0.4090$ | — | $-0.2160$ |
| | $\mathcal{A}/\mathcal{A}_{\text{Uni}}$ for training | 2.52/0.05 | 7.67/0.44 | 99.83/99.76 | 99.77/99.62 |
| | $\mathcal{A}/\mathcal{A}_{\text{Uni}}$ for testing | 2.48/0.06 | 7.16/0.39 | 55.51/50.77 | 53.55/49.18 |
| DenseNet121 | $\hat{\rho}$ | 894.4895 | 1597.8596 | 900.5263 | 1761.0126 |
| | $s_{\text{pos}}$ | $8.4473 \pm 0.8321$ | $3.0569 \pm 1.6496$ | $8.1030 \pm 0.4805$ | $4.0875 \pm 0.2246$ |
| | $s_{\text{neg}}$ | $-0.0842 \pm 1.6340$ | $-6.6552 \pm 2.6035$ | $-0.0800 \pm 0.4365$ | $-5.8613 \pm 0.7152$ |
| | $\hat{b}$ | $-0.0012 \pm 0.2239$ | $0.8313 \pm 0.0983$ | $0.0016 \pm 0.0948$ | $1.1306 \pm 0.0145$ |
| | $\alpha(\hat{b})$ | — | $-0.2714$ | — | $-0.2141$ |
| | $\mathcal{A}/\mathcal{A}_{\text{Uni}}$ for training | 99.15/94.38 | 99.38/99.23 | 99.80/99.78 | 99.98/99.97 |
| | $\mathcal{A}/\mathcal{A}_{\text{Uni}}$ for testing | 37.48/24.20 | 39.93/35.19 | 50.31/37.87 | 52.41/49.81 |

## B.2. Experimental results of neural collapse

In this section, we show the experimental results of neural collapse. Most of these results are calculated on the training data of the three datasets.

**NC metrics, the final classifier bias, and the final decision scores**. Figs. 4 - 6 shows the evolution of the three NC metrics in the training of ResNet18, ResNet50, DenseNet121 on MNIST, CIFAR10, and CIFAR100 with CE and BCE. In the training on MNIST and CIFAR10, the NC metrics of both CE and BCE approach zero at the terminal phase of training, and that of BCE decrease faster than that of CE at the first 20 epochs. In the training on CIFAR100, which is a more challenging dataset than MNIST and CIFAR10, the NC metrics of models trained by SGD do not decrease to zero, while that of models trained by AdamW approach zero, and the NC metrics of BCE decrease faster than that of CE in most cases. Table 6 - 8 present the numerical results of the final models at the 100th epoch. In these tables, $\hat{\rho} = \|\hat{\boldsymbol{W}}\|_F^2$, where $\hat{\boldsymbol{W}} = [\hat{\boldsymbol{w}}_1, \hat{\boldsymbol{w}}_2, \cdots, \hat{\boldsymbol{w}}_K]^T \in \mathbb{R}^{K \times d}$ is the final trained classifier weight; "$s_{\text{pos}}$" rows list the mean and standard deviations of the final positive decision scores without biases, i.e.,

$$\text{Mean}(s_{\text{pos}}) = \frac{1}{nK} \sum_{k=1}^{K} \sum_{i=1}^{n} \hat{\boldsymbol{w}}_k \boldsymbol{h}_i^{(k)}, \tag{51}$$

$$\text{Std}(s_{\text{pos}}) = \sqrt{\sum_{k=1}^{K} \sum_{i=1}^{n} \frac{\left(\hat{\boldsymbol{w}}_k \boldsymbol{h}_i^{(k)} - \text{Mean}(s_{\text{pos}})\right)^2}{nK}}, \tag{52}$$

"$s_{\text{neg}}$" rows list that of the final negative decision scores without biases, i.e.,

$$\text{Mean}(s_{\text{neg}}) = \frac{1}{nK(K-1)} \sum_{k=1}^{K} \sum_{\substack{j=1 \\ j \neq k}}^{K} \sum_{i=1}^{n} \hat{\boldsymbol{w}}_j \boldsymbol{h}_i^{(k)}, \tag{53}$$

$$\text{Std}(s_{\text{neg}}) = \sqrt{\sum_{k=1}^{K} \sum_{\substack{j=1 \\ j \neq k}}^{K} \sum_{i=1}^{n} \frac{\left(\hat{\boldsymbol{w}}_j \boldsymbol{h}_i^{(k)} - \text{Mean}(s_{\text{neg}})\right)^2}{nK(K-1)}}, \tag{54}$$

and "$\hat{b}$" rows list that of the final classifier bias $\hat{\boldsymbol{b}} = [\hat{b}_1, \hat{b}_2, \cdots, \hat{b}_K]^T \in \mathbb{R}^K$, i.e.,

$$\text{Mean}(\hat{b}) = \frac{1}{K} \sum_{k=1}^{K} \hat{b}_k, \tag{55}$$

$$\text{Std}(\hat{b}) = \sqrt{\frac{\sum_{k=1}^{K} \left(\hat{b}_k - \text{Mean}(\hat{b})\right)^2}{K}}. \tag{56}$$

"$\alpha(\hat{b})$" rows list the value of function $\alpha(b)$ at point $\text{Mean}(\hat{b})$, where

$$\alpha(b) = -\frac{K-1}{K\left(1 + \exp\left(b + \sqrt{\frac{\lambda_{\boldsymbol{W}}}{n\lambda_{\boldsymbol{H}}}} \frac{\rho}{K(K-1)}\right)\right)} + \frac{1}{K\left(1 + \exp\left(\sqrt{\frac{\lambda_{\boldsymbol{W}}}{n\lambda_{\boldsymbol{H}}}} \frac{\rho}{K} - b\right)\right)} + \lambda_{\boldsymbol{b}} b, \tag{57}$$

is the function at the RHS of Eq. (12).

Besides the classification accuracy $\mathcal{A}$ and uniform accuracy $\mathcal{A}_{\text{Uni}}$ of the final models on the training data, Tables 6, 7, and 8 have also presented that on the testing data.

*Table 9.* The numerical results of ResNet18 trained on MNIST with fixed weight decay $\lambda_{\boldsymbol{b}}$ for the classifier bias.

| Loss | Opt. | $\bar{b}$ | $\hat{\rho}$ | $s_{\text{pos}}$ | $s_{\text{neg}}$ | $\hat{b}$ | $\alpha(\hat{b})$ |
|---|---|---|---|---|---|---|---|
| CE | SGD | 0 | 218.9428 | $5.6648 \pm 0.1673$ | $-0.6323 \pm 0.2360$ | $-0.0179 \pm 0.1228$ | — |
| | | 1 | 218.8023 | $5.6337 \pm 0.1473$ | $-0.6290 \pm 0.2097$ | $0.9821 \pm 0.1149$ | — |
| | | 2 | 218.3450 | $5.6456 \pm 0.1556$ | $-0.6318 \pm 0.2213$ | $1.9821 \pm 0.1122$ | — |
| | | 3 | 218.3319 | $5.6399 \pm 0.1521$ | $-0.6295 \pm 0.2132$ | $2.9821 \pm 0.1163$ | — |
| | | 4 | 219.2994 | $5.6628 \pm 0.1600$ | $-0.6321 \pm 0.2281$ | $3.9820 \pm 0.1307$ | — |
| | | 5 | 219.5797 | $5.6611 \pm 0.1780$ | $-0.6329 \pm 0.2411$ | $4.9820 \pm 0.1279$ | — |
| | | 6 | 220.0522 | $5.6458 \pm 0.1598$ | $-0.6301 \pm 0.2245$ | $5.9820 \pm 0.1312$ | — |
| | | 8 | 219.4256 | $5.6410 \pm 0.1608$ | $-0.6311 \pm 0.2284$ | $7.9821 \pm 0.1194$ | — |
| | | 10 | 219.2911 | $5.6411 \pm 0.1601$ | $-0.6300 \pm 0.2152$ | $9.9821 \pm 0.1250$ | — |
| | AdamW | 0 | 212.2146 | $5.6360 \pm 0.0250$ | $-0.6262 \pm 0.0189$ | $-0.0180 \pm 0.0486$ | — |
| | | 1 | 212.2138 | $5.6355 \pm 0.0353$ | $-0.6262 \pm 0.0194$ | $0.9828 \pm 0.0493$ | — |
| | | 2 | 212.2151 | $5.6336 \pm 0.0258$ | $-0.6260 \pm 0.0189$ | $1.9821 \pm 0.0487$ | — |
| | | 3 | 212.2152 | $5.6336 \pm 0.0264$ | $-0.6260 \pm 0.0189$ | $2.9825 \pm 0.0486$ | — |
| | | 4 | 212.2161 | $5.6307 \pm 0.0274$ | $-0.6257 \pm 0.0191$ | $3.9823 \pm 0.0491$ | — |
| | | 5 | 212.2143 | $5.6308 \pm 0.0264$ | $-0.6257 \pm 0.0189$ | $4.9809 \pm 0.0486$ | — |
| | | 6 | 212.2143 | $5.6323 \pm 0.0264$ | $-0.6258 \pm 0.0189$ | $5.9822 \pm 0.0486$ | — |
| | | 8 | 212.2163 | $5.6347 \pm 0.0262$ | $-0.6261 \pm 0.0189$ | $7.9812 \pm 0.0486$ | — |
| | | 10 | 212.2151 | $5.6340 \pm 0.0263$ | $-0.6260 \pm 0.0189$ | $9.9829 \pm 0.0486$ | — |
| BCE | SGD | 0 | 393.2500 | $7.1748 \pm 0.1277$ | $-2.8219 \pm 0.1379$ | $3.0789 \pm 0.0489$ | $-0.0120$ |
| | | 1 | 374.9337 | $7.7515 \pm 0.1578$ | $-2.2877 \pm 0.1468$ | $3.6658 \pm 0.0709$ | $-0.0070$ |
| | | 2 | 362.5949 | $8.1822 \pm 0.1525$ | $-1.9121 \pm 0.1604$ | $4.1078 \pm 0.1053$ | $-0.0045$ |
| | | 3 | 355.2978 | $8.5608 \pm 0.1634$ | $-1.6192 \pm 0.1568$ | $4.4557 \pm 0.0981$ | $-0.0030$ |
| | | 4 | 354.6479 | $8.8711 \pm 0.1473$ | $-1.3347 \pm 0.1725$ | $4.7949 \pm 0.1094$ | $-0.0019$ |
| | | 5 | 355.9634 | $9.2305 \pm 0.1503$ | $-1.0452 \pm 0.1960$ | $5.1493 \pm 0.1192$ | $-0.0009$ |
| | | 6 | 361.1938 | $9.5688 \pm 0.1355$ | $-0.7519 \pm 0.1688$ | $5.5084 \pm 0.0869$ | $-0.0002$ |
| | | 8 | 385.6802 | $10.3761 \pm 0.1400$ | $-0.0997 \pm 0.2436$ | $6.3418 \pm 0.0989$ | $0.0007$ |
| | | 10 | 426.3013 | $11.5173 \pm 0.1430$ | $0.7786 \pm 0.3075$ | $7.4858 \pm 0.1021$ | $0.0010$ |
| | AdamW | 0 | 350.4272 | $9.3081 \pm 0.0352$ | $-1.0348 \pm 0.0321$ | $5.2388 \pm 0.0609$ | $-0.0006$ |
| | | 1 | 350.4283 | $9.3015 \pm 0.0345$ | $-1.0340 \pm 0.0321$ | $5.2389 \pm 0.0609$ | $-0.0006$ |
| | | 2 | 350.4292 | $9.3029 \pm 0.0357$ | $-1.0342 \pm 0.0321$ | $5.2388 \pm 0.0609$ | $-0.0006$ |
| | | 3 | 350.4275 | $9.3028 \pm 0.0364$ | $-1.0342 \pm 0.0321$ | $5.2388 \pm 0.0609$ | $-0.0006$ |
| | | 4 | 350.4248 | $9.3039 \pm 0.0362$ | $-1.0343 \pm 0.0320$ | $5.2388 \pm 0.0609$ | $-0.0006$ |
| | | 5 | 350.4250 | $9.3100 \pm 0.0358$ | $-1.0350 \pm 0.0320$ | $5.2388 \pm 0.0608$ | $-0.0006$ |
| | | 6 | 350.4302 | $9.3063 \pm 0.0345$ | $-1.0346 \pm 0.0321$ | $5.2388 \pm 0.0608$ | $-0.0006$ |
| | | 8 | 350.4304 | $9.3094 \pm 0.0356$ | $-1.0349 \pm 0.0321$ | $5.2389 \pm 0.0609$ | $-0.0006$ |
| | | 10 | 350.4330 | $9.3109 \pm 0.0369$ | $-1.0351 \pm 0.0321$ | $5.2388 \pm 0.0609$ | $-0.0006$ |

**The failures in the experiments of neural collapse.** According to the above figures and tables, one can find the models trained with SGD are easily to fail in the experiments of neural collapse, including the ResNet50 trained on MNIST,

ResNet50 and DenseNet121 trained on CIFAR10, and the three models trained CIFAR100. The standard deviations of positive/negative decision scores produced by these models are usually larger than $0.5$. These failed models in the neural collapse can be roughly classified into two types:

- The two ResNet50 trained on CIFAR100 with SGD. They are completely failed models. The standard deviations of the decision scores are very high, even more than $20$, and, for the BCE-trained model, the means of the positive and negative decision scores are relatively close, while for the CE-trained model, the mean of positive scores is even less than that of negative ones, indicating that most of the samples were not correctly classified. The classification accuracy $\mathcal{A}$ on the training dataset are only $2.52\%$ and $7.67\%$ with CE and BCE. The ResNet50 trained on CIFAR10 with SGD and CE. On the training dataset, there exists a clear gap between the means of positive and negative decision scores, while the standard deviations are still very high. In correspond to the decision score results, its classification accuracy $\mathcal{A}$ is $99.61\%$, almost $100\%$, while the uniform classification accuracy $\mathcal{A}_{\text{Uni}}$ is only $0.01\%$. In other words, the almost all samples have been correctly classified, while the intra-class compactness and inter-class distinctiveness are very bad. As a comparison, the BCE-trained ResNet50 on CIFAR10 with SGD achieves high classification performance and uniform classification performance.

  In contrast, on the testing dataset, the CE-trained ResNet50 achieves relatively high classification accuracy and uniform classification accuracy, $76.28\%$ and $72.51\%$. The generalization of uniform classification performance of models is still an unresolved issue.

  Fig. presents the feature distributions of these two ResNet50 on the training and testing data of CIFAR10.

- The other failed models trained with SGD, including the ResNet50 trained on MNIST and CIFAR10, DenseNet121 trained on CIFAR10, ResNet18 and DenseNet121 trained on CIFAR100. These models have achieved almost $100\%$ classification accuracy and uniform accuracy on the training dataset. However, according to the standard deviations of decision scores and the NC metrics, we conclude that they do not reach the state of neural collapse.

These failures in the experiments of neural collapse reveal more relationships among classification and neural collapse. In the training, zero classification error appears before zero uniform classification error, which appears before the neural collapse, or, in contrary, the model reaching the neural collapse has the uniform accuracy of $100\%$, and the model with the uniform accuracy of $100\%$ has also the accuracy $100\%$ on the classification. Both the reverses are not true.

**The bias decay parameter** $\lambda_{\boldsymbol{b}}$. In Sec. 4, we conducted experiments with fixed $\lambda_{\boldsymbol{b}} = 0$ and varying $\lambda_{\boldsymbol{b}} = 0.5, 0.05, 5 \times 10^{-3}, 5 \times 10^{-4}, 5 \times 10^{-5}, 5 \times 10^{-6}$ to further compare CE and BCE in neural collapse. Fig. 3 have visually shown the results, and we here present the numerical results in Tables 9 and 10. In our experiments, the classifier weight $\boldsymbol{W}$ and bias $\boldsymbol{b}$ are initialized using "kaiming uniform", i.e., He initialization (He et al., 2015). The initialized classifier bias is with zero-mean, i.e., $\frac{1}{K}\sum_{k=1}^{K} b_k \approx 0$, and we add them with $0, 1, 2, 3, 4, 5, 6, 8, 10$, respectively, to adjust their average value in the experiments with fixed $\lambda_{\boldsymbol{b}}$.

**The batch size**. In the proof of Theorem 3.1 and 3.2, it applied the feature matrix $\boldsymbol{H}$ including the features of all samples, to explore the the lower bounds for the CE and BCE losses, i.e.,

$$\boldsymbol{H} = \left[ h_1^{(1)}, h_2^{(1)}, \cdots, h_n^{(1)}, h_1^{(2)}, h_2^{(2)}, \cdots, h_n^{(2)}, \cdots, h_1^{(K)}, h_2^{(K)}, \cdots, h_n^{(K)} \right]. \tag{58}$$

However, batch algorithm was applied in the practical training of deep models, and the batch size would affect the experimental numerical results. To verify this conclusion, a group of experiments were conducted with varying batch size. We trained ResNet18 on MNIST using SGD and AdamW using setting-1 and setting-2, while the initial learning rates were adjusted according to the batch size, $0.01 \times \frac{\text{batch size}}{128}$ for SGD and $0.001 \times \frac{\text{batch size}}{128}$ for AdamW. Fig. 7 visually shows the distributions of the final classifier bias and the positive/negative decision scores, and Table 11 lists the final numerical results. From these results, one can find that the bias results still conform to our analysis when batch size $\leq 1024$, i.e., the classifier bias converges to zero in the training with CE loss and $\lambda_{\boldsymbol{b}} > 0$, and the clssifier bias separates the positive and negative decision scores in the training with BCE loss.

The decision score results are very different from that in the experiments with fixed batch size. For examples, in the training with CE loss and fixed batch size $= 128$, the positive and negative decision scores converge to about $5.64$ and $-0.63$, respectively, and the value of $\hat{\rho} = \|\hat{\boldsymbol{W}}\|_F^2$ converge to about $219$ and $212$ in the training by SGD and AdamW, respectively, as shown in Tables 9 and 10. In contrast, these values varies as the batch size in the experiments with varying batch sizes.

*Table 10.* The numerical results of ResNet18 trained on MNIST with varying weight decay $\lambda_b$ for the classifier bias.

| Loss | Opt. | $\lambda_b$ | $\hat{\rho}$ | $s_{\text{pos}}$ | $s_{\text{neg}}$ | $\hat{b}$ | $\alpha(\hat{b})$ |
|------|------|-------------|--------------|------------------|------------------|-----------|-------------------|
| CE | SGD | $5 \times 10^{-1}$ | 218.6677 | $5.6511 \pm 0.1144$ | $-0.6304 \pm 0.1854$ | $-0.0000 \pm 0.0002$ | — |
| | | $5 \times 10^{-2}$ | 218.6658 | $5.6662 \pm 0.1176$ | $-0.6321 \pm 0.2031$ | $-0.0000 \pm 0.0017$ | — |
| | | $5 \times 10^{-3}$ | 218.5622 | $5.6427 \pm 0.1076$ | $-0.6296 \pm 0.1917$ | $0.0013 \pm 0.0156$ | — |
| | | $5 \times 10^{-4}$ | 219.4882 | $5.6527 \pm 0.1287$ | $-0.6322 \pm 0.2352$ | $4.0998 \pm 0.0796$ | — |
| | | $5 \times 10^{-5}$ | 219.0555 | $5.6526 \pm 0.1407$ | $-0.6310 \pm 0.2192$ | $9.1337 \pm 0.1038$ | — |
| | | $5 \times 10^{-6}$ | 219.2227 | $5.6426 \pm 0.1507$ | $-0.6307 \pm 0.2209$ | $9.8940 \pm 0.1111$ | — |
| | AdamW | $5 \times 10^{-1}$ | 212.2359 | $5.6329 \pm 0.0340$ | $-0.6259 \pm 0.0037$ | $-0.0000 \pm 0.0001$ | — |
| | | $5 \times 10^{-2}$ | 212.2369 | $5.6372 \pm 0.0335$ | $-0.6264 \pm 0.0037$ | $0.0000 \pm 0.0010$ | — |
| | | $5 \times 10^{-3}$ | 212.2328 | $5.6382 \pm 0.0186$ | $-0.6265 \pm 0.0038$ | $0.0000 \pm 0.0083$ | — |
| | | $5 \times 10^{-4}$ | 212.2152 | $5.6339 \pm 0.0257$ | $-0.6260 \pm 0.0128$ | $0.0010 \pm 0.0324$ | — |
| | | $5 \times 10^{-5}$ | 212.2158 | $5.6316 \pm 0.0221$ | $-0.6257 \pm 0.0174$ | $3.4803 \pm 0.0448$ | — |
| | | $5 \times 10^{-6}$ | 212.2147 | $5.6330 \pm 0.0256$ | $-0.6259 \pm 0.0186$ | $8.9169 \pm 0.0480$ | — |
| BCE | SGD | $5 \times 10^{-1}$ | 472.0906 | $4.2473 \pm 0.1306$ | $-5.6495 \pm 0.1260$ | $0.0036 \pm 0.0000$ | $-0.1683$ |
| | | $5 \times 10^{-2}$ | 471.6918 | $4.2916 \pm 0.1134$ | $-5.5975 \pm 0.1029$ | $0.0362 \pm 0.0003$ | $-0.1640$ |
| | | $5 \times 10^{-3}$ | 452.0422 | $4.6706 \pm 0.1199$ | $-5.1987 \pm 0.0936$ | $0.4031 \pm 0.0037$ | $-0.1269$ |
| | | $5 \times 10^{-4}$ | 358.9137 | $9.0244 \pm 0.1190$ | $-0.7897 \pm 0.1281$ | $4.8403 \pm 0.0604$ | $-0.0018$ |
| | | $5 \times 10^{-5}$ | 414.4364 | $11.0715 \pm 0.1306$ | $0.5388 \pm 0.2787$ | $7.0401 \pm 0.0959$ | $0.0008$ |
| | | $5 \times 10^{-6}$ | 424.8451 | $11.4847 \pm 0.1327$ | $0.7536 \pm 0.3067$ | $7.4372 \pm 0.0973$ | $0.0010$ |
| | AdamW | $5 \times 10^{-1}$ | 483.3321 | $4.2399 \pm 0.0308$ | $-5.6315 \pm 0.0215$ | $0.0036 \pm 0.0000$ | $-0.1636$ |
| | | $5 \times 10^{-2}$ | 482.1844 | $4.2698 \pm 0.0306$ | $-5.5977 \pm 0.0213$ | $0.0358 \pm 0.0003$ | $-0.1598$ |
| | | $5 \times 10^{-3}$ | 470.6640 | $4.5928 \pm 0.0281$ | $-5.2753 \pm 0.0201$ | $0.3577 \pm 0.0033$ | $-0.1256$ |
| | | $5 \times 10^{-4}$ | 356.5036 | $7.7870 \pm 0.0130$ | $-2.0822 \pm 0.0285$ | $3.5514 \pm 0.0330$ | $-0.0083$ |
| | | $5 \times 10^{-5}$ | 347.1199 | $9.0726 \pm 0.0303$ | $-1.1593 \pm 0.0304$ | $4.9903 \pm 0.0537$ | $-0.0012$ |
| | | $5 \times 10^{-6}$ | 350.0225 | $9.2915 \pm 0.0372$ | $-1.0489 \pm 0.0319$ | $5.2119 \pm 0.0599$ | $-0.0006$ |

*Table 11.* The numerical results of ResNet18 trained on MNIST with varying batch size and $\lambda_W = \lambda_H = \lambda_b = 5 \times 10^{-4}$.

| Loss | Opt. | batch size | $\hat{\rho}$ | $s_{\text{pos}}$ | $s_{\text{neg}}$ | $\hat{b}$ | $\alpha(\hat{b})$ |
|------|------|------------|--------------|------------------|------------------|-----------|-------------------|
| CE | SGD | 16 | 100.9731 | $6.7176 \pm 0.3270$ | $-0.7538 \pm 0.1950$ | $-0.0074 \pm 0.0523$ | — |
| | | 32 | 130.1404 | $6.3375 \pm 0.2425$ | $-0.7110 \pm 0.1709$ | $-0.0074 \pm 0.0478$ | — |
| | | 64 | 168.6290 | $6.0159 \pm 0.1562$ | $-0.6737 \pm 0.2052$ | $-0.0074 \pm 0.0547$ | — |
| | | 128 | 219.0960 | $5.6439 \pm 0.1437$ | $-0.6302 \pm 0.2073$ | $-0.0074 \pm 0.0852$ | — |
| | | 256 | 285.6314 | $5.3200 \pm 0.1586$ | $-0.5936 \pm 0.2070$ | $-0.0074 \pm 0.1259$ | — |
| | | 512 | 379.3403 | $4.9776 \pm 0.2735$ | $-0.5535 \pm 0.2921$ | $-0.0073 \pm 0.2526$ | — |
| | | 1024 | 522.5523 | $4.6562 \pm 1.3926$ | $-0.5173 \pm 0.8343$ | $-0.0073 \pm 1.0641$ | — |
| | | 2048 | 473.7898 | $3.5759 \pm 2.6771$ | $-0.3972 \pm 2.0373$ | $-0.0072 \pm 1.8399$ | — |
| | AdamW | 16 | 87.6451 | $6.5511 \pm 0.0110$ | $-0.7279 \pm 0.0089$ | $0.0003 \pm 0.0211$ | — |
| | | 32 | 118.0328 | $6.2558 \pm 0.0101$ | $-0.6951 \pm 0.0104$ | $0.0003 \pm 0.0253$ | — |
| | | 64 | 158.4980 | $5.9506 \pm 0.0106$ | $-0.6612 \pm 0.0117$ | $0.0002 \pm 0.0293$ | — |
| | | 128 | 212.2180 | $5.6331 \pm 0.0120$ | $-0.6259 \pm 0.0127$ | $0.0001 \pm 0.0328$ | — |
| | | 256 | 282.9370 | $5.3168 \pm 0.0148$ | $-0.5908 \pm 0.0133$ | $0.0000 \pm 0.0357$ | — |
| | | 512 | 375.4274 | $4.9968 \pm 0.0209$ | $-0.5552 \pm 0.0140$ | $-0.0001 \pm 0.0380$ | — |
| | | 1024 | 496.6912 | $4.6627 \pm 0.0631$ | $-0.5199 \pm 0.0238$ | $-0.0190 \pm 0.0472$ | — |
| | | 2048 | 668.3063 | $4.3236 \pm 0.3703$ | $-0.4906 \pm 0.2909$ | $-0.0153 \pm 0.2964$ | — |
| BCE | SGD | 16 | 199.6890 | $6.1841 \pm 0.3002$ | $-5.9379 \pm 0.2665$ | $0.7828 \pm 0.0223$ | $-0.0660$ |
| | | 32 | 255.9898 | $6.1508 \pm 0.2184$ | $-5.2761 \pm 0.1932$ | $1.1506 \pm 0.0214$ | $-0.0546$ |
| | | 64 | 324.7408 | $6.2846 \pm 0.1600$ | $-4.4319 \pm 0.1295$ | $1.6456 \pm 0.0254$ | $-0.0399$ |
| | | 128 | 407.1362 | $6.4008 \pm 0.1236$ | $-3.4987 \pm 0.1137$ | $2.2170 \pm 0.0308$ | $-0.0268$ |
| | | 256 | 501.1286 | $6.6422 \pm 0.1347$ | $-2.5493 \pm 0.1501$ | $2.8605 \pm 0.0740$ | $-0.0167$ |
| | | 512 | 631.7796 | $6.6413 \pm 0.2725$ | $-1.9155 \pm 0.2544$ | $3.2338 \pm 0.1859$ | $-0.0127$ |
| | | 1024 | 816.6544 | $6.3274 \pm 0.4653$ | $-1.5393 \pm 0.4515$ | $3.3466 \pm 0.3554$ | $-0.0119$ |
| | | 2048 | 351.9647 | $1.7449 \pm 2.4487$ | $-0.5243 \pm 1.6982$ | $2.6332 \pm 1.5391$ | $0.0077$ |
| | AdamW | 16 | 189.2794 | $6.5169 \pm 0.0240$ | $-5.3841 \pm 0.0215$ | $1.2651 \pm 0.0119$ | $-0.0457$ |
| | | 32 | 242.1592 | $6.7110 \pm 0.0169$ | $-4.5302 \pm 0.0202$ | $1.7885 \pm 0.0167$ | $-0.0322$ |
| | | 64 | 300.8807 | $7.1079 \pm 0.0118$ | $-3.4518 \pm 0.0229$ | $2.5261 \pm 0.0234$ | $-0.0188$ |
| | | 128 | 357.9696 | $7.7460 \pm 0.0113$ | $-2.1233 \pm 0.0291$ | $3.5134 \pm 0.0337$ | $-0.0086$ |
| | | 256 | 455.2137 | $7.6247 \pm 0.0112$ | $-1.6013 \pm 0.0256$ | $3.8010 \pm 0.0325$ | $-0.0068$ |
| | | 512 | 590.9918 | $7.2831 \pm 0.0271$ | $-1.3210 \pm 0.0270$ | $3.8500 \pm 0.0375$ | $-0.0064$ |
| | | 1024 | 790.8874 | $6.6204 \pm 0.1011$ | $-1.3148 \pm 0.1126$ | $3.5830 \pm 0.0899$ | $-0.0089$ |
| | | 2048 | 1019.6438 | $5.9625 \pm 0.2969$ | $-1.2607 \pm 0.2750$ | $3.3303 \pm 0.2111$ | $-0.0122$ |

*Figure 7.* The distributions of the final classifier bias and positive/negative decision scores for ResNet18 trained on MNIST with different batch sizes, while $\lambda_{\boldsymbol{W}} = \lambda_{\boldsymbol{H}} = \lambda_{\boldsymbol{b}} = 5 \times 10^{-4}$.

In addition, the positive/negative decision scores did not converge to the theoretical values in Eq. (14) in our experiments; we believe it is resulted from the difference between the batch algorithm and the proof of Theorems. We roughly replaced $n$ with $\frac{\text{batch size}}{K}$ in computing $\alpha(\hat{b})$.

### B.3. Experimental results of classification

In the experiments of classification in Sec. 4.2, we train the models for 100 epochs. In each training, the model with best classification accuracy $\mathcal{A}$ is chosen as the final model, which was used to compute the uniform accuracy $\mathcal{A}_{\text{Uni}}$ presented in Table 1. In Table 12 and 13, we list their numerical results on the training and test dataset of CIFAR10 and CIFAR100. In these experiments, though the classification accuracy $\mathcal{A}$ of some models on the training datasets have reached $100\%$, neural collapse has not caused during the training. An obvious evidence is that both positive and negative decision scores have not converged, with large standard deviations, whether on the training set or testing set. The small standard deviations of the final classification bias might be more resulted from their initialization.

From Tables 12 and 13, one can find that, the gaps between the means of positive and negative decision scores of BCE-trained models are usually larger than that of CE-trained models, while in some cases, the standard deviations of the positive/negative decision scores of BCE-trained models are higher than that of CE-trained models. However, without any modification, the standard deviations and the gap between the positive and negative means cannot be precisely used to evaluate the intra-class compactness and inter-class distinctiveness. The decision score is calculated by the norm of the classifier vector and the feature vector, with the angle between them. The diverse $\hat{\rho}$ of CE-trained and BCE-trained models indicates different norms of the classifier vectors.

*Table 12.* The numerical results of ResNet18, ResNet50, DenseNet121 trained on CIFAR10 for classification.

| $\mathcal{M}$ | Op. | DA | Loss | classifier | | on training data | | | | on testing data | |
|---|---|---|---|---|---|---|---|---|---|---|---|
| | | | | $\hat{\rho}$ | $\hat{b}$ | $s_{\text{pos}}$ | $s_{\text{neg}}$ | $\mathcal{A}$ | $\mathcal{A}_{\text{Uni}}$ | $s_{\text{pos}}$ | $s_{\text{neg}}$ |
| ResNet18 | SGD | 1 | CE | 34.86 | $-0.01 \pm 0.03$ | $14.9 \pm 3.54$ | $-1.68 \pm 2.64$ | 99.98 | 97.55 | $13.9 \pm 4.73$ | $-1.56 \pm 2.89$ |
| | | | BCE | 52.33 | $2.89 \pm 0.03$ | $12.9 \pm 2.75$ | $-9.70 \pm 2.67$ | 100.00 | 99.99 | $11.3 \pm 4.93$ | $-9.25 \pm 3.30$ |
| | | 1&2 | CE | 12.59 | $-0.01 \pm 0.02$ | $3.23 \pm 0.38$ | $-0.37 \pm 0.62$ | 98.02 | 95.99 | $3.09 \pm 0.61$ | $-0.35 \pm 0.70$ |
| | | | BCE | 19.66 | $2.84 \pm 0.02$ | $3.86 \pm 0.45$ | $-0.86 \pm 0.66$ | 98.71 | 98.01 | $3.66 \pm 0.80$ | $-0.84 \pm 0.77$ |
| | AdamW | 1 | CE | 85.52 | $-0.00 \pm 0.01$ | $12.3 \pm 3.60$ | $-12.4 \pm 4.12$ | 99.99 | 99.57 | $10.9 \pm 5.52$ | $-12.1 \pm 4.58$ |
| | | | BCE | 113.9 | $2.16 \pm 0.02$ | $16.3 \pm 2.95$ | $-20.0 \pm 4.50$ | 100.00 | 100.00 | $14.2 \pm 6.37$ | $-19.0 \pm 5.75$ |
| | | 1&2 | CE | 36.26 | $-0.01 \pm 0.01$ | $2.54 \pm 0.18$ | $-1.13 \pm 0.38$ | 99.96 | 99.88 | $2.41 \pm 0.52$ | $-1.12 \pm 0.50$ |
| | | | BCE | 44.16 | $2.14 \pm 0.01$ | $3.57 \pm 0.20$ | $-1.74 \pm 0.38$ | 99.96 | 99.94 | $3.34 \pm 0.81$ | $-1.72 \pm 0.56$ |
| ResNet50 | SGD | 1 | CE | 18.74 | $0.00 \pm 0.03$ | $17.4 \pm 3.16$ | $-2.00 \pm 3.30$ | 99.99 | 98.09 | $16.1 \pm 4.56$ | $-1.86 \pm 3.73$ |
| | | | BCE | 29.07 | $2.83 \pm 0.04$ | $13.7 \pm 2.33$ | $-12.4 \pm 3.07$ | 99.99 | 99.98 | $11.9 \pm 5.08$ | $-11.8 \pm 3.83$ |
| | | 1&2 | CE | 8.18 | $0.00 \pm 0.04$ | $3.28 \pm 0.35$ | $-0.39 \pm 0.56$ | 98.25 | 96.65 | $3.14 \pm 0.63$ | $-0.37 \pm 0.66$ |
| | | | BCE | 13.86 | $2.65 \pm 0.03$ | $3.68 \pm 0.45$ | $-1.08 \pm 0.61$ | 98.79 | 98.24 | $3.47 \pm 0.85$ | $-1.06 \pm 0.75$ |
| | AdamW | 1 | CE | 143.9 | $0.01 \pm 0.02$ | $16.7 \pm 5.64$ | $-18.6 \pm 6.76$ | 100.00 | 98.95 | $14.9 \pm 7.87$ | $-18.2 \pm 7.23$ |
| | | | BCE | 153.4 | $2.20 \pm 0.01$ | $21.9 \pm 6.82$ | $-28.5 \pm 9.09$ | 99.97 | 99.96 | $19.4 \pm 10.1$ | $-27.2 \pm 10.4$ |
| | | 1&2 | CE | 79.80 | $0.00 \pm 0.01$ | $2.44 \pm 0.25$ | $-1.16 \pm 0.33$ | 99.96 | 99.89 | $2.28 \pm 0.57$ | $-1.16 \pm 0.44$ |
| | | | BCE | 102.6 | $2.14 \pm 0.00$ | $3.35 \pm 0.24$ | $-1.58 \pm 0.44$ | 99.95 | 99.94 | $3.16 \pm 0.72$ | $-1.55 \pm 0.56$ |
| DenseNet121 | SGD | 1 | CE | 48.02 | $0.00 \pm 0.02$ | $10.5 \pm 2.37$ | $-1.16 \pm 2.18$ | 99.30 | 93.29 | $9.57 \pm 3.41$ | $-1.06 \pm 2.41$ |
| | | | BCE | 64.94 | $2.93 \pm 0.03$ | $9.06 \pm 1.76$ | $-6.05 \pm 1.74$ | 99.45 | 99.24 | $7.75 \pm 3.63$ | $-5.71 \pm 2.32$ |
| | | 1&2 | CE | 14.99 | $0.00 \pm 0.02$ | $2.89 \pm 0.67$ | $-0.32 \pm 0.67$ | 91.20 | 86.80 | $2.77 \pm 0.80$ | $-0.30 \pm 0.71$ |
| | | | BCE | 19.60 | $2.86 \pm 0.02$ | $3.69 \pm 0.88$ | $-0.65 \pm 0.73$ | 92.38 | 90.28 | $3.52 \pm 1.08$ | $-0.62 \pm 0.80$ |
| | AdamW | 1 | CE | 139.4 | $0.00 \pm 0.01$ | $10.2 \pm 3.06$ | $-10.6 \pm 4.44$ | 99.97 | 98.48 | $8.70 \pm 5.00$ | $-10.4 \pm 4.86$ |
| | | | BCE | 156.6 | $2.17 \pm 0.01$ | $13.1 \pm 2.78$ | $-15.1 \pm 4.22$ | 99.97 | 99.97 | $10.9 \pm 5.93$ | $-14.4 \pm 5.13$ |
| | | 1&2 | CE | 39.93 | $0.00 \pm 0.01$ | $2.31 \pm 0.28$ | $-1.28 \pm 0.48$ | 98.83 | 98.10 | $2.14 \pm 0.64$ | $-1.26 \pm 0.58$ |
| | | | BCE | 40.53 | $2.18 \pm 0.01$ | $3.40 \pm 0.42$ | $-1.65 \pm 0.52$ | 98.81 | 98.51 | $3.13 \pm 0.94$ | $-1.60 \pm 0.67$ |

*Table 13.* The numerical results of ResNet18, ResNet50, DenseNet121 trained on CIFAR100 for classification.

| $\mathcal{M}$ | Opt. | DA | Loss | classifier | | on training data | | | | on testing data | |
|---|---|---|---|---|---|---|---|---|---|---|---|
| | | | | $\hat{\rho}$ | $\hat{b}$ | $s_{\text{pos}}$ | $s_{\text{neg}}$ | $\mathcal{A}$ | $\mathcal{A}_{\text{Uni}}$ | $s_{\text{pos}}$ | $s_{\text{neg}}$ |
| ResNet18 | SGD | 1 | CE | 317.6 | $0.00 \pm 0.02$ | $15.8 \pm 3.15$ | $-0.18 \pm 3.04$ | 99.79 | 76.32 | $13.0 \pm 5.06$ | $-0.15 \pm 3.06$ |
| | | | BCE | 408.8 | $2.89 \pm 0.02$ | $9.39 \pm 2.87$ | $-10.0 \pm 2.94$ | 99.94 | 99.69 | $5.28 \pm 5.95$ | $-9.64 \pm 3.06$ |
| | | 1&2 | CE | 138.8 | $0.00 \pm 0.02$ | $5.82 \pm 1.33$ | $-0.07 \pm 0.98$ | 88.26 | 73.67 | $5.09 \pm 1.67$ | $-0.06 \pm 1.00$ |
| | | | BCE | 163.7 | $2.89 \pm 0.01$ | $3.42 \pm 1.44$ | $-3.22 \pm 0.99$ | 88.56 | 80.23 | $2.64 \pm 1.90$ | $-3.19 \pm 1.02$ |
| | AdamW | 1 | CE | 1007. | $0.00 \pm 0.02$ | $12.5 \pm 4.18$ | $-13.4 \pm 5.16$ | 99.98 | 92.02 | $7.47 \pm 7.81$ | $-13.1 \pm 5.19$ |
| | | | BCE | 1372. | $2.14 \pm 0.02$ | $15.3 \pm 4.85$ | $-21.2 \pm 6.34$ | 99.98 | 99.97 | $7.05 \pm 10.2$ | $-19.7 \pm 6.47$ |
| | | 1&2 | CE | 476.9 | $0.00 \pm 0.02$ | $4.49 \pm 0.82$ | $-2.04 \pm 0.99$ | 99.25 | 95.86 | $3.15 \pm 1.77$ | $-2.14 \pm 1.09$ |
| | | | BCE | 576.1 | $2.18 \pm 0.02$ | $3.67 \pm 0.80$ | $-4.13 \pm 0.84$ | 99.18 | 98.25 | $2.22 \pm 1.84$ | $-4.01 \pm 0.96$ |
| ResNet50 | SGD | 1 | CE | 258.8 | $0.00 \pm 0.01$ | $17.7 \pm 3.12$ | $-0.19 \pm 3.56$ | 99.90 | 79.70 | $14.5 \pm 5.17$ | $-0.16 \pm 3.58$ |
| | | | BCE | 328.3 | $2.87 \pm 0.01$ | $10.0 \pm 2.82$ | $-11.6 \pm 3.40$ | 99.86 | 99.62 | $5.37 \pm 6.20$ | $-10.9 \pm 3.46$ |
| | | 1&2 | CE | 102.7 | $0.00 \pm 0.01$ | $5.97 \pm 1.41$ | $-0.07 \pm 1.07$ | 87.46 | 72.45 | $5.29 \pm 1.71$ | $-0.06 \pm 1.05$ |
| | | | BCE | 118.4 | $2.86 \pm 0.01$ | $3.59 \pm 1.48$ | $-3.33 \pm 0.98$ | 89.17 | 81.80 | $2.75 \pm 1.96$ | $-3.29 \pm 1.02$ |
| | AdamW | 1 | CE | 2157. | $0.00 \pm 0.01$ | $13.9 \pm 5.49$ | $-19.4 \pm 7.18$ | 99.98 | 87.42 | $8.29 \pm 9.45$ | $-19.2 \pm 7.20$ |
| | | | BCE | 2863. | $2.15 \pm 0.02$ | $17.6 \pm 4.92$ | $-25.7 \pm 7.34$ | 99.98 | 99.97 | $8.49 \pm 11.2$ | $-23.5 \pm 7.67$ |
| | | 1&2 | CE | 1334. | $0.00 \pm 0.02$ | $4.42 \pm 0.67$ | $-1.96 \pm 0.87$ | 99.69 | 97.86 | $3.06 \pm 1.90$ | $-2.25 \pm 1.07$ |
| | | | BCE | 1440. | $2.18 \pm 0.02$ | $3.81 \pm 0.82$ | $-4.27 \pm 0.80$ | 99.67 | 99.22 | $2.28 \pm 1.96$ | $-4.21 \pm 0.93$ |
| DenseNet121 | SGD | 1 | CE | 337.9 | $-0.00 \pm 0.02$ | $12.9 \pm 3.33$ | $-0.12 \pm 2.83$ | 92.46 | 56.75 | $10.5 \pm 4.75$ | $-0.10 \pm 2.82$ |
| | | | BCE | 383.8 | $2.95 \pm 0.02$ | $6.03 \pm 2.64$ | $-7.83 \pm 2.81$ | 92.85 | 87.12 | $3.36 \pm 4.28$ | $-7.34 \pm 2.91$ |
| | | 1&2 | CE | 143.2 | $-0.00 \pm 0.02$ | $4.62 \pm 1.67$ | $-0.04 \pm 1.01$ | 67.23 | 47.68 | $4.26 \pm 1.84$ | $-0.04 \pm 1.01$ |
| | | | BCE | 161.5 | $2.90 \pm 0.01$ | $2.14 \pm 1.68$ | $-2.95 \pm 1.04$ | 68.15 | 56.43 | $1.74 \pm 1.85$ | $-2.93 \pm 1.05$ |
| | AdamW | 1 | CE | 1090. | $-0.00 \pm 0.01$ | $9.39 \pm 3.69$ | $-12.3 \pm 4.98$ | 99.89 | 83.67 | $4.74 \pm 6.83$ | $-12.1 \pm 4.99$ |
| | | | BCE | 1146. | $2.17 \pm 0.01$ | $9.78 \pm 2.72$ | $-16.0 \pm 4.77$ | 99.86 | 99.55 | $3.66 \pm 7.20$ | $-14.6 \pm 5.03$ |
| | | 1&2 | CE | 430.2 | $-0.00 \pm 0.01$ | $3.82 \pm 1.13$ | $-2.00 \pm 1.00$ | 91.18 | 80.57 | $2.85 \pm 1.83$ | $-2.07 \pm 1.06$ |
| | | | BCE | 474.5 | $2.20 \pm 0.01$ | $2.70 \pm 1.16$ | $-3.85 \pm 0.89$ | 90.66 | 85.83 | $1.79 \pm 1.83$ | $-3.82 \pm 0.97$ |

# C. Proof of Theorem 3.2

Zhou et al. (2022b) have proved that the loss satisfying contrastive property can cause neural collapse. CE loss, focal loss, and label smoothing loss satisfy this property, while BCE does not, and we proof that BCE can also result in the neural collapse in this paper.

**Definition C.1.** (Contrastive property (Zhou et al., 2022b)). A loss function $\mathcal{L}(\boldsymbol{z})$ satisfies the contrastive property if there exists a function $\phi$ such that $\mathcal{L}(\boldsymbol{z})$ can be lower bounded by

$$\mathcal{L}(\boldsymbol{z}) \geq \phi\left(\sum_{\substack{j=1 \\ j \neq k}}^{K} \left(z_j - z_k\right)\right) \tag{59}$$

where the equality holds only when $z_j = z_\ell$ for $\forall j, \ell \neq k$, and the function $\phi(x)$ satisfies

$$x^\star = \arg\min_x \phi(x) + c|x| \tag{60}$$

is unique for $\forall c > 0$, and $x^\star \leq 0$. ∎

**Theorem C.2.** (*Zhou et al., 2022b*) *Assume that the feature dimension $d$ is larger than the category number $K$, i.e., $d \geq K - 1$, and $\mathcal{L}$ is satisfying the contrastive property. Then any global minimizer $(\boldsymbol{W}^\star, \boldsymbol{H}^\star, \boldsymbol{b}^\star)$ of $f(\boldsymbol{W}, \boldsymbol{H}, \boldsymbol{b})$ defined using $\mathcal{L}$ with Eq. (3) obeys the following properties,*

$$\|\boldsymbol{w}^\star\| = \|\boldsymbol{w}_1^\star\| = \|\boldsymbol{w}_2^\star\| = \cdots = \|\boldsymbol{w}_K^\star\|, \tag{61}$$

$$\boldsymbol{h}_i^{(k)\star} = \sqrt{\frac{\lambda_{\boldsymbol{W}}}{n\lambda_{\boldsymbol{H}}}}\boldsymbol{w}_k^\star, \ \forall\, k \in [K], \ i \in [n], \tag{62}$$

$$\tilde{\boldsymbol{h}}_i^\star := \frac{1}{K}\sum_{j=1}^{K} \boldsymbol{h}_i^{(k)\star} = \boldsymbol{0}, \forall\, i \in [n], \tag{63}$$

$$\boldsymbol{b}^\star = b^\star \boldsymbol{1}_K, \tag{64}$$

*where either $b^\star = 0$ or $\lambda_{\boldsymbol{b}} = 0$. The matrix $\boldsymbol{W}^{\star T}$ forms a $K$-simplex ETF in the sense that*

$$\frac{1}{\|\boldsymbol{w}^\star\|_2^2}\boldsymbol{W}^{\star T}\boldsymbol{W}^\star = \frac{K}{K-1}\left(\boldsymbol{I}_K - \frac{1}{K}\boldsymbol{1}_K\boldsymbol{1}_K^T\right), \tag{65}$$

*where $\boldsymbol{I}_K \in \mathbb{R}^{K \times K}$ denotes the identity matrix, $\boldsymbol{1}_K \in \mathbb{R}^K$ denotes the all ones vector.* ∎

Neural collapse for BCE loss

$$\mathcal{L}_{\text{bce}}(\boldsymbol{W}\boldsymbol{h}_i^{(k)} - \boldsymbol{b}, \boldsymbol{y}_k) = \log\left(1 + \exp(-\boldsymbol{w}_k^T\boldsymbol{h}_i^{(k)} + b_k)\right) + \sum_{\substack{j=1 \\ j \neq k}}^{K} \log\left(1 + \exp(\boldsymbol{w}_j^T\boldsymbol{h}_i^{(k)} - b_j)\right). \tag{66}$$

**Theorem C.3.** *Assume that the feature dimension $d$ is larger than the number of classes $K$, i.e., $d \geq K - 1$. Then any global minimizer $(\boldsymbol{W}^\star, \boldsymbol{H}^\star, \boldsymbol{b}^\star)$ of*

$$\min_{\boldsymbol{W}, \boldsymbol{H}, \boldsymbol{b}} f_{\text{bce}}(\boldsymbol{W}, \boldsymbol{H}, \boldsymbol{b}) := g_{\text{bce}}(\boldsymbol{W}\boldsymbol{H} - \boldsymbol{b}\boldsymbol{1}^T) + \frac{\lambda_{\boldsymbol{W}}}{2}\|\boldsymbol{W}\|_F^2 + \frac{\lambda_{\boldsymbol{H}}}{2}\|\boldsymbol{H}\|_F^2 + \frac{\lambda_{\boldsymbol{b}}}{2}\|\boldsymbol{b}\|_2^2 \tag{67}$$

*with*

$$g_{\text{bce}}(\boldsymbol{W}\boldsymbol{H} - \boldsymbol{b}\boldsymbol{1}^T) := \frac{1}{N}\sum_{k=1}^{K}\sum_{i=1}^{n} \mathcal{L}_{\text{bce}}(\boldsymbol{W}\boldsymbol{h}_i^{(k)} - \boldsymbol{b}, \boldsymbol{y}_k), \tag{68}$$

*obeys the following*

$$\|\boldsymbol{w}^\star\| = \|\boldsymbol{w}_1^\star\| = \|\boldsymbol{w}_2^\star\| = \cdots = \|\boldsymbol{w}_K^\star\|, \ \text{and} \ \boldsymbol{b}^\star = b^\star\boldsymbol{1}, \tag{69}$$

$$\boldsymbol{h}_i^{(k)\star} = \sqrt{\frac{\lambda_{\boldsymbol{W}}}{n\lambda_{\boldsymbol{H}}}}\boldsymbol{w}_k^\star, \ \forall\, k \in [K], \ i \in [n], \ \text{and} \ \tilde{\boldsymbol{h}}_i^\star := \frac{1}{K}\sum_{j=1}^{K} \boldsymbol{h}_i^{(k)\star} = \boldsymbol{0}, \forall\, i \in [n], \tag{70}$$

*and the matrix $\frac{1}{\|\boldsymbol{w}^\star\|_2}\boldsymbol{W}^{\star T}$ forms a $K$-simplex ETF in the sense that*

$$\frac{1}{\|\boldsymbol{w}^\star\|_2^2}\boldsymbol{W}^{\star T}\boldsymbol{W}^\star = \frac{K}{K-1}\left(\boldsymbol{I}_K - \frac{1}{K}\boldsymbol{1}_K\boldsymbol{1}_K^T\right), \tag{71}$$

*where $b^\star$ is the solution of equation*

$$\lambda_{\boldsymbol{b}}b = \frac{K-1}{K\left(1+\exp\left(b+\sqrt{\frac{\lambda_{\boldsymbol{W}}}{n\lambda_{\boldsymbol{H}}}}\frac{\rho}{K(K-1)}\right)\right)} - \frac{1}{K\left(1+\exp\left(\sqrt{\frac{\lambda_{\boldsymbol{W}}}{n\lambda_{\boldsymbol{H}}}}\frac{\rho}{K}-b\right)\right)}. \tag{72}$$

**Proof** *According to Lemma C.4, any critical point $(\boldsymbol{W},\boldsymbol{H},\boldsymbol{b})$ of $f(\boldsymbol{W},\boldsymbol{H},\boldsymbol{b})$ satisfies*

$$\boldsymbol{W}^T\boldsymbol{W} = \frac{\lambda_{\boldsymbol{H}}}{\lambda_{\boldsymbol{W}}}\boldsymbol{H}^T\boldsymbol{H}. \tag{73}$$

*Let $\rho = \|\boldsymbol{W}\|_F^2$ for any critical point $(\boldsymbol{W},\boldsymbol{H},\boldsymbol{b})$. Then, according to Lemma C.6, for any $c_1,c_2 \geq 0$,*

$$\begin{aligned}
&f_{\text{bce}}(\boldsymbol{W},\boldsymbol{H},\boldsymbol{b}) \\
&\geq \left[\lambda_{\boldsymbol{W}} - \left(\frac{2K-1}{N(1+c_2)} - \frac{1}{N(1+c_1)}\right)\sqrt{\frac{n\lambda_{\boldsymbol{W}}}{\lambda_{\boldsymbol{H}}}}\right]\rho - \frac{1}{2K\lambda_{\boldsymbol{b}}}\left(\frac{K-1}{1+c_2} - \frac{1}{1+c_1}\right)^2 + C
\end{aligned} \tag{74}$$

*where*

$$C = \frac{c_1}{1+c_1}\log\left(\frac{1+c_1}{c_1}\right) + \frac{\log(1+c_1)}{1+c_1} + \frac{K-1}{1+c_2}\left[c_2\log\left(\frac{1+c_2}{c_2}\right) + \log(1+c_2)\right]. \tag{75}$$

*According to Lemma C.7, the inequality (74) achieves its equality when*

$$\|\boldsymbol{w}_1\| = \|\boldsymbol{w}_2\| = \cdots = \|\boldsymbol{w}_K\|, \ \text{and} \ \boldsymbol{b} = b^\star\boldsymbol{1}, \tag{76}$$

$$\boldsymbol{h}_i^{(k)} = \sqrt{\frac{\lambda_{\boldsymbol{W}}}{n\lambda_{\boldsymbol{H}}}}\boldsymbol{w}_k, \ \forall \ k \in [K], \ i \in [n], \ \text{and} \ \tilde{\boldsymbol{h}}_i = \frac{1}{K}\sum_{k=1}^K \boldsymbol{h}_i^{(k)} = \boldsymbol{0}, \forall i \in [n], \tag{77}$$

$$\boldsymbol{W}\boldsymbol{W}^T = \frac{\rho}{K-1}\left(\boldsymbol{I}_K - \frac{1}{K}\boldsymbol{1}_K\boldsymbol{1}_K^T\right), \tag{78}$$

$$c_1 = \exp\left(\sqrt{\frac{\lambda_{\boldsymbol{W}}}{n\lambda_{\boldsymbol{H}}}}\frac{\rho}{K} - b^\star\right), \ \text{and} \ c_2 = \exp\left(b^\star + \sqrt{\frac{\lambda_{\boldsymbol{W}}}{n\lambda_{\boldsymbol{H}}}}\frac{\rho}{K(K-1)}\right), \tag{79}$$

*where $b^\star$ is the solution of equation*

$$\lambda_{\boldsymbol{b}}b = \frac{K-1}{K\left(1+\exp\left(b+\sqrt{\frac{\lambda_{\boldsymbol{W}}}{n\lambda_{\boldsymbol{H}}}}\frac{\rho}{K(K-1)}\right)\right)} - \frac{1}{K\left(1+\exp\left(\sqrt{\frac{\lambda_{\boldsymbol{W}}}{n\lambda_{\boldsymbol{H}}}}\frac{\rho}{K}-b\right)\right)}. \tag{80}$$

*According to Lemma C.8, the equation (80) in terms of $b$ has only one solution $b^\star$.*

*Given $\lambda_{\boldsymbol{W}},\lambda_{\boldsymbol{H}},\lambda_{\boldsymbol{b}} > 0$, $f_{\text{bce}}(\boldsymbol{W},\boldsymbol{H},\boldsymbol{b})$ is convex function, which achieves its minimum with finite $\boldsymbol{W},\boldsymbol{H},\boldsymbol{b}$. Therefore, the right side of inequality (74) is a consistent when $\lambda_{\boldsymbol{W}},\lambda_{\boldsymbol{H}},\lambda_{\boldsymbol{b}}$ are fixed and Eqs. (76, 77, 78, 79) hold, which finishes the proof.* ∎

**Lemma C.4.** *Any critical point $(\boldsymbol{W},\boldsymbol{H},\boldsymbol{b})$ of Eq. (67) obeys*

$$\boldsymbol{W}^T\boldsymbol{W} = \frac{\lambda_{\boldsymbol{H}}}{\lambda_{\boldsymbol{W}}}\boldsymbol{H}\boldsymbol{H}^T, \ \text{and} \ \|\boldsymbol{W}\|_F^2 = \frac{\lambda_{\boldsymbol{H}}}{\lambda_{\boldsymbol{W}}}\|\boldsymbol{H}\|_F^2. \tag{81}$$

**Proof** *See Lemma D.2 in reference (Zhu et al., 2021).* ∎

**Lemma C.5.** *For any $\boldsymbol{h}_i^{(k)}$ with $c_1, c_2 > 0$, the BCE loss is lower bounded by*

$$\mathcal{L}_{\text{bce}}(\boldsymbol{W}\boldsymbol{h}_i^{(k)}, \boldsymbol{y}_k) \geq \frac{1}{1 + c_1}\Big( -\boldsymbol{w}_k^T \boldsymbol{h}_i^{(k)} + b_k \Big) + \frac{1}{1 + c_2} \sum_{\substack{j=1 \\ j \neq k}}^{K} \Big( \boldsymbol{w}_j^T \boldsymbol{h}_i^{(k)} - b_j \Big) + C, \tag{82}$$

*where*

$$C = \frac{c_1}{1 + c_1} \log\left(\frac{1 + c_1}{c_1}\right) + \frac{\log\left(1 + c_1\right)}{1 + c_1} + \frac{K - 1}{1 + c_2}\left[ c_2 \log\left(\frac{1 + c_2}{c_2}\right) + \log\left(1 + c_2\right) \right]. \tag{83}$$

*The inequality becomes an equality when*

$$\boldsymbol{w}_j^T \boldsymbol{h}_i^{(k)} - b_j = \boldsymbol{w}_\ell^T \boldsymbol{h}_i^{(k)} - b_\ell, \ \ \forall\, j, \ell \neq k, \tag{84}$$

*and*

$$c_1 = \exp\Big( \boldsymbol{w}_k^T \boldsymbol{h}_i^{(k)} - b_k \Big), \tag{85}$$

$$c_2 = \exp\Big( b_j - \boldsymbol{w}_j^T \boldsymbol{h}_i^{(k)} \Big), \ \ j \neq k. \tag{86}$$

**Proof** *By the concavity of the $\log(1 + e^x)$, we have,*

$$\sum_{k=1}^{K} \log\left(1 + \exp(x_k)\right) \geq K \log\left(1 + \exp\left(\frac{\sum_{k=1}^{K} x_k}{K}\right)\right), \ \ \forall x_k \in \mathbb{R}. \tag{87}$$

*Then,*

$$\mathcal{L}_{\text{bce}}(\boldsymbol{W}\boldsymbol{h}_i^{(k)} + \boldsymbol{b}, \boldsymbol{y}_k) \tag{88}$$

$$= \ \log\left(1 + \exp(-\boldsymbol{w}_k^T \boldsymbol{h}_i^{(k)} + b_k)\right) + \sum_{\substack{j=1 \\ j \neq k}}^{K} \log\left(1 + \exp(\boldsymbol{w}_j^T \boldsymbol{h}_i^{(k)} - b_j)\right) \tag{89}$$

$$\geq \ \log\left(1 + \exp(-\boldsymbol{w}_k^T \boldsymbol{h}_i^{(k)} + b_k)\right) + \left(K - 1\right) \log\left[1 + \exp\left(\frac{\sum_{\substack{j=1 \\ j \neq k}}^{K} \left(\boldsymbol{w}_j^T \boldsymbol{h}_i^{(k)} - b_j\right)}{K - 1}\right)\right] \tag{90}$$

$$= \ \log\left(\frac{c_1}{1 + c_1}\frac{1 + c_1}{c_1} + \frac{1 + c_1}{1 + c_1} \exp\left( -\boldsymbol{w}_k^T \boldsymbol{h}_i^{(k)} + b_k \right)\right)$$

$$+ \left(K - 1\right) \log\left[\frac{c_2}{1 + c_2}\frac{1 + c_2}{c_2} + \frac{1 + c_2}{1 + c_2} \exp\left(\frac{\sum_{\substack{j=1 \\ j \neq k}}^{K} \left(\boldsymbol{w}_j^T \boldsymbol{h}_i^{(k)} - b_j\right)}{K - 1}\right)\right] \tag{91}$$

$$\geq \ \frac{c_1}{1 + c_1} \log\left(\frac{1 + c_1}{c_1}\right) + \frac{1}{1 + c_1} \log\left((1 + c_1) \exp\left( -\boldsymbol{w}_k^T \boldsymbol{h}_i^{(k)} + b_k \right)\right)$$

$$+ \left(K - 1\right)\left\{\frac{c_2}{1 + c_2} \log\left(\frac{1 + c_2}{c_2}\right) + \frac{1}{1 + c_2} \log\left[(1 + c_2) \exp\left(\frac{\sum_{\substack{j=1 \\ j \neq k}}^{K} \left(\boldsymbol{w}_j^T \boldsymbol{h}_i^{(k)} - b_j\right)}{K - 1}\right)\right]\right\} \tag{92}$$

$$= \ \frac{1}{1 + c_1}\left( -\boldsymbol{w}_k^T \boldsymbol{h}_i^{(k)} + b_k \right) + \frac{1}{1 + c_2} \sum_{\substack{j=1 \\ j \neq k}}^{K} \left( \boldsymbol{w}_j^T \boldsymbol{h}_i^{(k)} - b_j \right)$$

$$+ \underbrace{\frac{c_1}{1 + c_1} \log\left(\frac{1 + c_1}{c_1}\right) + \frac{\log\left(1 + c_1\right)}{1 + c_1} + \frac{K - 1}{1 + c_2}\left[ c_2 \log\left(\frac{1 + c_2}{c_2}\right) + \log\left(1 + c_2\right) \right]}_{C}. \tag{93}$$

*The first inequality is derived from the concavity of $\log(1 + e^x)$, i.e., Eq. (87), which achieves the equality if and only if*

$$\boldsymbol{w}_j^T \boldsymbol{h}_i^{(k)} - b_j = \boldsymbol{w}_\ell^T \boldsymbol{h}_i^{(k)} - b_\ell, \ \ \forall\, j, \ell \neq k \in [K]. \tag{94}$$

*The second inequality is derived from the concavity of $\log(x)$,*

$$\log\big(tx_1 + (1-t)x_2\big) \geq t\log(x_1) + (1-t)\log(x_2), \quad \forall x_1, x_2 \in \mathbb{R} \ \text{ and } \ t \in [0,1], \tag{95}$$

*which achieves its equality if and only if $x_1 = x_2$, or $t = 0$, or $t = 1$. Then, the second inequality holds for any $c_1, c_2 \geq 0$, and it becomes an equality if and only if*

$$\frac{1+c_1}{c_1} = (1+c_1)\exp\Big(-\boldsymbol{w}_k^T \boldsymbol{h}_i^{(k)} + b_k\Big) \ \text{ or } \ c_1 = 0 \ \text{ or } \ c_1 = +\infty, \ \text{ and} \tag{96}$$

$$\frac{1+c_2}{c_2} = (1+c_2)\exp\Bigg(\frac{\sum_{\substack{j=1 \\ j\neq k}}^{K}\big(\boldsymbol{w}_j^T \boldsymbol{h}_i^{(k)} - b_j\big)}{K-1}\Bigg) \ \text{ or } \ c_1 = 0 \ \text{ or } \ c_1 = +\infty. \tag{97}$$

*It is trivial when $c_1 = 0$ or $c_1 = +\infty$ or $c_2 = 0$ or $c_2 = +\infty$. Then, we get*

$$c_1 = \exp\Big(\boldsymbol{w}_k^T \boldsymbol{h}_i^{(k)} - b_k\Big), \tag{98}$$

$$c_2 = \exp\Bigg(\frac{\sum_{\substack{j=1 \\ j\neq k}}^{K}\big(b_j - \boldsymbol{w}_j^T \boldsymbol{h}_i^{(k)}\big)}{K-1}\Bigg) \overset{(94)}{=} \exp\Big(b_j - \boldsymbol{w}_j^T \boldsymbol{h}_i^{(k)}\Big), \ \ j \neq k, \tag{99}$$

*which are desired.* ∎

**Lemma C.6.** *Let*

$$\boldsymbol{W} = \big[\boldsymbol{w}_1, \boldsymbol{w}_2, \cdots, \boldsymbol{w}_K\big]^T \in \mathbb{R}^{K \times d}, \tag{100}$$

$$\boldsymbol{H} = \big[h_1^{(1)}, \cdots, h_n^{(1)}, \cdots, h_1^{(K)}, \cdots, h_n^{(K)}\big] \in \mathbb{R}^{d \times N} \tag{101}$$

*with $N = nK$. Then, for any critical point $(\boldsymbol{W}, \boldsymbol{H}, \boldsymbol{b})$ of Eq. (67) and any $c_1, c_2 \geq 0$, we have*

$$
\begin{aligned}
&f_{\text{bce}}(\boldsymbol{W}, \boldsymbol{H}, \boldsymbol{b}) \\
&\geq \left[\lambda_{\boldsymbol{W}} - \Big(\frac{1}{N(1+c_2)} + \frac{1}{N(1+c_1)}\Big)\sqrt{\frac{n\lambda_{\boldsymbol{W}}}{\lambda_{\boldsymbol{H}}}}\right]\rho - \frac{1}{2K\lambda_{\boldsymbol{b}}}\Big(\frac{K-1}{1+c_2} - \frac{1}{1+c_1}\Big)^2 + C
\end{aligned} \tag{102}
$$

*with $C = \frac{c_1}{1+c_1}\log\big(\frac{1+c_1}{c_1}\big) + \frac{\log(1+c_1)}{1+c_1} + \frac{K-1}{1+c_2}\big[c_2\log\big(\frac{1+c_2}{c_2}\big) + \log(1+c_2)\big]$.*

**Proof** *According to Lemma C.4, Eq. (82) holds for any $c_1, c_2 > 0$ and any $\boldsymbol{h}_i^{(k)}$ with $k \in [K]$, $i \in [n]$. We take the same $c_1$ and $c_2$ for all $\boldsymbol{h}_i^{(k)}$, then*

$$(1+c_1)(1+c_2)\big[g_{\text{bce}}(\boldsymbol{W}\boldsymbol{H} + \boldsymbol{b}\boldsymbol{1}^T) - C\big] \tag{103}$$

$$= (1+c_1)(1+c_2)\bigg[\frac{1}{N}\sum_{k=1}^{K}\sum_{i=1}^{n}\mathcal{L}_{\text{bce}}(\boldsymbol{W}\boldsymbol{h}_i^{(k)} + \boldsymbol{b}, \boldsymbol{y}_k) - C\bigg] \tag{104}$$

$$\geq \frac{1}{N}\sum_{k=1}^{K}\sum_{i=1}^{n}\bigg[(1+c_2)\Big(-\boldsymbol{w}_k^T \boldsymbol{h}_i^{(k)} + b_k\Big) + (1+c_1)\sum_{\substack{j=1 \\ j\neq k}}^{K}\Big(\boldsymbol{w}_j^T \boldsymbol{h}_i^{(k)} - b_j\Big)\bigg] \tag{105}$$

$$= \frac{1+c_1}{N}\sum_{k=1}^{K}\sum_{i=1}^{n}\sum_{\substack{j=1 \\ j\neq k}}^{K}\Big(\boldsymbol{w}_j^T \boldsymbol{h}_i^{(k)} - b_j\Big) - \frac{1+c_2}{N}\sum_{k=1}^{K}\sum_{i=1}^{n}\Big(\boldsymbol{w}_k^T \boldsymbol{h}_i^{(k)} - b_k\Big) \tag{106}$$

$$= \frac{1+c_1}{N}\sum_{k=1}^{K}\sum_{i=1}^{n}\bigg(\sum_{j=1}^{K}\Big(\boldsymbol{w}_j^T \boldsymbol{h}_i^{(k)} - b_j\Big) - \boldsymbol{w}_k^T \boldsymbol{h}_i^{(k)} + b_k\bigg) - \frac{1+c_2}{N}\sum_{k=1}^{K}\sum_{i=1}^{n}\Big(\boldsymbol{w}_k^T \boldsymbol{h}_i^{(k)} - b_k\Big) \tag{107}$$

$$= \frac{1+c_1}{N}\sum_{k=1}^{K}\sum_{i=1}^{n}\sum_{j=1}^{K}\Big(\boldsymbol{w}_j^T \boldsymbol{h}_i^{(k)} - b_j - \boldsymbol{w}_k^T \boldsymbol{h}_i^{(k)} + b_k\Big) + \frac{1+c_1}{N}\sum_{k=1}^{K}\sum_{i=1}^{n}\sum_{\substack{j=1 \\ j\neq k}}^{K}\Big(\boldsymbol{w}_k^T \boldsymbol{h}_i^{(k)} - b_k\Big)$$

$$- \frac{1+c_2}{N} \sum_{k=1}^{K} \sum_{i=1}^{n} \left( \boldsymbol{w}_k^T \boldsymbol{h}_i^{(k)} - b_k \right) \tag{108}$$

$$= \frac{1+c_1}{N} \left[ \sum_{k=1}^{K} \sum_{i=1}^{n} \sum_{j=1}^{K} \left( \boldsymbol{w}_j^T \boldsymbol{h}_i^{(k)} - b_j \right) - \sum_{k=1}^{K} \sum_{i=1}^{n} \sum_{j=1}^{K} \left( \boldsymbol{w}_k^T \boldsymbol{h}_i^{(k)} - b_k \right) \right]$$

$$+ \left( \frac{1+c_1}{N}(K-1) - \frac{1+c_2}{N} \right) \sum_{k=1}^{K} \sum_{i=1}^{n} \boldsymbol{w}_k^T \boldsymbol{h}_i^{(k)} - \left( \frac{1+c_1}{N}(K-1) - \frac{1+c_2}{N} \right) \sum_{k=1}^{K} \sum_{i=1}^{n} b_k \tag{109}$$

$$= \frac{1+c_1}{N} \sum_{i=1}^{n} \left[ \sum_{k=1}^{K} \left( \sum_{j=1}^{K} \boldsymbol{w}_k^T \boldsymbol{h}_i^{(j)} - K \boldsymbol{w}_k^T \boldsymbol{h}_i^{(k)} \right) \underbrace{- \sum_{k=1}^{K} \sum_{j=1}^{K} b_j + \sum_{k=1}^{K} \sum_{j=1}^{K} b_k}_{0} \right]$$

$$+ \left( \frac{1+c_1}{N}(K-1) - \frac{1+c_2}{N} \right) \sum_{k=1}^{K} \sum_{i=1}^{n} \boldsymbol{w}_k^T \boldsymbol{h}_i^{(k)} - \left( \frac{1+c_1}{K}(K-1) - \frac{1+c_2}{K} \right) \sum_{k=1}^{K} b_k \tag{110}$$

$$= \frac{1+c_1}{n} \sum_{i=1}^{n} \sum_{k=1}^{K} \boldsymbol{w}_k^T \left( \tilde{\boldsymbol{h}}_i - \boldsymbol{h}_i^{(k)} \right) + \left( \frac{1+c_1}{N}(K-1) - \frac{1+c_2}{N} \right) \sum_{k=1}^{K} \sum_{i=1}^{n} \boldsymbol{w}_k^T \boldsymbol{h}_i^{(k)}$$

$$- \left( \frac{1+c_1}{K}(K-1) - \frac{1+c_2}{K} \right) \sum_{k=1}^{K} b_k \tag{111}$$

where $\tilde{\boldsymbol{h}}_i = \frac{1}{K} \sum_{k=1}^{K} \boldsymbol{h}_i^{(k)}$.

According to the AM-GM inequality, we have

$$\boldsymbol{u}^T \boldsymbol{v} \geq -\frac{c}{2} \|\boldsymbol{u}\|_2^2 - \frac{1}{2c} \|\boldsymbol{v}\|_2^2, \ \forall \ \boldsymbol{u}, \ \boldsymbol{v} \in \mathbb{R}^d, \ \forall \ c \geq 0. \tag{112}$$

Then,

$$(1+c_1)(1+c_2) \left[ g_{\mathrm{bce}}(\boldsymbol{W}\boldsymbol{H} + \boldsymbol{b}\boldsymbol{1}^T) - C \right]$$

$$\geq -\frac{1+c_1}{n} \left( \frac{c_3}{2} \sum_{i=1}^{n} \sum_{k=1}^{K} \|\boldsymbol{w}_k\|_2^2 + \frac{1}{2c_3} \sum_{i=1}^{n} \sum_{k=1}^{K} \left\| \tilde{\boldsymbol{h}}_i - \boldsymbol{h}_i^{(k)} \right\|_2^2 \right)$$

$$- \left( \frac{1+c_1}{N}(K-1) - \frac{1+c_2}{N} \right) \left( \frac{c_4}{2} \sum_{k=1}^{K} \sum_{i=1}^{n} \|\boldsymbol{w}_k\|_2^2 + \frac{1}{2c_4} \sum_{k=1}^{K} \sum_{i=1}^{n} \|\boldsymbol{h}_i^{(k)}\|_2^2 \right)$$

$$- \left( \frac{1+c_1}{K}(K-1) - \frac{1+c_2}{K} \right) \sum_{k=1}^{K} b_k \tag{113}$$

$$= -\frac{1+c_1}{n} \left[ \frac{c_3}{2} \sum_{i=1}^{n} \sum_{k=1}^{K} \|\boldsymbol{w}_k\|_2^2 + \frac{1}{2c_3} \sum_{i=1}^{n} \left( \sum_{k=1}^{K} \left\| \boldsymbol{h}_i^{(k)} \right\|_2^2 - K \|\tilde{\boldsymbol{h}}_i\|_2^2 \right) \right]$$

$$- \left( \frac{1+c_1}{N}(K-1) - \frac{1+c_2}{N} \right) \left( \frac{c_4}{2} \sum_{k=1}^{K} \sum_{i=1}^{n} \|\boldsymbol{w}_k\|_2^2 + \frac{1}{2c_4} \sum_{k=1}^{K} \sum_{i=1}^{n} \|\boldsymbol{h}_i^{(k)}\|_2^2 \right)$$

$$- \left( \frac{1+c_1}{K}(K-1) - \frac{1+c_2}{K} \right) \sum_{k=1}^{K} b_k \tag{114}$$

$$= -\frac{1+c_1}{n} \left( \frac{c_3}{2} \sum_{i=1}^{n} \sum_{k=1}^{K} \|\boldsymbol{w}_k\|_2^2 + \frac{1}{2c_3} \sum_{i=1}^{n} \sum_{k=1}^{K} \left\| \boldsymbol{h}_i^{(k)} \right\|_2^2 \right)$$

$$- \left( \frac{1+c_1}{N}(K-1) - \frac{1+c_2}{N} \right) \left( \frac{c_4}{2} \sum_{k=1}^{K} \sum_{i=1}^{n} \|\boldsymbol{w}_k\|_2^2 + \frac{1}{2c_4} \sum_{k=1}^{K} \sum_{i=1}^{n} \|\boldsymbol{h}_i^{(k)}\|_2^2 \right)$$

$$- \left( \frac{1+c_1}{K}(K-1) - \frac{1+c_2}{K} \right) \sum_{k=1}^{K} b_k + \frac{1+c_1}{2nc_3} \sum_{i=1}^{n} K \|\tilde{\boldsymbol{h}}_i\|_2^2 \tag{115}$$

$$= - \frac{1+c_1}{n} \left( \frac{nc_3}{2} \|\boldsymbol{W}\|_F^2 + \frac{1}{2c_3} \|\boldsymbol{H}\|_F^2 \right)$$

$$- \left( \frac{1+c_1}{N}(K-1) - \frac{1+c_2}{N} \right) \left( \frac{nc_4}{2} \|\boldsymbol{W}\|_F^2 + \frac{1}{2c_4} \|\boldsymbol{H}\|_F^2 \right)$$

$$- \left( \frac{1+c_1}{K}(K-1) - \frac{1+c_2}{K} \right) \sum_{k=1}^{K} b_k + \frac{1+c_1}{2nc_3} \sum_{i=1}^{n} K \|\tilde{\boldsymbol{h}}_i\|_2^2 \tag{116}$$

*and the inequality becomes an equality if and only if*

$$c_3 \boldsymbol{w}_k = \boldsymbol{h}_i^{(k)} - \tilde{\boldsymbol{h}}_i, \quad \forall \, k \in [K], \ i \in [n], \ \text{ and} \tag{117}$$

$$c_4 \boldsymbol{w}_k = -\boldsymbol{h}_i^{(k)}, \qquad \forall \, k \in [K], \ i \in [n], \tag{118}$$

*which can be achieved only when* $\tilde{\boldsymbol{h}}_i = \boldsymbol{0}$.

*Let* $\rho = \|\boldsymbol{W}\|_F^2$. *Then, by using Lemma C.4, we have* $\|\boldsymbol{H}\|_F^2 = \frac{\lambda_{\boldsymbol{W}}}{\lambda_{\boldsymbol{H}}} \rho$, *and*

$$f_{\text{bce}}(\boldsymbol{W}, \boldsymbol{H}, \boldsymbol{b})$$

$$= g_{\text{bce}}(\boldsymbol{W}\boldsymbol{H} + \boldsymbol{b}\boldsymbol{1}^T) + \frac{\lambda_{\boldsymbol{W}}}{2} \|\boldsymbol{W}\|_F^2 + \frac{\lambda_{\boldsymbol{H}}}{2} \|\boldsymbol{H}\|_F^2 + \frac{\lambda_{\boldsymbol{b}}}{2} \|\boldsymbol{b}\|_2^2 \tag{119}$$

$$\geq - \frac{1}{n(1+c_2)} \left( \frac{nc_3}{2} \|\boldsymbol{W}\|_F^2 + \frac{1}{2c_3} \|\boldsymbol{H}\|_F^2 \right)$$

$$- \left( \frac{K-1}{N(1+c_2)} - \frac{1}{N(1+c_1)} \right) \left( \frac{nc_4}{2} \|\boldsymbol{W}\|_F^2 + \frac{1}{2c_4} \|\boldsymbol{H}\|_F^2 \right)$$

$$- \left( \frac{K-1}{K(1+c_2)} - \frac{1}{K(1+c_1)} \right) \sum_{k=1}^{K} b_k + \frac{1}{2nc_3(1+c_2)} \sum_{i=1}^{n} K \|\tilde{\boldsymbol{h}}_i\|_2^2 + C$$

$$+ \frac{\lambda_{\boldsymbol{W}}}{2} \rho + \frac{\lambda_{\boldsymbol{H}}}{2} \frac{\lambda_{\boldsymbol{W}}}{\lambda_{\boldsymbol{H}}} \rho + \frac{\lambda_{\boldsymbol{b}}}{2} \|\boldsymbol{b}\|_2^2 \tag{120}$$

$$= - \frac{1}{n(1+c_2)} \left( \frac{nc_3}{2} \rho + \frac{1}{2c_3} \frac{\lambda_{\boldsymbol{W}}}{\lambda_{\boldsymbol{H}}} \rho \right) - \left( \frac{K-1}{N(1+c_2)} - \frac{1}{N(1+c_1)} \right) \left( \frac{nc_4}{2} \rho + \frac{1}{2c_4} \frac{\lambda_{\boldsymbol{W}}}{\lambda_{\boldsymbol{H}}} \rho \right)$$

$$- \left( \frac{K-1}{K(1+c_2)} - \frac{1}{K(1+c_1)} \right) \sum_{k=1}^{K} b_k + \frac{1}{2nc_3(1+c_2)} \sum_{i=1}^{n} K \|\tilde{\boldsymbol{h}}_i\|_2^2 + C + \lambda_{\boldsymbol{W}} \rho + \frac{\lambda_{\boldsymbol{b}}}{2} \|\boldsymbol{b}\|_2^2 \tag{121}$$

$$= \left[ \lambda_{\boldsymbol{W}} - \frac{1}{n(1+c_2)} \left( \frac{nc_3}{2} + \frac{1}{2c_3} \frac{\lambda_{\boldsymbol{W}}}{\lambda_{\boldsymbol{H}}} \right) - \left( \frac{K-1}{N(1+c_2)} - \frac{1}{N(1+c_1)} \right) \left( \frac{nc_4}{2} + \frac{1}{2c_4} \frac{\lambda_{\boldsymbol{W}}}{\lambda_{\boldsymbol{H}}} \right) \right] \rho$$

$$+ \frac{\lambda_{\boldsymbol{b}}}{2} \|\boldsymbol{b}\|_2^2 - \left( \frac{K-1}{K(1+c_2)} - \frac{1}{K(1+c_1)} \right) \sum_{k=1}^{K} b_k + \frac{1}{2nc_3(1+c_2)} \sum_{i=1}^{n} K \|\tilde{\boldsymbol{h}}_i\|_2^2 + C \tag{122}$$

$$= \left[ \lambda_{\boldsymbol{W}} - \frac{1}{n(1+c_2)} \left( \frac{nc_3}{2} + \frac{1}{2c_3} \frac{\lambda_{\boldsymbol{W}}}{\lambda_{\boldsymbol{H}}} \right) - \left( \frac{K-1}{N(1+c_2)} - \frac{1}{N(1+c_1)} \right) \left( \frac{nc_4}{2} + \frac{1}{2c_4} \frac{\lambda_{\boldsymbol{W}}}{\lambda_{\boldsymbol{H}}} \right) \right] \rho$$

$$+ \frac{\lambda_{\boldsymbol{b}}}{2} \sum_{k=1}^{K} \left[ b_k - \frac{1}{\lambda_{\boldsymbol{b}}} \left( \frac{K-1}{K(1+c_2)} - \frac{1}{K(1+c_1)} \right) \right]^2 - \frac{1}{2\lambda_{\boldsymbol{b}}} \sum_{k=1}^{K} \left( \frac{K-1}{K(1+c_2)} - \frac{1}{K(1+c_1)} \right)^2$$

$$+ \frac{1}{2nc_3(1+c_2)} \sum_{i=1}^{n} K \|\tilde{\boldsymbol{h}}_i\|_2^2 + C \tag{123}$$

$$\geq \left[ \lambda_{\boldsymbol{W}} - \frac{1}{n(1+c_2)} \left( \frac{nc_3}{2} + \frac{1}{2c_3} \frac{\lambda_{\boldsymbol{W}}}{\lambda_{\boldsymbol{H}}} \right) - \left( \frac{K-1}{N(1+c_2)} - \frac{1}{N(1+c_1)} \right) \left( \frac{nc_4}{2} + \frac{1}{2c_4} \frac{\lambda_{\boldsymbol{W}}}{\lambda_{\boldsymbol{H}}} \right) \right] \rho$$

$$+ \frac{\lambda_b}{2} \sum_{k=1}^{K} \left[ b_k - \frac{1}{\lambda_b} \left( \frac{K-1}{K(1+c_2)} - \frac{1}{K(1+c_1)} \right) \right]^2 - \frac{1}{2K\lambda_b} \left( \frac{K-1}{1+c_2} - \frac{1}{1+c_1} \right)^2 + C \tag{124}$$

$$\geq \left[ \lambda_W - \frac{1}{n(1+c_2)} \left( \frac{nc_3}{2} + \frac{1}{2c_3} \frac{\lambda_W}{\lambda_H} \right) - \left( \frac{K-1}{N(1+c_2)} - \frac{1}{N(1+c_1)} \right) \left( \frac{nc_4}{2} + \frac{1}{2c_4} \frac{\lambda_W}{\lambda_H} \right) \right] \rho$$

$$- \frac{1}{2K\lambda_b} \left( \frac{K-1}{1+c_2} - \frac{1}{1+c_1} \right)^2 + C, \tag{125}$$

*where the inequality (124) achieves its equality if and only if*

$$\tilde{h}_i = \mathbf{0}, \ \ \forall i \in [n], \tag{126}$$

*and the inequality (125) becomes an equality whenever either*

$$\lambda_b = 0 \ \ or \ \ b_k = \frac{1}{\lambda_b} \left( \frac{K-1}{K(1+c_2)} - \frac{1}{K(1+c_1)} \right), \ \ \forall k \in [K]. \tag{127}$$

*Due to $\lambda_b > 0$ and $c_1, c_2$ are same for any $k \in [K]$, therefore*

$$b_k = b_j, \ \ \forall k, j \in [K]. \tag{128}$$

*Based on Eqs. (117) and (126), we have*

$$c_3 \boldsymbol{w}_k = \boldsymbol{h}_i^{(k)} \Rightarrow c_3^2 = \frac{\sum_{i=1}^n \sum_{k=1}^K \|\boldsymbol{h}_i^{(k)}\|_2^2}{\sum_{i=1}^n \sum_{k=1}^K \|\boldsymbol{w}_k\|_2^2} = \frac{\|\boldsymbol{H}\|_F^2}{n\|\boldsymbol{W}\|_F^2} = \frac{\lambda_W}{n\lambda_H} \Rightarrow c_3 = \sqrt{\frac{\lambda_W}{n\lambda_H}}; \tag{129}$$

*similarly, from Eq. (118), we get*

$$c_4 \boldsymbol{w}_k = -\boldsymbol{h}_i^{(k)} \Rightarrow c_4^2 = \frac{\sum_{i=1}^n \sum_{k=1}^K \|\boldsymbol{h}_i^{(k)}\|_2^2}{\sum_{i=1}^n \sum_{k=1}^K \|\boldsymbol{w}_k\|_2^2} = \frac{\|\boldsymbol{H}\|_F^2}{n\|\boldsymbol{W}\|_F^2} = \frac{\lambda_W}{n\lambda_H} \Rightarrow c_4 = -\sqrt{\frac{\lambda_W}{n\lambda_H}}. \tag{130}$$

*Plugging them into Eq. (122), we get*

$$f_{\text{bce}}(\boldsymbol{W}, \boldsymbol{H}, \boldsymbol{b})$$

$$\geq \left[ \lambda_W - \frac{1}{n(1+c_2)} \left( \frac{nc_3}{2} + \frac{1}{2c_3} \frac{\lambda_W}{\lambda_H} \right) - \left( \frac{K-1}{N(1+c_2)} - \frac{1}{N(1+c_1)} \right) \left( \frac{nc_4}{2} + \frac{1}{2c_4} \frac{\lambda_W}{\lambda_H} \right) \right] \rho$$

$$- \frac{1}{2K\lambda_b} \left( \frac{K-1}{1+c_2} - \frac{1}{1+c_1} \right)^2 + C \tag{131}$$

$$= \left[ \lambda_W - \left( \frac{1}{n(1+c_2)} - \frac{K-1}{N(1+c_2)} + \frac{1}{N(1+c_1)} \right) \left( \frac{n}{2} \sqrt{\frac{\lambda_W}{n\lambda_H}} + \frac{1}{2} \sqrt{\frac{n\lambda_H}{\lambda_W}} \frac{\lambda_W}{\lambda_H} \right) \right] \rho$$

$$- \frac{1}{2K\lambda_b} \left( \frac{K-1}{1+c_2} - \frac{1}{1+c_1} \right)^2 + C \tag{132}$$

$$= \left[ \lambda_W - \left( \frac{1}{n(1+c_2)} - \frac{K-1}{N(1+c_2)} + \frac{1}{N(1+c_1)} \right) \sqrt{\frac{n\lambda_W}{\lambda_H}} \right] \rho$$

$$- \frac{1}{2K\lambda_b} \left( \frac{K-1}{1+c_2} - \frac{1}{1+c_1} \right)^2 + C \tag{133}$$

$$= \left[ \lambda_W - \left( \frac{1}{N(1+c_2)} + \frac{1}{N(1+c_1)} \right) \sqrt{\frac{n\lambda_W}{\lambda_H}} \right] \rho - \frac{1}{2K\lambda_b} \left( \frac{K-1}{1+c_2} - \frac{1}{1+c_1} \right)^2 + C \tag{134}$$

*which is desired.* ∎

**Lemma C.7.** *Under the same assumptions of Lemma C.6, the lower bound in Eq. (102) is achieved for any critical point* $(\boldsymbol{W}, \boldsymbol{H}, \boldsymbol{b})$ *of Eq. (67) if and only if the following hold*

$$\|\boldsymbol{w}_1\| = \|\boldsymbol{w}_2\| = \cdots = \|\boldsymbol{w}_K\|, \ \text{and} \ \boldsymbol{b} = b^\star \boldsymbol{1}, \tag{135}$$

$$\boldsymbol{h}_i^{(k)} = \sqrt{\frac{\lambda_{\boldsymbol{W}}}{n\lambda_{\boldsymbol{H}}}} \boldsymbol{w}_k, \ \forall\, k \in [K], \ i \in [n], \ \text{and} \ \tilde{\boldsymbol{h}}_i = \frac{1}{K}\sum_{k=1}^{K}\boldsymbol{h}_i^{(k)} = \boldsymbol{0}, \forall\, i \in [n], \tag{136}$$

$$\boldsymbol{W}\boldsymbol{W}^T = \frac{\rho}{K-1}\left(\boldsymbol{I}_K - \frac{1}{K}\boldsymbol{1}_K\boldsymbol{1}_K^T\right), \tag{137}$$

$$c_1 = \exp\left(\sqrt{\frac{\lambda_{\boldsymbol{W}}}{n\lambda_{\boldsymbol{H}}}}\frac{\rho}{K} - b^\star\right), \ \text{and} \ c_2 = \exp\left(b^\star + \sqrt{\frac{\lambda_{\boldsymbol{W}}}{n\lambda_{\boldsymbol{H}}}}\frac{\rho}{K(K-1)}\right), \tag{138}$$

*where* $b^\star$ *is the solution of equation*

$$\lambda_{\boldsymbol{b}} b = \left[\frac{K-1}{K\left(1 + \exp\left(b + \sqrt{\frac{\lambda_{\boldsymbol{W}}}{n\lambda_{\boldsymbol{H}}}}\frac{\rho}{K(K-1)}\right)\right)} - \frac{1}{K\left(1 + \exp\left(\sqrt{\frac{\lambda_{\boldsymbol{W}}}{n\lambda_{\boldsymbol{H}}}}\frac{\rho}{K} - b\right)\right)}\right]. \tag{139}$$

**Proof** *With the proof of Lemma C.6, to achieve the lower bound, it needs at least Eqs. (117), (118), and (126) to hold, i.e.,*

$$\tilde{\boldsymbol{h}}_i = \frac{1}{K}\sum_{k=1}^{K}\boldsymbol{h}_i^{(k)} = \boldsymbol{0}, \ \forall\, i \in [n], \ \text{and} \ \sqrt{\frac{\lambda_{\boldsymbol{W}}}{n\lambda_{\boldsymbol{H}}}}\boldsymbol{w}_k = \boldsymbol{h}_i^{(k)}, \ \forall\, k \in [K], \ i \in [n], \tag{140}$$

*and further implies*

$$\sum_{k=1}^{K}\boldsymbol{w}_k = \sqrt{\frac{n\lambda_{\boldsymbol{H}}}{\lambda_{\boldsymbol{W}}}}\sum_{k=1}^{K}\boldsymbol{h}_i^{(k)} = \boldsymbol{0}. \tag{141}$$

*Then,*

$$c_1 = \exp\left(\boldsymbol{w}_k^T\boldsymbol{h}_i^{(k)} - b_k\right) = \exp\left(\sqrt{\frac{\lambda_{\boldsymbol{W}}}{n\lambda_{\boldsymbol{H}}}}\|\boldsymbol{w}_k\|_2^2 - b_k\right), \ \forall k \in [K], \tag{142}$$

$$c_2 = \exp\left(b_j - \boldsymbol{w}_j^T\boldsymbol{h}_i^{(k)}\right) = \exp\left(b_j - \sqrt{\frac{\lambda_{\boldsymbol{W}}}{n\lambda_{\boldsymbol{H}}}}\boldsymbol{w}_k^T\boldsymbol{w}_j\right), \ \forall j \neq k \in [K], \tag{143}$$

*Since that* $c_1, c_2$ *are chosen to be the same for any* $j \neq k \in [K]$*, therefore,*

$$\sqrt{\frac{\lambda_{\boldsymbol{W}}}{n\lambda_{\boldsymbol{H}}}}\|\boldsymbol{w}_k\|_2^2 - b_k = \sqrt{\frac{\lambda_{\boldsymbol{W}}}{n\lambda_{\boldsymbol{H}}}}\|\boldsymbol{w}_j\|_2^2 - b_j, \ \forall k, j \in [K], \tag{144}$$

$$\sqrt{\frac{\lambda_{\boldsymbol{W}}}{n\lambda_{\boldsymbol{H}}}}\boldsymbol{w}_k^T\boldsymbol{w}_j - b_j = \sqrt{\frac{\lambda_{\boldsymbol{W}}}{n\lambda_{\boldsymbol{H}}}}\boldsymbol{w}_k^T\boldsymbol{w}_\ell - b_\ell, \ \forall j \neq \ell \in [K], \forall k \in [K], \tag{145}$$

*With the proof of Lemma C.5, to achieve the lower bound, it needs at least Eqs. (94) to hold, then,*

$$\sqrt{\frac{\lambda_{\boldsymbol{W}}}{n\lambda_{\boldsymbol{H}}}}\|\boldsymbol{w}_k\|_2^2 - b_k$$

$$\stackrel{(141)}{=} -\sqrt{\frac{\lambda_{\boldsymbol{W}}}{n\lambda_{\boldsymbol{H}}}}\sum_{\substack{j=1 \\ j \neq k}}^{K}\boldsymbol{w}_j^T\boldsymbol{w}_k - b_k \tag{146}$$

$$\stackrel{(145)}{=} -\sqrt{\frac{\lambda_{\boldsymbol{W}}}{n\lambda_{\boldsymbol{H}}}}\sum_{\substack{j=1 \\ j \neq k}}^{K}\boldsymbol{w}_k^T\boldsymbol{w}_\ell - b_k + \sum_{\substack{j=1 \\ j \neq k}}(b_\ell - b_j) \tag{147}$$

$$= -(K-1)\sqrt{\frac{\lambda_{\boldsymbol{W}}}{n\lambda_{\boldsymbol{H}}}} \underbrace{\boldsymbol{w}_k^T \boldsymbol{w}_\ell}_{\ell \neq k} -2b_k + (K-1)b_\ell - K\bar{b} \tag{148}$$

$$\overset{(144,145)}{\Longrightarrow} -2b_k + (K-1)b_\ell - K\bar{b} = -2b_\ell + (K-1)b_j - K\bar{b} \tag{149}$$

$$\Longleftrightarrow b_k = b_\ell, \ \ \forall \ell \neq k \in [K], \tag{150}$$

*which is conforming to Eq. (128) when $\lambda_{\boldsymbol{b}} > 0$. Then, combining with Eqs. (144) and (141),*

$$\left\|\boldsymbol{w}_k\right\|_2^2 = \left\|\boldsymbol{w}_j\right\|_2^2 = \frac{\left\|\boldsymbol{W}\right\|_F^2}{K} = \frac{\rho}{K}, \ \ \forall k, j \in [K], \tag{151}$$

$$\left\|\boldsymbol{w}_k\right\|_2^2 = -(K-1)\sum_{\substack{j=1 \\ j \neq k}}^{K} \boldsymbol{w}_k^T \boldsymbol{w}_j \Rightarrow \boldsymbol{w}_k^T \boldsymbol{w}_j = -\frac{1}{K-1}\frac{\rho}{K}, \ \ \forall j \neq k \in [K]. \tag{152}$$

*Therefore,*

$$\boldsymbol{W}\boldsymbol{W}^T = \frac{\rho}{K-1}\left(\boldsymbol{I}_K - \frac{1}{K}\boldsymbol{1}_K\boldsymbol{1}_K^T\right). \tag{153}$$

*Plugging (151) and (152) into (142) and (143)*

$$c_1 = \exp\left(\sqrt{\frac{\lambda_{\boldsymbol{W}}}{n\lambda_{\boldsymbol{H}}}}\frac{\rho}{K} - b\right), \tag{154}$$

$$c_2 = \exp\left(b + \sqrt{\frac{\lambda_{\boldsymbol{W}}}{n\lambda_{\boldsymbol{H}}}}\frac{\rho}{K(K-1)}\right), \tag{155}$$

*where $b = b_k = b_j$. When $\lambda_{\boldsymbol{b}} > 0$, substitute Eqs. (154) and (155) into (127), we have*

$$b = \frac{1}{\lambda_{\boldsymbol{b}}}\left(\frac{K-1}{K(1+c_2)} - \frac{1}{K(1+c_1)}\right) \tag{156}$$

$$= \frac{1}{\lambda_{\boldsymbol{b}}}\left[\frac{K-1}{K\left(1 + \exp\left(b + \sqrt{\frac{\lambda_{\boldsymbol{W}}}{n\lambda_{\boldsymbol{H}}}}\frac{\rho}{K(K-1)}\right)\right)} - \frac{1}{K\left(1 + \exp\left(\sqrt{\frac{\lambda_{\boldsymbol{W}}}{n\lambda_{\boldsymbol{H}}}}\frac{\rho}{K} - b\right)\right)}\right]. \tag{157}$$

*When $\lambda_{\boldsymbol{b}} = 0$, substitute Eq. (140) into*

$$\frac{\partial f_{\text{bce}}}{\partial b_k} = \frac{1}{nK}\left(n - \sum_{j=1}^{K}\sum_{i=1}^{n}\frac{1}{1 + e^{-\boldsymbol{w}_k\boldsymbol{h}_i^{(j)} + b_k}}\right) = 0, \ \ \forall k \in [K], \tag{158}$$

*we have*

$$0 = \frac{K-1}{K\left(1 + \exp\left(b + \sqrt{\frac{\lambda_{\boldsymbol{W}}}{n\lambda_{\boldsymbol{H}}}}\frac{\rho}{K(K-1)}\right)\right)} - \frac{1}{K\left(1 + \exp\left(\sqrt{\frac{\lambda_{\boldsymbol{W}}}{n\lambda_{\boldsymbol{H}}}}\frac{\rho}{K} - b\right)\right)}, \tag{159}$$

*by combining with Eqs. (151) and (152).* ∎

**Lemma C.8.** *The equation*

$$\lambda_{\boldsymbol{b}}\, b = \frac{K-1}{K\left(1 + \exp\left(b + \sqrt{\frac{\lambda_{\boldsymbol{W}}}{n\lambda_{\boldsymbol{H}}}}\frac{\rho}{K(K-1)}\right)\right)} - \frac{1}{K\left(1 + \exp\left(\sqrt{\frac{\lambda_{\boldsymbol{W}}}{n\lambda_{\boldsymbol{H}}}}\frac{\rho}{K} - b\right)\right)} \tag{160}$$

*has only one solution.*

**Proof** *A number $b^\star$ is a solution of equation ([160](#)) if and only if it is a solution of*

$$\underbrace{\lambda_{\boldsymbol{b}} K b + \frac{1}{1 + \exp\left(\sqrt{\frac{\lambda_{\boldsymbol{W}}}{n\lambda_{\boldsymbol{H}}}} \frac{\rho}{K} - b\right)}}_{\beta_1(b)} = \underbrace{\frac{K - 1}{1 + \exp\left(b + \sqrt{\frac{\lambda_{\boldsymbol{W}}}{n\lambda_{\boldsymbol{H}}}} \frac{\rho}{K(K-1)}\right)}}_{\beta_2(b)} . \tag{161}$$

*When $\lambda_{\boldsymbol{b}} > 0$,*

$$\beta_1(b) \to -\infty, \quad \beta_2(b) \to K - 1 \quad as \ \ b \to -\infty \tag{162}$$
$$\beta_1(b) \to +\infty, \quad \beta_2(b) \to 0 \qquad as \ \ b \to +\infty, \tag{163}$$

*and if $\lambda_{\boldsymbol{b}} = 0$,*

$$\beta_1(b) = 0, \qquad \beta_2(b) \to K - 1 \quad as \ \ b \to -\infty \tag{164}$$
$$\beta_1(b) \to +\infty, \quad \beta_2(b) \to 0 \qquad as \ \ b \to +\infty. \tag{165}$$

*Therefore, the curves of $\beta_1(b)$ and $\beta_2(b)$ must intersect at least once in the plane, i.e., the equations ([160](#)) and ([161](#)) have solutions.*

*In addition,*

$$\frac{\mathrm{d}\beta_1(b)}{\mathrm{d}b} = \lambda_{\boldsymbol{b}} K + \frac{\exp\left(\sqrt{\frac{\lambda_{\boldsymbol{W}}}{n\lambda_{\boldsymbol{H}}}} \frac{\rho}{K} - b\right)}{\left(1 + \exp\left(\sqrt{\frac{\lambda_{\boldsymbol{W}}}{n\lambda_{\boldsymbol{H}}}} \frac{\rho}{K} - b\right)\right)^2} > 0, \tag{166}$$

$$\frac{\mathrm{d}\beta_2(b)}{\mathrm{d}b} = -\frac{(K-1)\exp\left(b + \sqrt{\frac{\lambda_{\boldsymbol{W}}}{n\lambda_{\boldsymbol{H}}}} \frac{\rho}{K(K-1)}\right)}{\left(1 + \exp\left(b + \sqrt{\frac{\lambda_{\boldsymbol{W}}}{n\lambda_{\boldsymbol{H}}}} \frac{\rho}{K(K-1)}\right)\right)^2} < 0, \tag{167}$$

*i.e., $\beta_1(b)$ is strictly increasing, while $\beta_2(b)$ is strictly decreasing. Therefore, they can intersect at only one point.* ∎

**Lemma C.9.** *When the class number $K > 2$ and*

$$\lambda_{\boldsymbol{b}} \sqrt{\frac{\lambda_{\boldsymbol{W}}}{n\lambda_{\boldsymbol{H}}}} \frac{\rho}{K-1} + \frac{1}{2(K-1)} > \frac{1}{1 + \exp\left(\sqrt{\frac{\lambda_{\boldsymbol{W}}}{n\lambda_{\boldsymbol{H}}}} \frac{\rho}{K-1}\right)}, \tag{168}$$

*the final critical bias $b^\star$ could uniformly separate the all positive decision scores*

$$\left\{ \boldsymbol{w}_k^{\star T} \boldsymbol{h}_i^{(k)\star} : k \in [K], i \in [n] \right\} \tag{169}$$

*and the all negative decision scores*

$$\left\{ \boldsymbol{w}_j^{\star T} \boldsymbol{h}_i^{(k)\star} : k, j \in [K], i \in [n], k \neq j \right\}, \tag{170}$$

*where*

$$\boldsymbol{W}^\star = \left[ \boldsymbol{w}_1^\star, \boldsymbol{w}_2^\star, \cdots, \boldsymbol{w}_K^\star \right]^T \tag{171}$$
$$\boldsymbol{H}^\star = \left[ \boldsymbol{h}_1^{(1)\star}, \cdots, \boldsymbol{h}_n^{(1)\star}, \cdots, \boldsymbol{h}_1^{(K)\star}, \cdots, \boldsymbol{h}_n^{(K)\star} \right] \tag{172}$$
$$\boldsymbol{b}^\star = (b^\star, b^\star, \cdots, b^\star)^T = b^\star \mathbf{1}_K \tag{173}$$

*form the critical point of function $f(\boldsymbol{W}, \boldsymbol{H}, \boldsymbol{b})$ in Eq. ([67](#)).*

**Proof** *According to Lemma [C.7](#), for the critical point $(\boldsymbol{W}^\star, \boldsymbol{H}^\star, \boldsymbol{b}^\star)$, we have*

$$\boldsymbol{w}_k^{\star T} \boldsymbol{h}_i^{(k)\star} = \sqrt{\frac{\lambda_{\boldsymbol{W}}}{n\lambda_{\boldsymbol{H}}}} \frac{\rho}{K}, \quad \forall k \in [K], i \in [n] \tag{174}$$

$$\boldsymbol{w}_j^{\star T} \boldsymbol{h}_i^{(k)\star} = -\sqrt{\frac{\lambda_{\boldsymbol{W}}}{n\lambda_{\boldsymbol{H}}}} \frac{\rho}{K(K-1)}, \quad \forall k, j \in [K], i \in [n], k \neq j. \tag{175}$$

*Let $b_{\mathrm{neg}} = -\sqrt{\frac{\lambda_W}{n\lambda_H}}\frac{\rho}{K(K-1)}$, $b_{\mathrm{pos}} = \sqrt{\frac{\lambda_W}{n\lambda_H}}\frac{\rho}{K}$. Then, the critical $b^\star$ separating the all positive and negative score if and only if*

$$b_{\mathrm{neg}} = -\sqrt{\frac{\lambda_W}{n\lambda_H}}\frac{\rho}{K(K-1)} < b^\star < \sqrt{\frac{\lambda_W}{n\lambda_H}}\frac{\rho}{K} = b_{\mathrm{pos}} \tag{176}$$

*which, according to the proof of Lemma C.8, is equivalent to*

$$\beta_1(b_{\mathrm{neg}}) < \beta_2(b_{\mathrm{neg}}) \ \text{ and } \ \beta_1(b_{\mathrm{pos}}) > \beta_2(b_{\mathrm{pos}}). \tag{177}$$

*Due to*

$$\beta_1(b_{\mathrm{neg}}) < \beta_2(b_{\mathrm{neg}}) \Leftrightarrow -\lambda_b\sqrt{\frac{\lambda_W}{n\lambda_H}}\frac{\rho}{K-1} + \frac{1}{1+\exp\left(\sqrt{\frac{\lambda_W}{n\lambda_H}}\frac{\rho}{K-1}\right)} < \frac{K-1}{2}$$

$$\Leftarrow \frac{1}{1+\mathrm{e}^0} < \frac{K-1}{2} \Leftarrow 2 < K \tag{178}$$

$$\beta_1(b_{\mathrm{pos}}) > \beta_2(b_{\mathrm{pos}}) \Leftrightarrow \lambda_b\sqrt{\frac{\lambda_W}{n\lambda_H}}\rho + \frac{1}{2} > \frac{K-1}{1+\exp\left(\sqrt{\frac{\lambda_W}{n\lambda_H}}\frac{\rho}{K-1}\right)} \tag{179}$$

$$\Leftrightarrow \lambda_b\sqrt{\frac{\lambda_W}{n\lambda_H}}\frac{\rho}{K-1} + \frac{1}{2(K-1)} > \frac{1}{1+\exp\left(\sqrt{\frac{\lambda_W}{n\lambda_H}}\frac{\rho}{K-1}\right)}, \tag{180}$$

*it completes the proof.* ∎

## D. More discussion about decision scores in the training

In the training, the decision scores are updated along the negative direction of their gradients during the back propagation stage, i.e.,

$$\boldsymbol{w}_k^T \boldsymbol{h}^{(k)} \;\leftarrow\; \boldsymbol{w}_k^T \boldsymbol{h}^{(k)} - \eta \frac{\partial f_\mu(\boldsymbol{W},\boldsymbol{H},\boldsymbol{b})}{\partial\big(\boldsymbol{w}_k^T \boldsymbol{h}^{(k)}\big)}, \quad \forall k \in [K], \tag{181}$$

$$\boldsymbol{w}_j^T \boldsymbol{h}^{(k)} \;\leftarrow\; \boldsymbol{w}_j^T \boldsymbol{h}^{(k)} - \eta \frac{\partial f_\mu(\boldsymbol{W},\boldsymbol{H},\boldsymbol{b})}{\partial\big(\boldsymbol{w}_j^T \boldsymbol{h}^{(k)}\big)}, \quad \forall j \neq k \in [K], \tag{182}$$

where $\eta$ is the learning rate, and $\mu \in \{\mathrm{ce}, \mathrm{bce}\}$.

In the training with CE, the updating formulas are

$$\boldsymbol{w}_k^T \boldsymbol{h}^{(k)} \;\leftarrow\; \boldsymbol{w}_k^T \boldsymbol{h}^{(k)} + \eta \left( 1 - \frac{\mathrm{e}^{\boldsymbol{w}_k^T \boldsymbol{h}^{(k)} - b_k}}{\sum_\ell \mathrm{e}^{\boldsymbol{w}_\ell^T \boldsymbol{h}^{(k)} - b_\ell}} \right), \tag{183}$$

$$\boldsymbol{w}_j^T \boldsymbol{h}^{(k)} \;\leftarrow\; \boldsymbol{w}_j^T \boldsymbol{h}^{(k)} - \eta \frac{\mathrm{e}^{\boldsymbol{w}_j^T \boldsymbol{h}^{(k)} - b_j}}{\sum_\ell \mathrm{e}^{\boldsymbol{w}_\ell^T \boldsymbol{h}^{(k)} - b_\ell}}. \tag{184}$$

Then, for the samples with diverse initial decision scores, it is difficult to update their decision scores to the similar level, if they own the similar predicted probabilities belong to each categories.

In the training with BCE, the updating formulas are

$$\boldsymbol{w}_k^T \boldsymbol{h}^{(k)} \;\leftarrow\; \boldsymbol{w}_k^T \boldsymbol{h}^{(k)} + \eta \left( 1 - \frac{1}{1 + \mathrm{e}^{-\boldsymbol{w}_k^T \boldsymbol{h}^{(k)} + b_k}} \right), \tag{185}$$

$$\boldsymbol{w}_j^T \boldsymbol{h}^{(k)} \;\leftarrow\; \boldsymbol{w}_j^T \boldsymbol{h}^{(k)} - \eta \frac{1}{1 + \mathrm{e}^{-\boldsymbol{w}_j^T \boldsymbol{h}^{(k)} + b_j}}. \tag{186}$$

Then, for the sample with small positive decision score $\boldsymbol{w}_k \boldsymbol{h}^{(k)}$, its predicted probability $\frac{1}{1+\mathrm{e}^{-\boldsymbol{w}_k^T \boldsymbol{h}^{(k)} + b_k}}$ to its category will be also small, and the score updating amplitude $\eta \left( 1 - \frac{1}{1+\mathrm{e}^{-\boldsymbol{w}_k^T \boldsymbol{h}^{(k)} + b_k}} \right)$ will be large; in contrary, for the sample with large positive score $\boldsymbol{w}_k^T \boldsymbol{h}^{(k)}$, the probability $\frac{1}{1+\mathrm{e}^{-\boldsymbol{w}_k^T \boldsymbol{h}^{(k)} + b_k}}$ will be also large, and the updating amplitude $\eta \left( 1 - \frac{1}{1+\mathrm{e}^{-\boldsymbol{w}_k^T \boldsymbol{h}^{(k)} + b_k}} \right)$ will be small. This property helps to update the all positive decision scores to be in uniform high level.

Similarly, for the sample with large negative decision score $\boldsymbol{w}_j^T \boldsymbol{h}^{(k)}$, its predicted probability $\frac{1}{1+\mathrm{e}^{-\boldsymbol{w}_j^T \boldsymbol{h}^{(k)} + b_j}}$ to other category will be also large, so is the score updating amplitude $\eta \frac{1}{1+\mathrm{e}^{-\boldsymbol{w}_j^T \boldsymbol{h}^{(k)} + b_j}}$; in contrary, for the sample with small negative score $\boldsymbol{w}_j^T \boldsymbol{h}^{(k)}$, the probability $\frac{1}{1+\mathrm{e}^{-\boldsymbol{w}_j^T \boldsymbol{h}^{(k)} + b_k}}$ will be small, so is the updating amplitude $\eta \left( 1 - \frac{1}{1+\mathrm{e}^{-\boldsymbol{w}_j^T \boldsymbol{h}^{(k)} + b_k}} \right)$. This property helps to update the all negative decision scores to be in uniform low level.

