# OpenReview forum: "BCE vs. CE in Deep Feature Learning"
_ICML.cc/2025/Conference — ICML 2025 poster_

### Official Review · Reviewer_RW5r · 2025-03-10

**Overall Recommendation:** 3

**Summary:**

The paper compares Binary Cross-Entropy (BCE) and Cross-Entropy (CE) loss functions in deep feature learning, focusing on their ability to enhance intra-class compactness and inter-class distinctiveness. It theoretically proves that BCE, like CE, can lead to Neural Collapse (NC) when minimized, maximizing these properties. BCE is found to explicitly enhance feature compactness and distinctiveness by adjusting decision scores across samples, whereas CE implicitly improves features by classifying samples one-by-one. Experimental results show BCE improves classification performance and leads to faster NC.

**Claims And Evidence:**

Yes.

**Essential References Not Discussed:**

No.

**Experimental Designs Or Analyses:**

- The experiments mainly focus on CNNs such as ResNet and DenseNet. It would be better to provide more experiments results for Transformer-based models such as ViT.

**Methods And Evaluation Criteria:**

Yes.

**Other Comments Or Suggestions:**

- There are some typos. For example, in the abstract, where "the leaned features are compact" should be "the learned features are compact".

**Other Strengths And Weaknesses:**

**Strengths**

- The paper is well-written and organized.

- The paper provides a novel theoretical analysis demonstrating that BCE, like CE, can lead to neural collapse when minimized. This fills a gap in understanding BCE’s behavior.

**Weakness**

- While the paper experimentally validates BCE's advantages on long-tailed datasets, it does not provide a  theoretical analysis on imbalanced scenarios.

- The authors rely heavily on "decision scores" to assess compactness and distinctiveness of features, but as acknowledged by the authors themselves, these scores are not direct measurements of feature compactness or distinctiveness.

**Questions For Authors:**

Please see above.

**Relation To Broader Scientific Literature:**

The proposed method contributes to the broader scientific literature by providing a theoretical analysis of BCE from the perspective of neural collapse.

**Theoretical Claims:**

Yes.

---

> ### Author Rebuttal · Authors · 2025-03-31
>
> # **Thanks to reviewer RW5r for the comments!**
>
> # Response to “Experimental Designs Or Analyses”: Experiments on Transformer
>
> To further validate the advantages of BCE over CE, we trained ViT [1] and Swin Transformer [2] using BCE and CE on CIFAR10 and CIFAR100. Similar to the experimental setting in Section 4.1 of the paper, we conducted two groups of experiments: 1) with a fixed $\lambda_b=0$, setting the mean of the initialized classifier biases to 0, 1, 2, 3, 4, 5, 6, 8, and 10, respectively; 2) with varying $\lambda_b = 0.5, 0.05, 5\times10^{-3}, 5\times10^{-4}, 5\times10^{-5}$, and $5\times10^{-6}$, respectively, setting the mean of initialized classifier biases to 10.
> After the training, we visualized the distributions of unbiased positive and negative decision scores and classifier biases using violin plots for the different ViTs and Swin Transformers. One can find the results through the following anonymous link:
> https://anonymous.4open.science/r/BCE-vs-CE-6F45/
>
> From the results, we obtained the conclusions similar to those from CNNs: for models trained with CE, the final classifier biases have no substantial relationship with the unbiased decision scores of the sample features, which primarily depend on their initial values and regularization coefficients $\lambda_b$. In contrast, for models trained with BCE, regardless of the model or optimizer used, there is a clear correlation between the final classifier biases and the unbiased positive and negative decision scores.
> These results indicate that the classifier bias of BCE substantially affects the positive and negative decision scores, thereby affecting the compactness and distinctiveness of the sample features, which is consistent with the analysis in the paper.
>
> [1] A. Dosovitskiy, et al, An Image is Worth 16x16 Words: Transformers for Image Recognition at Scale, ICLR2021
>
> [2] Z. Liu, et al, Swin Transformer: Hierarchical Vision Transformer using Shifted Windows, ICCV2021
>
> # Response to “Weaknesses”:
>
> **1. theoretical analysis on imbalanced scenarios**
>
> On imbalanced datasets, the theoretically analyzing for neural collapse of loss functions is a more challenging task. Currently, the most theoretical analyses of loss function with neural collapse are conducted on balanced datasets, such as [1,2,3,4,5]. Some theoretical works [6] on neural collapse even simplify a class of sample features into a single vector. Currently, we only found few papers [7,8] that rigorously analyze the neural collapse for loss functions on imbalanced datasets. We are also investigating the neural collapse of BCE on imbalanced datasets, and we hope to theoretically address this issue in the near future.
>
> [1] Z. Zhu, et al, A Geometric Analysis of Neural Collapse with Unconstrained Features, NeurIPS 2021.
>
> [2] J. Zhou, et al, Are All Losses Created Equal: A Neural Collapse Perspective, NeurIPS 2022.
>
> [3] M. Munn, et al, The Impact of Geometric Complexity on Neural Collapse in Transfer Learning, NeurIPS2024
>
> [4] P. Li, et al, Neural Collapse in Multi-label Learning with Pick-all-label Loss, ICML 2024
>
> [5] J. Jiang, et al, Generalized Neural Collapse for a Large Number of Classes, ICML 2024
>
> [6] J. Lu and S. Steinerberger, Neural Collapse Under Cross-entropy Loss, Applied and Computational Harmonic Analysis, 2022
>
> [7] C. Fang, et al, Exploring Deep Neural Networks via Layer-peeled Model: Minority Collapse in Imbalanced Training, PNAS 2021
>
> [8] H. Dang, et al, Neural Collapse for Cross-entropy Class-Imbalanced Learning with Unconstrained ReLU Features Mode, ICML 2024
>
> **2. decision scores**
>
> Although decision scores do not directly measure the intra-class compactness and inter-class differences of features, they can indirectly reflect these two properties by anchoring to the classifier vectors $[w_k]_{k=1}^K$. In practice, the loss functions based on decision scores, such as CE and focal loss, etc., are prevalent in enhancing the feature properties. Therefore, this paper relies on decision scores to analyze CE and BCE.
> Currently, to fill the gap between "decision score" and “feature property”, we are theoretically analyzing CE and BCE, which directly measure the inner product or cosine similarity among sample features in contrastive learning.
>
> # Response to “Other Comments Or Suggestions”: Typos
>
> Thanks to the reviewer for pointing out the typos in the paper. We have conducted a thorough review to avoid such issues in our manuscript.

---

### Official Review · Reviewer_bxWz · 2025-03-10

**Overall Recommendation:** 4

**Summary:**

This paper shows that binary cross-entropy (BCE) loss, like cross-entropy (CE) loss, can induce neural collapse—maximize intra-class compactness and inter-class distinctiveness in multi-class tasks when the loss reaches its minimum. Through theoretical and empirical analysis, the authors show that models trained with BCE outperform those using CE. This advantage stems from BCE's classifier bias, which explicitly optimizes feature alignment toward class centers, enhancing intra-class compactness and inter-class distinctiveness. This finding reveals that BCE not only promotes beneficial feature distributions but also boosts model performance, outperforming CE in both classification and retrieval tasks due to its ability to learn more compact and separated features.

**Claims And Evidence:**

Yes.
Sections 3 and 4 provide theoretical and empirical analysis respectively.

**Essential References Not Discussed:**

No. The related work is sufficient.

**Experimental Designs Or Analyses:**

Section 5 about  Transformer and other deep architectures is indeed overly concise, lacking experimental validation. The authors did not personally verify the performance of BCE and CE in Transformer architectures, which diminishes the comprehensiveness and persuasiveness of the study.

**Methods And Evaluation Criteria:**

Yes.

**Other Comments Or Suggestions:**

It is recommended that the authors extend their experimental validation to include other architectures beyond those currently studied. For instance, exploring parameter-efficient fine-tuning (PEFT) methods, such as Visual Prompt Tuning, LoRA, or similar approaches, would significantly enhance the compelling evidence of the study.

**Other Strengths And Weaknesses:**

Strengths:
1. The paper presents a comprehensive theoretical analysis comparing the capabilities of BCE and CE in achieving neural collapse. It demonstrates that BCE can effectively attain high intra-class compactness and inter-class separation, underscoring BCE's robust theoretical potential in feature learning.
2. Beyond its theoretical contributions, the paper delves into the practical distinctions between BCE and CE, particularly in terms of decision score dynamics and boundary updates. The analysis reveals that BCE provides more consistent updates to decision scores, which imposes stronger constraints on feature representations. This practical insight enhances our understanding of how BCE can refine feature learning, offering a clear explanation for its superior performance in improving feature alignment and separation.

Weaknesses:
1. Figure 1 lacks corresponding experimental verification. t-SNE visualizations can be utilized to validate the feature distributions of CE and BCE.
2.  The authors focus exclusively on the original softmax formulation in their comparison of CE and BCE, concluding that CE measures the relative values of decision scores, while BCE adjusts the absolute values of positive/negative decision scores uniformly across all samples. What about the cosine similarity used in A-Softmax loss [1] and ArcFace[2]? These methods explicitly optimize the angular margin between classes, potentially leading to different decision scores compared to the standard softmax-based approaches. When employing cosine-based softmax, does BCE still retain its advantages in feature learning?

[1] W. Liu, et al., SphereFace: Deep Hypersphere Embedding for Face Recognition, CVPR 2017.
[2] J. Deng, et al., ArcFace: Additive Angular Margin Loss for Deep Face Recognition, CVPR 2019.

**Questions For Authors:**

The abbreviation "LTR" mentioned in Sec. 5 "Transformer and other deep architectures" lacks a clear explanation, and its full form is not provided anywhere in the text.

**Relation To Broader Scientific Literature:**

This paper demonstrates that BCE loss not only fosters a beneficial feature distribution but also boosts model performance. If the conclusion of this study is validated on Transformer-based architectures, it could advance existing research in several critical areas of computer vision, including imbalanced/long-tail learning[1], continual learning[2], noisy label[3] learning, and CV tasks.

[1] J. Shi, et al., Long-Tail Learning with Foundation Model: Heavy Fine-Tuning Hurts, in ICML 2024.

[2] L. Wang, et al., A Comprehensive Survey of Continual Learning: Theory, Method and Application, in TPAMI 2024.

[3] N. Natarajan, et al., Learning with Noisy Labels, in NIPS 2013.

**Theoretical Claims:**

Yes.

---

> ### Author Rebuttal · Authors · 2025-03-31
>
> # Thanks to reviewer bxWz for the comments!
>
> # Response to “Experimental Designs Or Analyses”: Experiments on Transformers
>
> As required by Reviewers **bxWz** and **RW5r**, we train ViT and Swin Transformer using BCE and CE on CIFAR10 and CIFAR100, to further validate the advantages of BCE over CE.
> The experimental setting and conclusions can be referenced in our response to reviewer **RW5r**.
> The results can be found through: https://anonymous.4open.science/r/BCE-vs-CE-6F45/
>
> # Response to “Supplementary Material”:
>
> We submitted a 24-page PDF file as supplementary material in the OpenReview, which includes the metrics used in our experiments, additional experimental results, and detailed proofs of the theorems presented in the paper.
>
> # Response to “Weaknesses”:
> **1. t-SNE visualizations for the feature distributions**
>
> We thank the reviewer for suggesting the use of t-SNE visualization to compare the feature distributions of CE and BCE. Using t-SNE, we dynamically demonstrate the feature distributions extracted by ResNet18 trained with BCE and CE on CIFAR10, from the first epoch to the 30th epoch. One can find the results through the following anonymous link:
> https://anonymous.4open.science/r/BCE-vs-CE-6F45/
>
> Although both CE and BCE can lead to neural collapse, ultimately maximizing the intra-class compactness and inter-class differences of the features, BCE leads to neural collapse more quickly during the feature learning. We provide a static display of the features from epochs 12 and 13 for both CE and BCE. One can observe that, compared to CE, the features from BCE at these two epochs are already distributed in distinct regions, within a more compact distribution for each class. In contrast, CE shows unclear boundaries between features of different classes, and the feature distribution within the same class is relatively loose.
>
> **2. CE and BCE using cosine similarity and margin**
>
> When using cosine similarity as the unbiased decision score, the CE loss still couples the positive and negative decision scores of every sample within one Softmax, constraining their relative values. In this regard, it is fundamentally no different from the original CE, which uses the inner product as the unbiased decision score.
>
> When an angular margin is added, the Softmax-based CE requires that there be at least a margin between the positive and negative decision scores for each sample. Although this margin is consistent across all samples, only when the marginal intervals between the positive and negative decision scores of all samples overlap, a unified threshold that can distinguish the positive and negative decision scores for all samples exists, to ensure that the positive decision scores of all samples are at a uniformly high level (as we have explained in the response to reviewer **eGz4** regarding the connection between ``uniformly high level’’ and intra-class compactness), while their negative decision scores are at a uniformly low level. In total, introducing the margin can indirectly enhance the intra-class compactness and inter-class disparity of sample features. In contrast, as explained in Eq. 16 of the paper, BCE uses the unified parameters (i.e., classifier biases) to directly constrain the decision scores of samples within the same class to be at uniformly high level or uniformly low level.
>
> In practice, although the gains brought by an appropriate margin to CE are generally higher than the gains of BCE compared to CE, the margin strategies can also be integrated into BCE for higher performance. This is discussed in more depth with experimental validation in SphereFace2 [1] and UniFace [2], both of which take BCE-based loss functions with cosine similarity and margin, and they achieved better face recognition performance than CE-based ArcFace, A-softmax, and CosFace.
>
> [1] Wen, Y., et al. SphereFace2: Binary classification is all you need for deep face recognition. ICLR 2022.
>
> [2] Zhou, J., et al. UniFace: Unified cross-entropy loss for deep face recognition. ICCV2023.
>
> # Response to “Comments Or Suggestions”: experiments with other architectures:
> We sincerely appreciate the reviewer for suggesting us to validate the advantages of BCE over CE across more methods and model architectures. We have conducted preliminary explorations of the advantages of BCE over CE using ViT and Swin Transformer. We believe that the benefits of BCE are universal, and in the future, we will consider using BCE in methods including visual prompt tuning and LoRA.
>
> # Response to “Questions For Authors”: LTR
> LTR is short for long-tailed recognition, which refers to the recognition tasks on imbalanced long-tailed datasets. We apologize for not including its full name in the paper, and we will add it in the revised manuscript.

---

> > ### Comment · Reviewer_bxWz · 2025-04-02
> >
> > Thanks for the authors' effort. I apologize for the carelessness. I initially thought the supplementary material was to be included with the main paper, but I later found it in a separate file. After reviewing the supplementary, I found the theoretical proof rigorous and well-structured.
> >
> > This work makes a valuable contribution by establishing a solid theoretical foundation, which could potentially support more challenging research directions such as imbalanced/long-tailed learning and OOD generalization, to name a few.
> >
> > Given its contributions, I lean to accept this paper.

---

> > > ### Author Response · Authors · 2025-04-02
> > >
> > > Thanks very much for your response and increasing the score!

---

### Official Review · Reviewer_eGz4 · 2025-03-15

**Overall Recommendation:** 3

**Summary:**

This paper provides a comparative analysis of Binary Cross-Entropy (BCE) and Cross-Entropy (CE) losses in the context of deep feature learning. The authors investigate whether BCE can lead to neural collapse (NC)—a phenomenon where intra-class variability collapses, class centers form a simplex equiangular tight frame, and classifier vectors align with their class centers. The study offers both theoretical proofs and empirical evidence demonstrating that BCE, like CE, can maximize intra-class compactness and inter-class distinctiveness. Furthermore, the authors argue that BCE explicitly enhances feature properties during training, unlike CE, which implicitly achieves this effect through classification decisions.

**Claims And Evidence:**

Most of claims are well supported by either the theoretical proof or emprical experiments. However, The authors claims that the BCE achieve better feature compactness and distinctiveness, resulting in higher classification performance. However, the better feature compactness and distinctiveness and the relationship with the accuracy is studied on the small dataset, such as CIFAR10 and CIFAR100. For larger dataset like ImageNet, the author only reports the classification, whether this claim is hold on this dataset in unclear.

**Essential References Not Discussed:**

There is some Neural collapse papers study different losses functions that are not cited in the papers, like MSE and contrastive loss.
[1] Zhou J, Li X, Ding T, et al. On the optimization landscape of neural collapse under mse loss: Global optimality with unconstrained features[C]//International Conference on Machine Learning. PMLR, 2022: 27179-27202.
[2] Xue Y, Joshi S, Gan E, et al. Which features are learnt by contrastive learning? On the role of simplicity bias in class collapse and feature suppression[C]//International Conference on Machine Learning. PMLR, 2023: 38938-38970.
[3] Jiang R, Nguyen T, Aeron S, et al. On neural and dimensional collapse in supervised and unsupervised contrastive learning with hard negative sampling[J]. arXiv preprint arXiv:2311.05139, 2023.

**Experimental Designs Or Analyses:**

i check the soundness/validity of any experimental designs or analyses. Most of the metric are adopted from the previous neural collapse study.

**Methods And Evaluation Criteria:**

No new proposed methods and evaluation criteria.

**Other Comments Or Suggestions:**

On page 5, lines 246 and 255, the term "uniformly high level" is unclear, as it lacks a precise definition and an explicit connection to intra-class compactness. It is not immediately evident what aspect of the decision scores is being referred to as "uniformly high" and why this uniformity directly contributes to enhanced compactness within the same class.

To improve clarity, the authors should explicitly define what "uniformly high level" means in the context of decision scores and provide a more detailed explanation of how this uniformity leads to greater intra-class compactness.

**Other Strengths And Weaknesses:**

**Strengths**:
1. Theoretical Contribution: The paper provides a rigorous proof that BCE, despite its structural differences from CE, can also induce neural collapse. It offers a mathematical explanation for how classifier biases in BCE constrain decision scores, leading to improved feature compactness and distinctiveness.
2. Empirical Validation: The comparison across different architectures (ResNet18, ResNet50, DenseNet121) ensures the results are not model-specific. The study includes extensive experiments across multiple datasets (MNIST, CIFAR-10, CIFAR-100, ImageNet-1k), showcasing that BCE achieves better intra-class compactness, inter-class distinctiveness, and classification accuracy than CE. Experiments demonstrate that BCE achieves neural collapse faster than CE in the initial training stages.
3. Comprehensive Comparison: The study explains BCE and CE differences geometrically and analytically, providing a deeper understanding of how these loss functions influence feature learning. The authors provide clear visualizations (e.g., decision region comparisons) and numerical metrics (e.g., compactness and distinctiveness scores) to support their claims.
4. Well-written and organization

**Weaknesses**:
1. Limited Discussion on Compactness and Distinctiveness in ImageNet: While the authors demonstrate that BCE improves feature compactness and distinctiveness on smaller datasets such as CIFAR-10 and CIFAR-100, they provide only accuracy comparisons for larger-scale datasets like ImageNet-1k. Although BCE achieves slightly higher classification accuracy than CE on ImageNet, the paper lacks an in-depth analysis of whether BCE also enhances feature compactness and inter-class distinctiveness on large-scale datasets. Given that ImageNet has significantly more classes and higher complexity, it is important to verify whether the same advantages observed on smaller datasets hold in this larger setting.
2. Practical benefits of BCE loss: One of the key arguments in favor of BCE over CE is its ability to explicitly constrain decision scores, leading to bounded feature distributions. However, this advantage is not entirely clear in practical deep learning scenarios. Typically, models trained with cross-entropy (CE) loss already incorporate weight decay as a regularization mechanism, which naturally constrains features within a bounded region. As a result, the explicit bounding effect of BCE may not provide a significant additional benefit beyond what is already achieved through weight decay.

**Questions For Authors:**

See above weakness and suggestions

**Relation To Broader Scientific Literature:**

From my understanding, there is no direct connection with broader scientific literature.

**Theoretical Claims:**

I rougly read the proof in the appendix, which use similar proof procedure in the previous work but bot rigorously check the correctness of any proofs for theoretical claims.

---

> ### Author Rebuttal · Authors · 2025-03-31
>
> # Thanks to reviewer eGz4 for the comments!
>
> # Response to “Claims and evidence”: feature properties on ImageNet
> On the validation set of ImageNet, we calculated the feature properties for ResNet50, ResNet101, and DenseNet161 trained by CE and BCE in Table 3 of the paper, and the results are presented in the table below.
>
> | |R50| R50|R101|R101|D161|D161|
> |----|----|----|----|----|----|----|
> | |$\mathcal E_{com}$|$\mathcal E_{dis}$|$\mathcal E_{com}$|$\mathcal E_{dis}$|$\mathcal E_{com}$|$\mathcal E_{dis}$|
> |CE	|82.46|12.21|82.88|**13.14**|78.25|12.04|
> |BCE|**82.99**|**12.50**|**83.81**|12.97|**79.09**|**12.05**|
>
> In the table, $\mathcal E_{com}$ and $\mathcal E_{dis}$ stand for the intra-class compactness and inter-class distinctiveness, respectively. From the table, one can find that although the more classes and higher complexity result in a less significant gain in feature properties for BCE, it still enhances the feature properties extracted by the model in most cases compared to CE. Only the inter-class distinctiveness of the ResNet101 trained by BCE has decreased, while the compactness and distinctiveness of other models trained by BCE have improved.
>
> In addition to the above results on ImageNet, we also compared BCE and CE using ViT and Swin Transformer. Please refer to our responses to reviewers **bxWz** and **RW5r**.
>
> # Response to “Essential References Not Discussed”:
> We thank reviewer **eGz4** for providing three new references. The first paper analyzes the neural collapse (NC) of the MSE loss, while the latter two investigate NC in contrastive learning. These references are indeed relevant to our research and will help advance our theoretical analysis of the BCE in contrastive learning. Once again, we appreciate the reviewer’s suggestions, and we will cite these papers.
>
> # Response to “Weaknesses”:
> **1. Feature properties on ImageNet**
>
> In the response to “Claims and evidence”, we have presented the feature properties of ResNet50, ResNet101, and DenseNet161 trained with CE and BCE on ImageNet. Although ImageNet has more classes and higher complexity, the results shows that the features extracted by the models trained with BCE exhibit better intra-class compactness and inter-class distinctiveness.
>
> **2. BCE and CE with weight decay**
>
> The core argument of our paper is that the classifier bias in BCE plays a significant role in the deep feature learning, while the bias in CE does not, which helps BCE learn features with better properties. In Section 3.2, we theoretically demonstrate this viewpoint through the neural collapse of BCE and CE. In Section 3.3, to provide a simple and intuitive understanding of the above argument, we did not use the weight decay; however, in both the theoretical proof (line 162) in Section 3.2 and the experimental validation (line 258, line 305, and line 322) in Section 4, we did employ weight decay.
>
> Under the same theoretical framework and experimental setting, the theoretical analysis and experimental results in the paper both indicate that when using weight decay, BCE achieves better feature properties and classification results than CE in most cases.
>
> # Response to “Other Comments Or Suggestions”: uniformly high level
> "Uniformly high level" refers to the situation where the values expressed by the positive (unbiased) decision scores for different samples are consistently at a high level. For the class $k$, it means that the inner products between the different sample features $[h_i^{(k)}]_{i=1}^{n_k}$ and their corresponding classifier vector $w_k$ are all at high level. We believe that, in this case, the sample features within the class exhibit good intra-class compactness.
>
> To understand this, in first, we incorporated the weight decay in the theoretical analysis of the paper, ensuring that the L2 norms of the sample features do not increase indefinitely during the training. The theorems in the paper have also indicated that the L2 norms of different feature vectors will converge to a fixed value. Secondly, as
> $$‖w_k-h^{(k)}‖_2^2 = ‖w_k ‖_2^2 + ‖h^{(k)}‖_2^2 - 2w_k^T h^{(k)},$$
>
> for different sample features of class $k$, when they have the same norm, the large unbiased positive decision scores $\big[w_k^T h_i^{(k)}\big]_{i=1}^{n_k}$ implies that they are close to the classifier vector $w_k$. Therefore, if all the unbiased positive decision scores of different sample features of class $k$ are at a uniformly high level, it means all of these features are all close to $w_k$, resulting in small distances between them and thus high intra-class compactness. Conversely, if there is an obviously difference in the distances of the different sample features from $w_k$, it can be reasonably inferred that there will also be considerable differences in the distances between these features, leading to lower intra-class compactness.

---

> > ### Comment · Reviewer_eGz4 · 2025-04-05
> >
> > After reading the rebuttal and additional experiments, my concern is mainly addressed. Therefore, I will keep my score.

---

> > > ### Author Response · Authors · 2025-04-07
> > >
> > > We would like to thank the reviewer for taking the time to review our manuscript and rebuttal. We are glad to have addressed the reviewer’s concerns.

---

### Decision · Program_Chairs · 2025-05-01

**Decision:**

Accept (poster)

**Comment:**

This paper provides a comparative analysis of Cross-Entropy (CE) and Binary Cross-Entropy (BCE) losses in the context of deep feature learning. The authors first demonstrate that BCE, like CE, can lead to neural collapse (NC). They further argue that BCE explicitly enhances feature properties during training, whereas CE achieves this effect implicitly through classification decisions. Experimental results show that BCE improves classification performance and accelerates the onset of neural collapse.

Overall, all reviewers acknowledged the paper’s contribution to the theoretical and empirical analysis of BCE.